# The genomic origin of the unique chaetognath body plan

Laura Piovani[1], Daria Gavriouchkina[2], Elise Parey[1], Luke A. Sarre[3,4], Katja T. C. A. Peijnenburg[5,6], José María Martín-Durán[3,4], Daniel S. Rokhsar[7,8,9], Noriyuki Satoh[10], Alex de Mendoza[3,4], Taichiro Goto[11] & Ferdinand Marlétaz[1 ✉]

The emergence of animal phyla, each with their unique body plan, was a rapid event in the history of animal life, yet its genomic underpinnings are still poorly understood[1]. Here we investigate at the genomic, regulatory and cellular levels, the origin of one of the most distinctive animal phyla, the chaetognaths, whose organismal characteristics have historically complicated their phylogenetic placement[2,3]. We show that these characteristics are reflected at the cell-type level by the expression of genes that originated in the chaetognath lineage, contributing to adaptation to planktonic life at the sensory and structural levels[4]. Similarly to other members of gnathiferans (which also include rotifers and several other microscopic phyla)[5,6], chaetognaths have undergone accelerated genomic evolution with gene loss and chromosomal fusions[7,8]. Furthermore, they secondarily duplicated thousands of genes[9,10], without evidence for a whole-genome duplication, yielding, for instance, tandemly expanded Hox genes, as well as many phylum-specific genes. We also detected repeat-rich highly methylated neocentromeres and a simplified DNA methylation toolkit that is involved in mobile element repression rather than transcriptional control. Consistent with fossil evidence[11,12], our observations suggest that chaetognaths emerged after a phase of morphological simplification through a reinvention of organ systems paralleled by massive genomic reorganization, explaining the uniqueness of their body plan.

Whereas different scenarios have been proposed to account for the emergence of animal phyla such as chordates[13], arthropods[14] or annelids[15], other lineages resist attempts to do so. A prominent example is the chaetognaths, whose phylogenetic position in the tree of bilaterians has long been disputed[2]. Although chaetognaths do not share obvious similarities with any other extant animal group and their ancient fossil record does not provide obvious clues into their origins, their body plan has remained remarkably stable since the Cambrian period[11,12]. Patterns of embryonic development (enterocoely and secondary mouth opening)[16] originally suggested an association with deuterostomes, whereas other traits were reminiscent of protostomes (chitinous grasping spines and nervous system)[3,4]. After years of sometimes contradictory molecular analyses[10,17], chaetognaths have been assigned to the clade Gnathifera, which represents the sister group to the rest of lophotrochozoan animals within spiralians[5,6] (Fig. 1a,b). This placement was then corroborated by the shared presence of the *MedPost* Hox gene—a unique Hox gene bearing median and posterior molecular signatures—with rotifers, another prominent gnathiferan group[18,19], and was further supported by fossil interpretation[11,12,20].

Understanding the genomic events that led to the emergence of such an unusual body plan thus represents a key challenge to understanding the origin of animals itself. Recent studies have shown how genomic and regulatory changes have sometimes (for example, vertebrates[21] and annelids[7]) but not always (for example, echinoderms[22]) accompanied the emergence of novel bodyplans. Many lophotrochozoans, such as annelids, molluscs or lophophorates, retained ancestral bilaterian genomic characters, such as gene content, introns[23], or conserved chromosomal ancestral linkage[7,24]. However, within their sister group (the Gnathifera) the only lineage characterized from a genomic standpoint are the rotifers, which underwent fast genomic evolution associated with the acquisition of asexuality in bdelloids[25,26]. Determining how the evolution of genes and genomes shaped the cell-type complement of gnathiferan lineages is thus crucial to understand the evolution of organismal complexity[27]. To try to determine whether gnathiferans preserved some of the ancestral traits uncovered in other spiralians and to grasp the origin of the enigmatic chaetognath body plan, we focused on the free-living predatory chaetognath *Paraspadella gotoi*, and generated a chromosome-scale genome, regulatory profiling resources (Hi-C-seq, assay for transposase-accessible chromatin with high-throughput sequencing (ATAC–seq) and methylome data) as well as a single-cell sequencing atlas, which, together, reveal how changes in the gene repertoire and gene regulation underlined the cell-type bases

[1]Centre for Life's Origin and Evolution, Department of Genetics, Evolution and Environment, University College London, London, UK. [2]UK Dementia Research Institute, University College London, London, UK. [3]School of Biological and Behavioural Sciences, Queen Mary University of London, London, UK. [4]Centre for Epigenetics, Queen Mary University of London, London, UK. [5]Marine Evolution and Ecology, Naturalis Biodiversity Center, Leiden, The Netherlands. [6]Department of Freshwater and Marine Ecology, Institute for Biodiversity and Ecosystem Dynamics, University of Amsterdam, Amsterdam, The Netherlands. [7]Molecular Genetics Unit, Okinawa Institute of Science and Technology Graduate University, Okinawa, Japan. [8]Department of Molecular and Cell Biology, University of California, Berkeley, CA, USA. [9]Chan-Zuckerberg BioHub, San Francisco, CA, USA. [10]Marine Genomics Unit, Okinawa Institute of Science and Technology Graduate University, Okinawa, Japan. [11]Department of Biology, Faculty of Education, Mie University, Tsu, Japan. ✉e-mail: f.marletaz@ucl.ac.uk

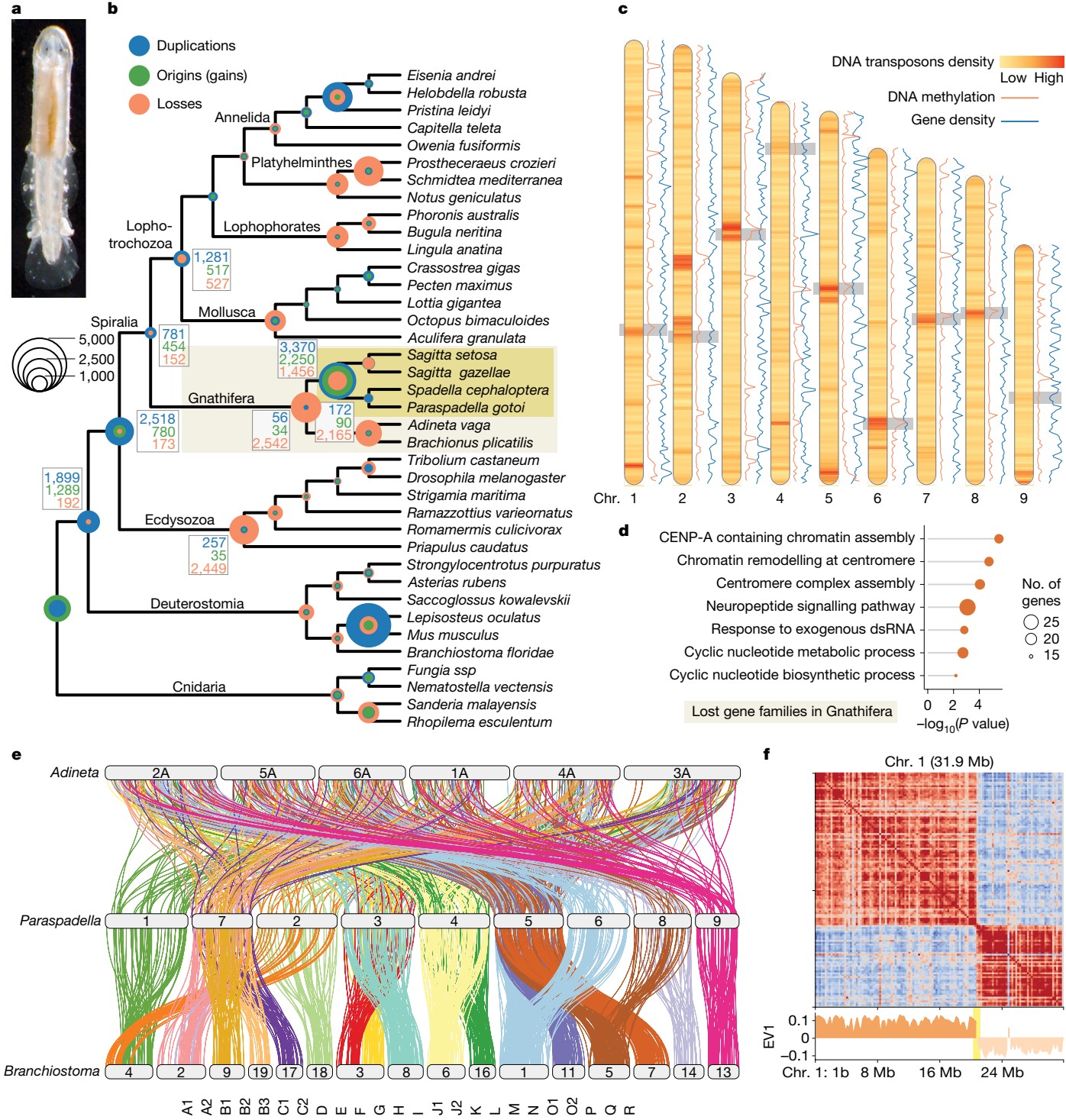

**Fig. 1 | Evolution of gene complement and genome architecture in gnathiferans and chaetognaths. a**, Adult *P. gotoi* (photograph credit: Taichiro Goto). **b**, Summary of gene family reconstruction in selected metazoan genomes (Supplementary Table 2) including inference of duplication events using gene tree–species tree reconciliation[63]. **c**, Chromosomes of *P. gotoi*, highlighting the binned density of DNA transposons elements, the level of DNA methylation and the gene density, and the position of centromeres inferred from Hi-C data (highlighted with a grey box). Chr., chromosome. **d**, Gene ontology terms (biological functions) enriched in gene families lost in the gnathiferan lineage (Fisher's exact test, two-sided). **e**, Evolution of gene linkage

in gnathiferans, highlighting the fusions that took place in chaetognath and the extensive chromosomal scrambling in the *Adineta* rotifer. Links between chromosomes showing significant reciprocal enrichment are displayed and coloured according to their bilaterian linkage group as defined by a comparison with amphioxus. Amphioxus chromosomes not showing an enrichment with *P. gotoi* are omitted. **f**, Pearson correlation matrix and corresponding eigenvector (EV1) derived from Hi-C data for chromosome 1. The shift in contacts associated with position of centromere is highlighted (for other chromosomes, see Extended Data Fig. 4).

of body plan evolution in this active predator with elaborate sensory organs and complex behaviours[28].

## Fast genome evolution in Gnathifera

We sequenced the genome of the chaetognath *P. gotoi* by using long and short reads from a five-generation inbred line and scaffolded the assembly to chromosome-scale level using proximity ligation data (Fig. 1c and Extended Data Fig. 1a). We obtained a relatively small assembly size of 257 Mb and found 9 major chromosome-size scaffolds, consistent with cytogenetic data in other species[2,29]. Repeats make up to 20% of the genome, with the major class being long terminal repeat (LTR) retrotransposons (8.6% of repeats), followed by long interspersed nuclear elements (LINEs) (4.6%) and DNA transposons (3.4%) and a large fraction of unclassified repeats (around 80%) (Extended Data Fig. 1b). Of note, several DNA transposons are enriched in localized putative centromeric regions (Fig. 1c). We annotated 22,072 protein-coding genes using long and short-read transcriptomic data (Supplementary Table 1).

To characterize the genomic evolution of gnathiferans, we compared our chaetognath genome and transcriptomes with those of other spiralians[7,25,30]. Gene family analysis uncovered extensive gene loss (2,542 ancestral gene families) in the branch leading to Gnathifera (Fig. 1b and Supplementary Tables 2 and 3). This observation is confirmed by the distribution of PANTHER families (Extended Data Fig. 2a) and maximum-likelihood estimation of gene loss (Extended Data Fig. 2b), and is accompanied with contraction of neural- and sensory-related gene families (Extended Data Fig. 2c,d). Among gnathiferans, both rotifers and to a lesser extent chaetognaths further experienced lineage-specific gene losses (2,165 and 1,456 families lost, respectively) and contractions of multiple gene families[31] (172 and 116 families, respectively; Extended Data Fig. 2c). Estimation of these gene family events is robust to alternative species tree[5,6] (Supplementary Fig. 1). Functional enrichment analyses surprisingly indicate that nearly all genes involved in CenH3 (also known as CENP-A) centromeric chromatin assembly have been lost in Gnathifera (12 out of 20, including the CenH3 and centromere protein T (CENP-T) genes; Fig. 1d and Supplementary Table 4), a condition that has been observed in insects exhibiting non-localized holocentromeres[32].

Absence of CenH3 centromeres has previously been associated with accelerated genomic rearrangements[33]. We thus investigated the retention in chaetognath and rotifers of bilaterian ancestral linkage groups (BLGs), which are conserved throughout bilaterian animal evolution, from chordates to annelids[7,8]. We found that most BLGs remain statistically detectable when comparing chaetognaths with amphioxus, sea star and sea scallop despite multiple fusions, whereas they appear completely scrambled in rotifers (Fig. 1e and Extended Data Fig. 3a–d). Most chaetognath chromosomes seem to derive from 2 to 4 fused BLGs with detected BLGs only represented once, and A2, B2, L, N or Q not detected. Among the four fusions-with-mixing ($\otimes$) previously reported in spiralians, namely H$\otimes$Q, J2$\otimes$L, K$\otimes$O2, and O1$\otimes$R[7,8,34], we identified the presence of K$\otimes$O2 and O1$\otimes$R in chaetognaths, indicating that they are likely to be shared by all spiralians (Fig. 1e and Extended Data Fig. 3). We could not detect Q and L, which could mean either that J2$\otimes$L and H$\otimes$Q fusions took place after the divergence of gnathiferans (particularly as we detected H and J2 alone) or that the signal is too limited. No chromosomal correspondence is detectable between chaetognaths and rotifers (Fig. 1e and Extended Data Fig. 3d), indicating a similar fast pace of chromosomal evolution in bdelloid and monogonont rotifers (Extended Data Fig. 3e,f). Together, these results suggest an accelerated rate of chromosomal rearrangement in the gnathiferan lineages.

The loss of CenH3 has been associated in other lineages with the presence of holocentromeres and correlated with an absence of chromatin compartmentalization at the level of chromosome arms in plants[33]. Here we conversely found a bimodal compartmentalization of the three-dimensional chromatin architecture for all *P. gotoi* chromosomes (Fig. 1f and Extended Data Fig. 4). The borders between these two interacting compartments are enriched in several mobile element categories observed in centromeric regions of other lineages, such as DNA transposons (for example, PIF/Harbinger and ISL2EU transposons; Fig. 1c) as well as LINE/SINE retrotransposons (Supplementary Table 5) and they show increased CpG methylation (Fig. 1c). These regions therefore exhibit classical features of centromeres indicating that chaetognaths possess localized centromeres despite lacking CenH3 and other molecular actors of centromere assembly[35] and corroborating prior microscopic observations[2,29]. Unlike chaetognaths, the rotifer *Adineta vaga* are likely to possesses holocentromeres[25,35,36], suggesting that centromeres have broadly diverged in the gnathiferan lineage and have been replaced in chaetognaths by neocentromeres making use of alternative molecular components.

At a smaller scale, chaetognath chromosomes appear to display very limited large-scale A/B compartmentalization (Extended Data Fig. 4a,e) and we did not observe an aggregated signal consistent with the presence of local topologically associating domains[37] (TADs) (Extended Data Fig. 4c,d).

## Ancestral and novel cell types

To better understand how the observed changes in genome architecture and gene complement have shaped the unusual body of chaetognaths, we generated a single-cell atlas of *P. gotoi* juveniles and adults comprising almost 30,000 cells classified into roughly 30 differentiated cell types (Fig. 2a,b and Extended Data Fig. 5a–d). We used a combination of in situ hybridization, literature surveys and cross-species comparisons to characterize these cell types and found neuronal, epidermal and muscle cell types, ciliary cells, stem cells, chaetal cells, gut cells and eye cells. To determine their evolutionary origin, we compared chaetognaths cell types to those of selected representatives of main animal lineages[38-40] (Extended Data Fig. 6). We found multiple cell types that are likely to be derived from the bilaterian ancestor (namely, neurons, muscles, gut, ciliary or germ cells), as supported by the shared expression of multiple orthologous genes (Fig. 2c). However, several cell types appear to be chaetognath-specific, such as their characteristic grasping spines, sensory papillae or ciliary sense organs.

Among ancestral cell types, we identified two neuronal clusters (clusters 11 and 26) that exhibit gene expression of the neuronal marker *Elav*, several synaptotagmins, calcium (for example *Cac1a-2*) and sodium (for example, *Scna-3*) channels and a dopamine-synthesizing enzyme (*Ddc*), and can be distinguished by distinct gene expression of glutamate receptors (*Grik2* and *Grm*) and degrading enzymes (*Glna*) (Fig. 2b). *Cac1a-2* is expressed in both the ventral and cephalic ganglia as well as in some peripheral neurons, an abundant cell type in the chaetognath nervous system[41] (Fig. 2d). These neuronal clusters map well to those of other protostomes (shared expression of the transcription factor genes *Bc11a*, *Cot1*, *Csde1* and *Meis* and the splicing factor gene *Rfox3*; Fig. 2c and Supplementary Table 9). Identified sensory cells comprise a photoreceptor cell-type in the eye (cluster 29) that expresses xenopsin 1 (*Opsd-1*) as described[42]. Notably, *P. gotoi* ciliary cell types express Lophotrochin, a gene that was previously shown to be expressed in ciliary bands of lophotrochozoans larvae and adults, including the corona of cilia of rotifers[39,43]. Two muscle cell types express titin and troponin as well as distinct actin paralogues, corroborating prior expression surveys of alpha actin (*Acts-1*), localized in longitudinal muscles (cluster 12), and Actin 1 (*Acti1-5*), in the muscles of the gnatha (cluster 19)[44]. Finally, two putative germline clusters (clusters 16 and 32; Fig. 2b) express stem cell markers, multiple copies of Vasa (*Ddx*) previously showed to be present in chaetognath germ cells[45], other germline markers such as *piwi*, *Maelstrom* and *Elav*, as well as several enzymes controlling DNA methylation (DNMT3.1, DNMT3.4 and TETb). More than a third of the total of cell-type marker genes are specific to these germline clusters (Extended Data Fig. 6), indicating a massive

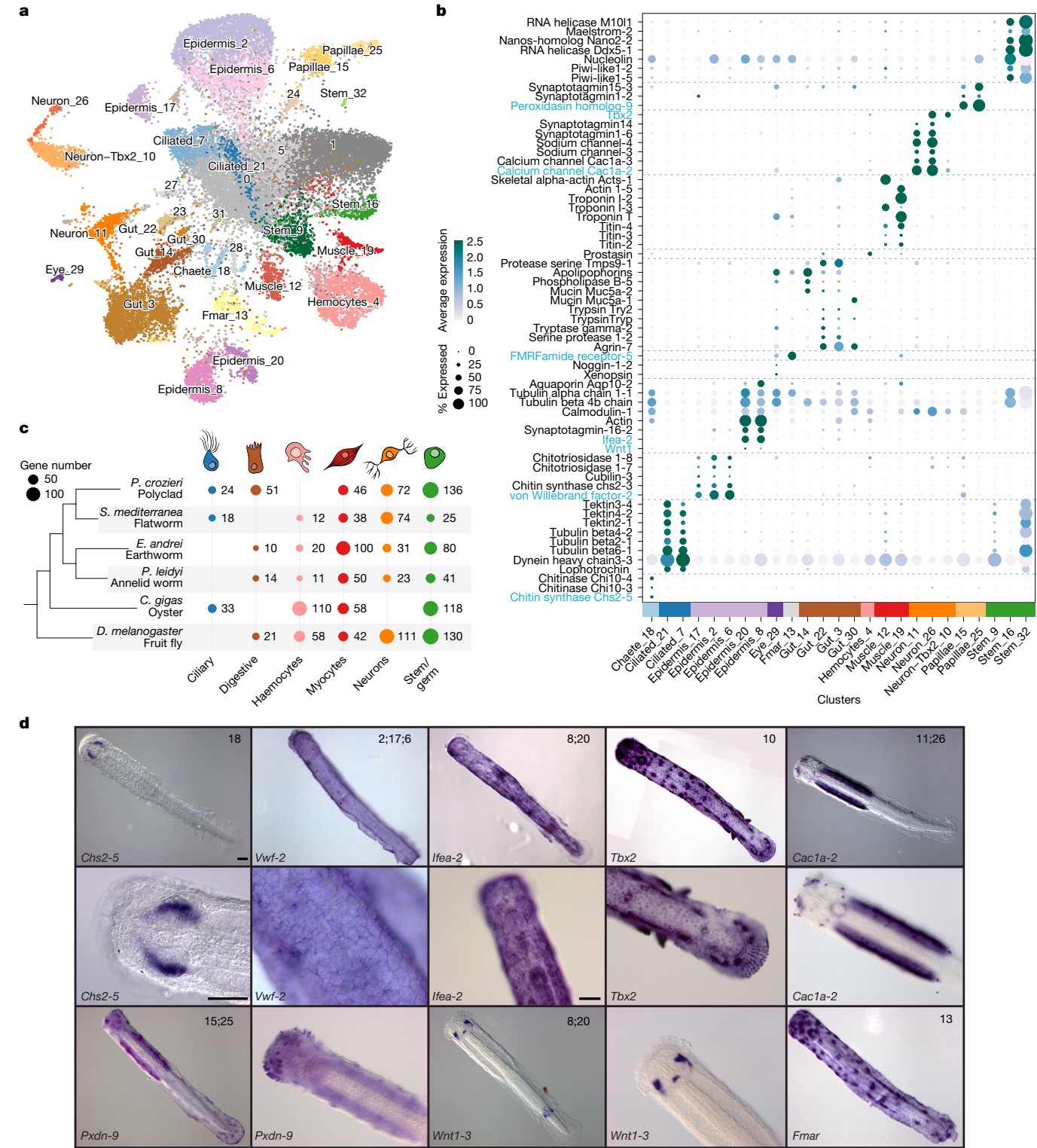

**Fig. 2 | Single-cell repertoire in the chaetognath *P. gotoi*. a**, Uniform manifold approximation and projection (UMAP) dimensionality reduction of cell expression profiles from pooled juvenile and adult libraries (Extended Data Fig. 5) with inferred clusters coloured and labelled. **b**, Dot plot of selected cell-type marker genes for each cluster, with genes selected as markers for in situ hybridization highlighted in blue, and dotted lines separating markers characteristics of broad cell-type categories. **c**, Summary of SAMap comparisons between chaetognath and selected bilaterians. Dot plot and text labels indicate the number of conserved cell-type genes between *P. gotoi* and selected species (full species name in Fig. 1b). **d**, Microphotographs of whole-mount in situ hybridization staining for selected markers in juvenile stages (3 days post-

fertilization (dpf)). All in situ hybridization experiments were carried out on at least three distinct individuals. The second row shows higher magnification of images in the first row. Numbers indicate clusters in **a** that are represented by the labelled cells. *Chs2-5* is indicative of the chaete cells located near the grasping spines. *Vwf-2* is expressed in the epidermal cells, in a subset of which *Wnt1* seems to be localized. *Tbx2* and *Fmar-5* are expressed in ciliary organs. *Cac1a-2* is expressed in the main ventral neural ganglion as well as other cephalic cells. *Pxdn-9* marks the papillae sensory cells in the head. A full list of cell-type markers is provided in Supplementary Table 6 and molecular probes are listed in Supplementary Table 7. Scale bars, 50 μm.

expression of maternal transcripts during oogenesis, as previously observed in other species[46]. Chaetognath stem cells co-express multiple chromatin modifiers (for example, *Atrx*, *Chd5* and *Dek*) or splicing factors (*Tra2b* and *Srrm1*) as well as the transcription factors *Yboxh-2* with other protostomes (Supplementary Table 9). Overall, we observed a small overlap in transcription factor co-expression among protostome cell types, which agrees with similar comparisons[39,40,47] and was further confirmed by a 1:1 comparison between the earthworm and a flatworm dataset (*Schmidtea mediterranea*; Supplementary Fig. 2).

Chaetognath-specific cell types—that is, those that appear unrelated to any other bilaterian cell types in our cross-species comparison—are associated with phylum-specific sensory and structural roles. Among these, we find two peripheral sensory papillae cell types (expressing neuronal genes such as synaptotagmins and acetylcholine receptors *Acha7-1*) (clusters 15 and 25) positioned around the mouth and interspersed in the epidermis throughout the body (see *Pxdn-9* (ref. 48) in Fig. 2d). We also recover two cell types (clusters 10 and 13) that are associated with ciliary sense organs, mechanoreceptors distributed on the body of the animal that constitute their primary sensory system for swimming and predation, as indicated by expression of *Tbx2* and *Fmar-5* in these structures[48] (Fig. 2b,d). The expression of chitin synthase (*Chs2-5*) near chaetognath grasping spines indicates that these are probably ensuring the growth and synthesis of the grasping spines (cluster 18), whereas the gene *Vwf-2* (clusters 2, 6 and 17) and *Ifea-2* (clusters 8 and 20) show expression in the epidermal layer (Fig. 2d). The presence of multiple epidermal cell types is consistent with the presence of a complex pluristratified epidermis in chaetognaths, a condition rarely found outside of vertebrates[49]. All these structural cell types express various chitin synthase and chitinases, with alternative paralogues present in the 'chaete' cells (clusters 18: *Chs2-5*, *Chi10-1*, *Chi10-3* and *Chi10-4*) and epidermal cells (clusters 2, 6 and 17: *Chs2-3*, *Chit1-7* and *Chit1-8*).

To further understand the origin and diversification of chaetognath cell types, we then investigated the evolution of the genes involved in their making.

## Gene turnover and cell-type innovation

Phylogenetic reconciliation indicated that chaetognaths share a large burst of gene duplication, but whether this event results from a whole-genome duplication or not remains unclear[9,10] (Fig. 1b). The magnitude of this event (3,379 families) is comparable in extent to other gene duplication events such as the vertebrate 2R (4,203 families) or the clitellate-specific gene expansion event[34] (Fig. 1b). Functional enrichment in *P. gotoi* duplicated genes highlighted terms related to ion transmembrane transport, as well as developmental genes (Fig. 3a), which is also corroborated with phylogenetic estimation of gene family expansion in chaetognaths (Extended Data Fig. 2c,d). This gene expansion is linked to cell-type novelty, as for instance, *P. gotoi* has eight copies of the piezo gene, a mechanosensory ion channel (with one paralogue specifically expressed in the Papillae cluster 25), which is present as a single copy in limpet or sea urchin[50], or up to eight copies of *Elav* (expressed in neurons and germline cells), which is present as a single copy in most non-vertebrates (Supplementary Table 3).

To determine whether gene duplications occurred at the same evolutionary time, we examined the distribution of transversions at four-fold degenerate sites (FDTv) in pairwise gene comparison. We observe a single FDTv peak for genes that belong to duplicated subfamilies showing distinct numbers of paralogues, indicating that they probably originated in a single event (Fig. 3c). Duplicated genes mostly occur in pairs, but higher copy numbers make up more than 50% of these duplicates (3 or more copies; Extended Data Fig. 7a). Next, we examined the location of duplicates across the genome. The chromosomal distribution of duplicates does not match the distribution of ancestral linkages across chromosomes as only one copy of bilaterian linkage

groups is detected in *P. gotoi* chromosomes (Fig. 3b). We also found that about 25% of duplicate pairs are located on the same chromosomes, and that no chromosome pair displays a mutual enrichment in duplicates, as would be expected in a whole-genome duplication event (Fig. 3d,e). The separation between duplicate pairs shows a bimodal distribution, with some pairs approximately 10 kb apart and some around 10 Mb apart (Extended Data Fig. 7b). Similarly, we did not detect fewer introns in members of duplicated gene families, ruling out a burst of retrotransposition (Extended Data Fig. 7c). Together, these observations are more consistent with a burst of tandem duplication than with a single whole-genome duplication event, which is a rather unique condition among animals. The unusual expansion of median and posterior Hox genes of chaetognaths is an example of such tandem expansion (Fig. 4a).

This reshuffling of the chaetognath gene repertoire is not limited to gene duplications. In total, up to 2,250 gene families appear to be specific to chaetognaths (that is, found in at least two of the four chaetognath species considered, but not outside chaetognaths) (Fig. 1a). This is quite a remarkable condition for a bilaterian phylum; by contrast, we detected only 157 and 124 lineage-specific gene families in molluscs or echinoderms, respectively[51] (Fig. 1a). In *P. gotoi*, these families include 1,832 genes, 8.3% of all predicted genes; to ascertain these numbers, we further confirmed that 1,307 of these genes do not have detectable homology outside chaetognaths in a direct phylostratigraphy assignment, and we observed Pfam domains in only 397 of them (Supplementary Table 1). These distinct lines of evidence support that this observation is not a reconstruction artefact such as those pointed out by recent studies[31]. These novel genes appear to have a regulated expression and putative functional role, as 1,471 of them are among cell-type markers in at least a single-cell cluster (Fig. 3f,g and Supplementary Table 1). We did not observe similar bursts of gene novelties in similarly fast-evolving lineages such as platyhelminths or ecdysozoans, which further rules out the possiblity of homology detection failure (Fig. 1a).

We found that both novel and duplicated gene populations have a key role in chaetognath cell types, particularly novel ones, as many of them are cell-type-specific (for example, 'chaete', Fmar, ciliary and epidermis; Fig. 3g,h). We estimated the phylostratigraphic age of gene sets specific to each cell type by leveraging gene phylogenetic age and noticed that ciliary cells appear among the ones with the most recent profile (Extended Data Fig. 8a,b). Notably, such an incorporation of phylum-specific genes among cell-type markers is observed only in chaetognaths when compared to five other investigated species in which most marker genes are either ancient or species-specific (Fig. 3f and Extended Data Fig. 8c). We therefore hypothesize that after extensive gene loss in the gnathiferan lineage, particularly that affecting gene families involved in sensory and nervous functions, the evolution of cell types in chaetognaths relied on newly evolved genes as well as lineage-specific tandemly duplicated genes. This reshuffling of the gene repertoire therefore probably had an instrumental role in the establishment of their distinctive body plan of chaetognaths.

## Expansion of the chaetognath Hox cluster

As a primary example of tandem duplication, we found that *P. gotoi* possesses an increased complement of 14 Hox genes[19], related to an expansion of both median (7 genes, 3 copies of the PG6/8 paralogy group) and posterior Hox genes (5 genes), an exceptional situation outside vertebrates (Fig. 4a and Supplementary Fig. 3). The Hox cluster of *P. gotoi* is unusually large, spanning 2.4 Mb (Extended Data Fig. 9a), with genes arranged in the expected 'colinear' order, and also includes 15 interspersed non-Hox genes (Extended Data Fig. 9a). Remarkably, the *MedPost* gene, originally described as an oddity[19] and later as a synapomorphy of Gnathifera[18], is consistently located in the cluster between median and bona fide posterior Hox genes, as originally predicted; its

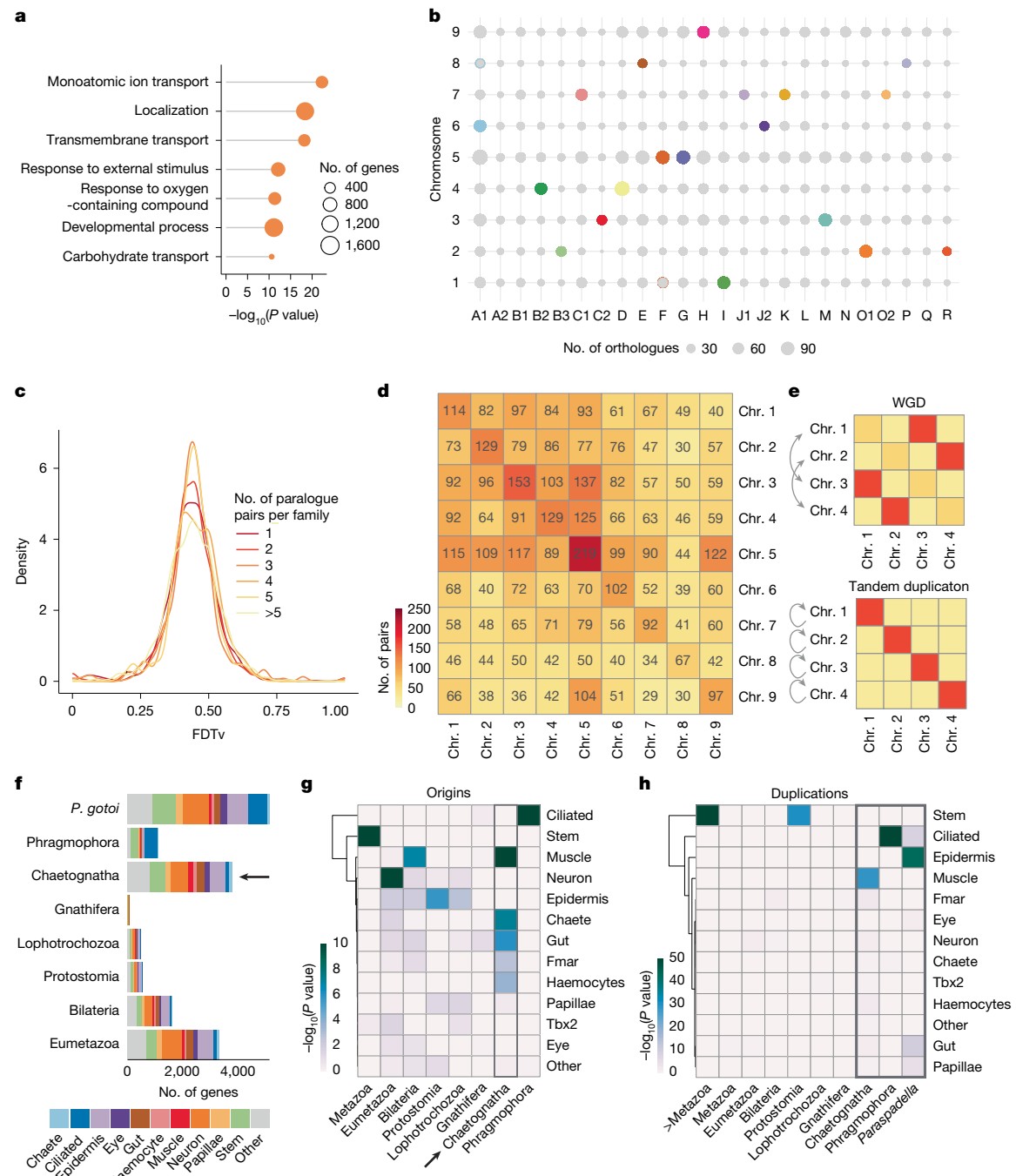

**Fig. 3 | Gene duplication and gene novelty in chaetognaths. a**, Overrepresented gene ontology terms in chaetognath duplicated genes (Fisher's exact test, two-sided). **b**, Distribution of ancestral BLGs in chaetognath chromosomes. Each bubble represents the number of shared orthologues between a pair of chromosomes, which is shown in colour (Fig. 1e) if the enrichment is significant (Fisher test) and is otherwise shown in grey. **c**, Distribution of synonymous transversions at fourfold degenerate sites (FDTvs) between pairs of paralogues in sets of duplicates showing distinct numbers of chaetognath-specific paralogues. **d**, Number of paralogue pairs shared across chaetognath

chromosomes indicate the absence of pairwise chromosome clustering. Each duplicated gene was only counted in paralogue pair to avoid giving excess weight to large paralogous families, thus the matrix is asymmetrical. **e**, Expected patterns of paralogue pairing in chromosomes in case of whole-genome duplication (WGD; top) or tandem duplication (bottom). **f**, Phylostratigraphic distribution of cell-type marker genes for *P. gotoi* (arrow). **g,h**, Hypergeometric enrichment of genes that originated (**g**) or duplicated (**h**) at successive phylostrata in broad cell types of chaetognaths. Box and arrow highlight recently evolved and duplicated genes, respectively.

location could not be corroborated in rotifers in which the Hox cluster is dispersed[25].

Hox genes are expressed in several cell types, mainly epidermal, neuronal and papillae cell types (Extended Data Fig. 9b). In situ hybridization on six Hox genes in *P. gotoi* juveniles (Fig. 4b,c and Supplementary Table 7) indicates that whereas anterior Hox genes show a broad

expression in the chaetognath nervous system (paired ventral and cephalic ganglions for *Hox1* and *Hox4*), median and posterior Hox genes show a more classical staggered expression in the ventral ganglion that appears coherent with the paralogy group (*MedN3* is more anterior than *MedN4*, and *PostN4* in the posterior-most part of the ganglion)[19] (Extended Data Fig. 9c). *PostN4* presents a dual domain, both in the

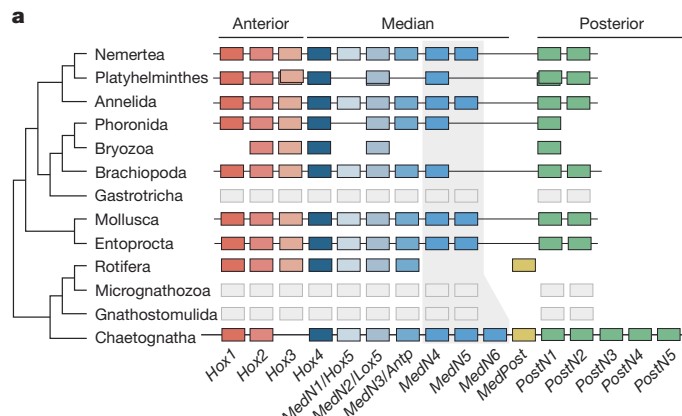

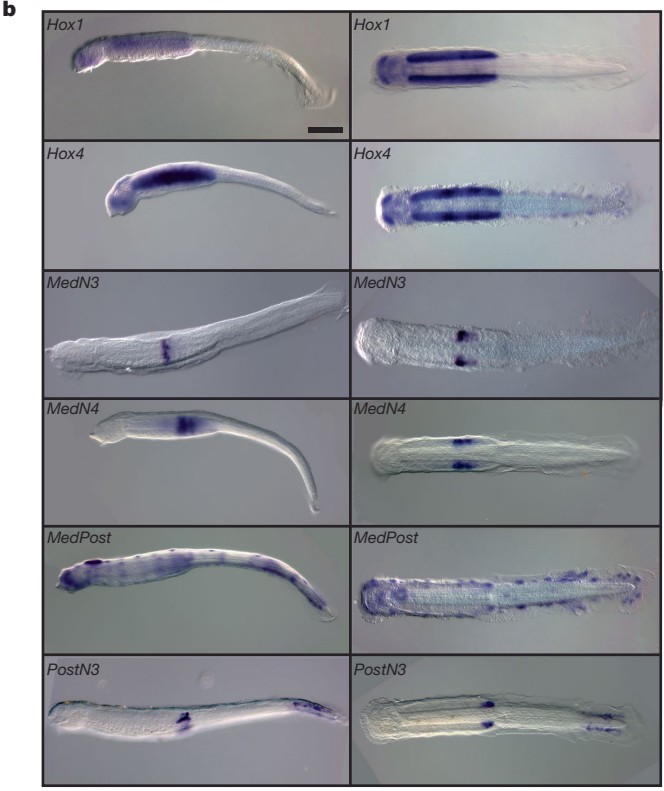

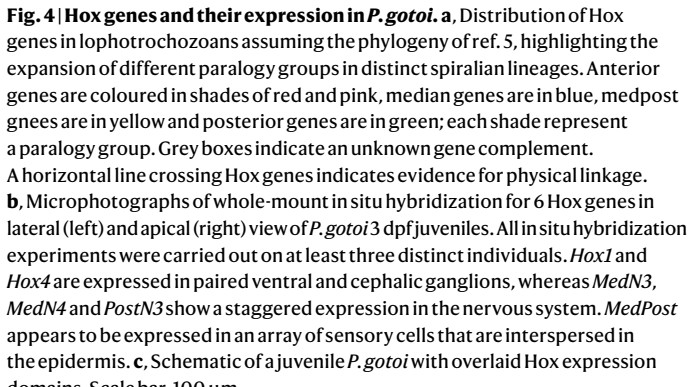

**Fig. 4 | Hox genes and their expression in *P. gotoi*. a**, Distribution of Hox genes in lophotrochozoans assuming the phylogeny of ref. 5, highlighting the expansion of different paralogy groups in distinct spiralian lineages. Anterior genes are coloured in shades of red and pink, median genes are in blue, medpost gnees are in yellow and posterior genes are in green; each shade represent a paralogy group. Grey boxes indicate an unknown gene complement. A horizontal line crossing Hox genes indicates evidence for physical linkage. **b**, Microphotographs of whole-mount in situ hybridization for 6 Hox genes in lateral (left) and apical (right) view of *P. gotoi* 3 dpf juveniles. All in situ hybridization experiments were carried out on at least three distinct individuals. *Hox1* and *Hox4* are expressed in paired ventral and cephalic ganglions, whereas *MedN3*, *MedN4* and *PostN3* show a staggered expression in the nervous system. *MedPost* appears to be expressed in an array of sensory cells that are interspersed in the epidermis. **c**, Schematic of a juvenile *P. gotoi* with overlaid Hox expression domains. Scale bar, 100 μm.

posterior-most part of the ventral ganglion and in the postanal region of the juvenile (Fig. 4b). Surprisingly, *MedPost* is localized in papillae peripheral sensory neurons interspersed in the epidermis (Fig. 4b and Extended Data Fig. 9b). The expression of *MedPost* in sensory neurons throughout the body is different from that observed in rotifers, where it is expressed specifically in the postanal components of the nervous system (Extended Data Fig. 9c). In chaetognaths, posterior Hox genes are expressed in this postanal region; these genes have been lost in rotifers[18]. The cell-type-specific expression of *MedPost* suggests that acquisition of new expression domains could have participated in the emergence of novelties at the body plan or cell-type level.

## Retargeting of DNA methylation

We set out to understand whether the reshuffling of chaetognath gene complement had an effect on their gene regulation[52]. We found that *P. gotoi* represents an outlier in terms of DNA methylation among animals. Whereas most non-vertebrate lineages have highest levels of DNA methylation on gene bodies, usually associated with robustly expressed genes[53], the vast majority of *P. gotoi* genes are not methylated (Fig. 5a and Supplementary Table 10), and the few that are contain protein domains that are typical of retrotransposons and have depleted chromatin accessibility, suggesting that they are silent genes with transposon origins (Extended Data Fig. 10a,b). Moreovoer, whereas transposable elements in most invertebrates are rarely directly methylated, *P. gotoi* shows a clear targeting of DNA methylation on transposons—most acutely on LTR retrotransposons, but also on other classes—but none on low-complexity repeats that are less likely to be parasitic (Fig. 5a and Extended Data Fig. 10c). This pattern of exclusive transposable element methylation and absence of gene body methylation represents a pattern reminiscent of fungi and other unicellular eukaryotes, and has been observed in only few animal lineages, including early diverging nematodes and ctenophores[54,55].

To explore how *P. gotoi* has modified its methylome compared to other animals, we examined the genetic toolkit responsible for DNA methylation in this species. We found that *DNMT1* and its partner *UHRF1* underwent gene duplications, with their paralogues undergoing probable sub-functionalization through the loss of ancestral protein domains such as zinc-finger CXXC in DNMT1 or the ubiquitin domain in UHRF1 (Fig. 5b and Extended Data Fig. 10d). Similarly, none of the three copies of *TET* (including a retrocopy) encode the ancestral zinc-finger CXXC domain combination that is probably shared by most spiralians[56]. Finally, six copies of *DNMT3* are present in chaetognaths, which all lost the ADD and PWWP domains; the latter drives DNA methylation to gene bodies by binding to H3K36me3 in other animals, a histone post translational modification that is typical of gene bodies[57]. Most DNA methylation genes are expressed in germ line cell types similarly to what has been observed in mice[58] (Extended Data Fig. 10e). In sum, *P. gotoi* represents an independent example of DNA methylation retargeting from gene bodies to transposable elements, associated with protein domain simplification of the DNA methylation toolkit.

## *Trans*-splicing and operons

Many genes in chaetognaths have been reported to undergo *trans*-splicing, a post-transcriptional spliceosomal modification that is associated with operonic transcription and resolves polycistronic transcripts[10,59]. At the genome scale, we found that *trans*-splicing is prevalent, occurring in nearly half the genes (10,197 out of 21,072 predicted genes), and 18% of all genes are organized in operons resolved by *trans*-splicing (Fig. 5c). *Trans*-splicing involves spliceosomal addition of splice leaders to messenger RNAs, with the splice leader sequence derived from genes often located in the ribosomal DNA clusters far from their target genes[60]. We identified three distinct splice leaders in *P. gotoi*, of which one (SLs1) is shared with *Spadella cephaloptera*[10].

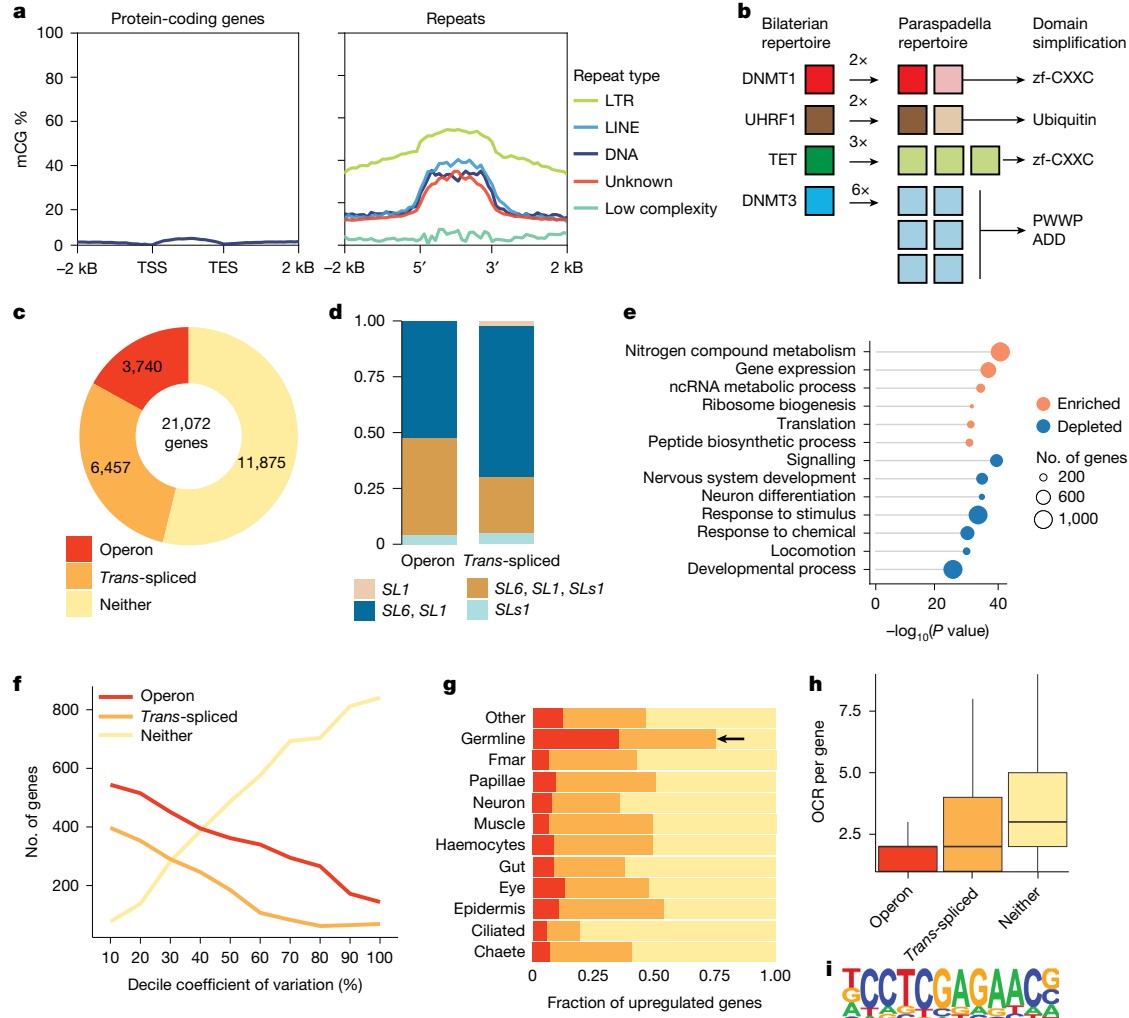

**Fig. 5 | Gene regulation, DNA methylation and *trans*-splicing. a**, DNA methylation in *P. gotoi* has reverted to exclusive transposon targeting. The metaplots show average methylation levels at CpG dinucleotides (mCG %) at protein-coding genes and repeats, classified for major transposable element type. Only repeats larger than 500 bp were used. TES, transcriptional end site. **b**, Diagram showing the expansion and simplification of the ancestral bilaterian DNA methylation toolkit in *P. gotoi*. Paralogues with simplified domain architectures are coloured in a lighter shade than those that retained the ancestral state. Full domain architectures of each gene are shown in Extended Data Fig. 10d. zf, zinc finger. **c**, Number of genes receiving *trans*-splicing and classified as belonging to an operon in *P. gotoi*. **d**, Occurrence of different annotated splice leaders and their combination in *trans*-splicing and operonic genes. **e**, Gene ontology enrichment (orange) and depletion (blue) for genes belonging to operons (Fisher's exact test, two-sided). ncRNA, noncoding RNA. **f**, *Trans*-spliced and operonic genes exhibit a lower expression specificity as measured by coefficient of variation calculated using cell-type averaged expression. **g**, Fraction of operonic, *trans*-spliced and other genes among marker genes in single-cell clusters with enrichment in germline marked by an arrow. **h**, Number of open chromatin regions associated with the different gene categories (box plot shows median, quartiles and extrema, *n* = 2,643, 5,586 and 8,851 genes for operons, *trans*-spliced genes and other genes, respectively), indicating the higher number of putative regulatory elements in genes that are not *trans*-spliced. **i**, Example of an enriched motif in open chromatin regions related to operons.

*Trans*-splicing and operonic transcription have a patchy distributions across animals, which suggests independent evolution in distinct lineages (nematodes, tunicates and rotifers)[10,60]. However, despite its proposed role in recovery from growth arrest[59] or germline maintenance[61], the evolution role of *trans*-splicing remains debated. Here, multiple splice leaders are found among the various transcripts of a given gene, and contrary to nematodes[60], we did not identify operon-specific splice leaders (Fig. 5d).

Functionally, operonic genes are preferentially related to nitrogen and protein metabolism and they are negatively associated with development, nervous system and signalling, suggesting that finely regulated genes are excluded from this process (Fig. 5e). For instance, the longest operon gathers nine genes, including genes encoding ribosomal proteins, ubiquitin and proteasome subunits (Supplementary Table 1). *Trans*-spliced genes have a higher expression level (Extended

Data Fig. 7f) and have, in general, a lower expression specificity at the cell-type level than those not subjected to splice leader addition (Fig. 5f); however, operonic and *trans*-spliced genes make up most of cell-type marker genes in the germline clusters (Fig. 5g). *Trans*-spliced genes are also not restricted to a phylogenetic origin and belong to all phylostrata (Extended Data Fig. 7g). Consistently, *trans*-spliced genes have a smaller number of putative regulatory elements (for example, as indicated by ATAC–seq peaks), in agreement with their lower regulatory and expression complexity (Fig. 5h). The presence of a large fraction of *trans*-spliced genes could have contributed to the overall compact regulatory landscape of *P. gotoi* compared to other lineages, with open chromatin regions located closer to the transcription start site (TSS)[7,21] (Extended Data Fig. 7h and Supplementary Table 11). However, we identified conserved distal regulatory regions and transcription factor binding properties with that of other animals (Extended

Data Fig. 7h–k). Interestingly, this also suggests that non operonic *trans*-spliced genes share these functional properties with operon genes rather than non-*trans*-spliced ones. We also detected enriched motifs associated with *trans*-spliced genes (for example, Fig. 5i), which do not correspond to transcription factors characterized in other lineages, corroborating the idea of independent origin of *trans*-splicing and operonic transcription in metazoans[10,60].

## Conclusion

Our results show that chaetognaths stand out from other animal lineages at the molecular and genomic levels, and provide clues to explain why their unique body plan appears so distinctive from other animal phyla[2]. Chaetognaths have undergone an extensive reshuffling of their gene complement as well as major fusions of ancestral chromosomal linkages (Fig. 1). This extensive duplication of genes does not appear to be related to a single whole-genome duplication event, but rather a burst of tandem duplications (Fig. 3). In contrast to the microscopic rotifers and other gnathiferans, large-bodied chaetognaths have an active predatory lifestyle with an array of sensory receptors, such as ciliary organs reminiscent of the lateral line[4], that are associated with innovation and diversification at the cell-type level (papillar, Fmar, ciliated and epidermal).

The gnathiferan lineage underwent accelerated genomic evolution, either as a single event or as multiple lineage-specific events that could plausibly have been coupled with morphological and body size reduction from their ancestors. In this view, chaetognaths would have secondarily evolved complex organ systems by mobilizing lineage-specific duplicated and novel genes instead of mainly relying on genes from the ancestral bilaterian toolkit, which had been decimated in this lineage, unlike what is observed in other animals (Fig. 3g,h and Extended Data Fig. 7). Of note, rotifers also acquired new genes, albeit through the distinct process of horizontal gene transfer, that were involved in the evolution of novel traits such as resistance to desiccation or ionizing radiation[62]. The paucity of Cambrian fossils for other gnathiferans such as rotifers or gnathostomulids, and their relatively large body size (for example, *Amiskwia*[12]) does not provide palaeontological support for a small-bodied gnathiferan ancestor. Further characterization of genomes and gene expression in other gnathiferan lineages (such as micrognathozoans or gnathostomulids) would help to further resolve organismal and genomic evolution in the gnathiferan lineage.

Chaetognaths also shed some light on the relationship between gene regulation and gene and genome evolution. The reshuffling of the gene repertoire in chaetognaths coincides with the evolution of novel strategies of gene regulation, such as the acquisition of *trans*-splicing[10]. Consistent with a previously proposed model[61], *trans*-spliced genes are particularly enriched in the germinal cells and show limited regulation. *Trans*-splicing may have a role in stabilizing housekeeping gene expression when gene body methylation is lost, suggesting a recurrent evolutionary process. Of note, *trans*-splicing is present in ctenophores, chaetognaths and nematodes[53,54], the three animal phyla in which gene body methylation has been repurposed for transposable element targeting. However, other evolutionary forces may have a role in the loss of gene body methylation, such as the DNA repair burden imposed by DNA methyltransferases[54]. Notably, lineages with *trans*-splicing, such as nematodes, tunicates, rotifers, platyhelminths and potentially ctenophores, are also prone to loss of ancestral linkage[8,60]. The absence of detectable sequence-insulated chromatin domains in chaetognaths, which have recently been shown to be a bilaterian innovation[55], would require further confirmation (for example, using micro-C[55]), but could suggest that such structures are less widespread than anticipated.

Our analysis of the genome, cell types and gene regulation provides clues to the origin of one of the most enigmatic bilaterian lineages, and a molecular explanation for its derived state and distinctive characteristics.

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

## Methods

### DNA isolation and sequencing

An inbred line was generated for five generations by crossing the progeny of an adult *P. gotoi* collected in the tidal zone near Amakusa (Kyushu prefecture, Japan). To this aim, juvenile chaetognaths were fed with increasingly older nauplii of the copepod *Tigriopus japonicus* to match their size until capable of feeding on adults according to previously established culture protocol[64]. DNA was extracted from about 20 adults using phenol-chloroform[65]. Illumina paired-end (25.9 Gb, about 100×), mate-pair (4 libraries: 3, 5, 7 and 10 kb insert) and Pacbio long-reads (5 Gb, about 20×) were sequenced from libraries prepared with this material. A proximity ligation library was constructed from about 20 adults collected at the same locality near Amakusa. In brief, animals were crosslinked in 1% paraformaledehyde, and chromatin was subsequently extracted, immobilized on solid-phase reversible immobilization beads, washed and digested with DpnII as described[66]. After end-labelling, proximity ligation was carried out using T4 DNA ligase and cross-linking reversed using Proteinase K, removed from the beads and the DNA fragments were purified again on SPRI beads. Sequencing library was constructed using the NEB Ultra library preparation kit (New England Biolabs) and sequenced on a NovaSeq6000 instrument.

### Genome assembly

The genome was assembled using meraculous (v.2.2.2.5) with a $k$-mer size of 41 and diploid mode set to '1' to enable haplotypes merging[67]. Primary contigs were screened for contamination using Blobtools (v.1.0.1) and three contaminant contigs were removed at this stage. Pacbio long-reads were then incorporated in the assembly using PBJelly with blasr alignment performed with the parameters '-minMatch 8 -sdpTupleSize 8 -minPctIdentity 75 -bestn 1 -nCandidates 10 -maxScore -500'. The resulting assembly was scaffolded using HiRise with the Hi-C data[68] as previously described[69]. The assembly was evaluated with BUSCO (v.5.4.1) with a final score of C:94.3% [S:91.3%, D:3.0%], F:1.3%, M:4.4% (n:954, metazoa_odb10). Examination of coverage and $k$-mer distribution using Merqury (v.1.3)[70] and BUSCO statistics[71] indicate that a minimal fraction of residual haplotypes has been retained (Extended Data Fig. 1). This is possibly due to the loss of polymorphisms during the inbreeding procedure.

### Annotation

Illumina RNA-seq and Pacbio Isoseq reads were generated using RNA extracted from pooled adult individuals. RNA-seq was aligned to the genome using STAR (v.2.5.2b), assembled using stringtie (v.1.3.3b) and also assembled as de novo transcripts using Trinity (v.2.5.1)[72]. For Isoseq reads, full length transcripts were obtained after circular consensus, detection of SMART adaptors, and polishing, as recommended by the manufacturer (https://github.com/PacificBiosciences/IsoSeq). These transcripts, as well as the trinity contigs, were aligned to the genome using GMAP (v.2017-09-05) and processed according to the Mikado pipeline (v.1.2.1) that leverages distinct transcriptomes, detects putative gene fusions by similarity search against Swissprot, enforces adequate splice-junctions from a curated set inferred using Portcullis (v.1.0.2) and inferred coding sequence using Trans-decoder[73]. High-quality transcripts derived from Mikado were used as a training set to generate a species-specific profile in the Augustus de novo prediction tool[74]. Intron and exon positions were subsequently used as hints to perform a gene prediction with Augustus, together with proteins of *Lottia gigantea* aligned using Exonerate (v.2.2.0). Gene models were further refined using PASA adding untranslated region (UTR) regions and isoforms by relying on the Mikado transcriptomes. We obtained a total of 22,082 genes (Supplementary Table 1). Repeat families were reconstructed using RepeatModeller (2.0.1) and subsequently used to annotate repetitive regions and compute repeat divergence in the genome with RepeatMasker (4.1.0) (Extended Data Fig. 1b). Rideogram was used for karyotype representation of gene and transposable elements contents

(Fig. 1c). Functional annotation was performed using Eggnog-mapper (v.2.1.5)[75] and Gene Ontology (GO) enrichments were performed using the goatools Python package[76]. For GO enrichment associated with family losses, we assigned non-redundant GO terms for each family from the emapper annotation obtained for selected species specified in Supplementary Table 2 and computed enrichment at the family level.

### Gene family and synteny analyses

Gene families were first reconstructed using Broccoli (v.1.1)[77] using the proteomes from the species listed in Supplementary Table 2. We constructed sequence alignments for all families including more than 6 genes, more than 3 species and fewer than 400 sequences in total using MAFFT (v.7.471)[78] filtered with CLIPKIT (v.1.1.6, -m gappy)[79] and an initial tree reconstructed with IQ-TREE (v.2.1.1) assuming the LG + R model[80]. These alignments and trees were then used as input for GeneRax (v.2.1.3)[63] with duplication and loss model (UndatedDL) and a per family rate. Reconciled trees in XML format were subsequently parsed to infer gene loss and gene duplication events for each node of the species tree (Fig. 1a and Extended Data Fig. 2b). Gene gains and losses were inferred following parsimony principles and using the ete3 Python library to process phylogenetic trees. For relevant lost families, we assessed whether the best blastp match in relevant species belonged to another gene family to disclose potential gene fragmentation related to evolutionary acceleration. Gene absence was also corroborated by tblastn against the genome of *P. gotoi*. For complementary phylostratigraphic analyses, we used GenEra (v.1.0.2)[81] with default options using the NCBI NR database. To compute FDTvs, we performed a self-alignment of the chaetognath proteome using a MMSeqs2 (r12-113e3)[82], and then reverse translated the alignments and computed the corresponding substitutions using custom code. For the CAFé analysis, a dated tree was obtained by extracted phylogenetic markers described in ref. 5 for the selected proteomes included in the analyses. An ultrametric dated tree was reconstructed using Phylobayes (v.4.1e) assuming a CAT + GTR model, a CIR relaxed clock model, birth-death priors on divergence times (0.5) and the following calibrations[83–85] (lower–upper bounds in million years): euarthropods (531–514), ecdysozoans (636–519), Gnathifera (636–518.5), bilaterians (636–521) and deuterostomes (635–516). CAFE (v.5.1) was then applied using gene families previously defined from Broccoli assuming a gamma model with one rate category ($k = 1$) and no error model[86].

For synteny comparisons, orthologous pairs were inferred by mutual best hit using MMSeqs2 (r12-113e3)[82]. Joint orthologue genome coordinates in chromosomal location were used for 'Oxford grid' plots (Fig. 1f and Extended Data Fig. 3) and Fisher's exact test was used to determine mutual enrichment of orthologues between chromosomes. Ribbon plots connecting orthologues in multiple species (Fig. 1d) were generated using Rideogram (v.0.2.2).

Hox genes were annotated based on similarity search toward previously annotated chaetognath Hox genes[19,87], inspection of diagnostic residues and phylogenetic analysis of selected bilaterian homeodomain genes (Supplementary Fig. 3) using IQ-TREE with a LG4X model and 1000 ultra-fast bootstraps.

### Hi-C processing

Hi-C reads were mapped using bwa mem (v.0.7.17) with parameters -5SP -TO. Ligation events in the aligned sequences were then identified with pairtools (v.1.0.2)[88]. In brief, we first used pairtools parse with parameters --min-mapq 30 --max-inter-align-gap 30 --no-flip --add-columns mapq --walks-policy 5unique, sorted resulting contact pairs with pairtools sort and removed duplicates with pairtools dedup to obtain the final filtered contact pairs file. Using juicer_tools (v.1.22.01)[89], filtered contact pairs were exported to a multi-resolution Hi-C matrix (2.5 Mb, 1 Mb, 500 kb, 250 kb, 100 kb, 50 kb, 25 kb, 15 kb, 10 kb, 5 kb) and normalized with the Knight-Ruiz (KR) procedure[90]. Contact map resolution was estimated using the approach proposed

in ref. 66. In brief, for each bin size stored in the multi-resolution Hi-C matrix, we count the number of bins with >1,000 contacts. The map resolution is given as the smallest bin size where >80% of bins have >1,000 contacts (5 kb here). All downstream analyses were performed using the fanc Python package (v.0.9.23)[91].

Putative centromere positions were inferred from Hi-C (Extended Data Fig. 4f): for each chromosome, we: (1) computed Pearson contact correlation matrices; (2) visually selected for each chromosome the eigenvector that best captured centromere information; and (3) identified centromere as the most insulating region (highest eigenvector value differential). We tested for the enrichment of specific repeats at inferred centromere location using a permutation procedure. Specifically, for each repeat family or subfamily 'r' overlapping at least 1 predicted centromere, and for each chromosome 'c', we: (1) generated $n = 1,000$ randomizations of 'r' over 'c' and (2) computed empirical permutation-based $P$ values based on the random background overlap of repeats at centromere.

To assess the organization of the *P. gotoi* genome into TADs, we computed insulation scores on the Hi-C matrix at 25 kb resolution using a window size of 250 kb. Putative TAD boundaries were called from insulation scores and TADs with boundary strength >0.5 (that is, discarding the first peak of weak boundary scores) were retained for the aggregated TAD and Hox cluster plots. Finally, organization into A/B compartments was assessed by: (1) selecting, for each chromosome, eigenvectors that best separate active and inactive regions on the basis of ATAC peak number; (2) orienting eigenvectors with ATAC peaks so that entries >0 correspond to active regions and <0 inactive; (3) representing interactions strength across active and inactive regions as a saddle plot (Extended Data Fig. 4e) using 50 eigenvector value categories.

## ATAC−seq
ATAC−seq was performed as previously described[92] following the Omni-ATAC protocol including digitonin (Promega)[93]. Tn5 tagmentase (Illumina) was applied to 50k nucleus and library subsequently constructed by PCR. ATAC−seq reads were aligned to the genome using bowtie2 (v.2.4.2)[94] with the parameters –very-sensitive and -k 10. Open chromatin regions (OCR or peaks) were called using Generich (v.0.6) (https://github.com/jsh58/Genrich) using ATAC model (-j), keeping unpaired alignments (-y), removing PCR duplicates (-r) and excluding reads mapped to mitochondria (-e MT). Unified OCRs were computed using pyBEDtools (github) as well as their relative location toward genes and 42,810 peaks in total were identified (Supplementary Table 11) and compared with their distribution in other species[7,21,92]. Mapping statistics and fraction of reads in peaks, calculated using bedtools (v.2.30.0), are provided in Supplementary Table 12. Motif enrichment was performed using HOMER (v.5) for both known and de novo motifs and plant derived motifs were excluded from results relying on known motifs[95].

## *Trans*-splicing and operons
Low-input RNA-seq libraries were generated from gastrula, hatchling and 3 days post-hatchling juveniles using low-input protocol (NEBNext Low Input RNA). We used SLIDR to perform a de novo annotation of splice leaders from RNA-seq data using default parameters[96] and compared the identified list with previously identified splice leaders in other chaetognath species[10]. We found that *P. gotoi* uses three major splice leaders of which two are shared with other chaetognath species. We performed a BLAST similarity search on Isoseq full length transcripts to annotate *trans*-spliced genes. To quantify *trans*-splicing in RNA-seq data, we aligned each set of RNA-seq reads to the genome using STAR (v.2.7.8a), detected splice leaders in reads using BBDuk from BBTools to with k = 25 and hdist=3 and set them aside as separate BAM (https://jgi.doe.gov/data-and-tools/software-tools/bbtools/bb-tools-user-guide/). We then quantified reads for our gene models using featureCount[97]. We considered a gene *trans*-spliced if at least five reads including a splice leader aligned to it. We annotated operons in the genome as sets of two

or more genes separated by less than 250 bp, which yielded 2,151 candidate operons containing 5,061 genes. Of those, 1,470 validated operons include at least 1 gene with annotated splice leader involving 3,565 genes.

## Single-cell transcriptomics
Adults and 3 dpf juveniles of *P. gotoi* were dissociated in filtered natural sea water in presence of 1% Pronase with gentle trituration, cells were washed, debris removed using 40-μm and 70-μm cell strainers (Flowmi) and resuspended in 50% Leibowitz L15 medium. Single-cell suspension quality and cell concentration was evaluated using C-Chip Disposable haemocytometer (NanoEntek) and viability was assessed using Trypan blue (Thermo Fisher). Around 15,000 cells were loaded into a 10x chip (10x Genomics, v.2) and 3 libraries (2 for adults and 1 for 3 dpf juveniles) were prepared and sequenced on a NovaSeq6000 and NovaSeq X instrument (Illumina). Reads were mapped onto the *P. gotoi* genome using Cell Ranger (v.7.0.1) after extending the 3′ UTR of the genome annotation by 900 bp with the script utr_extension.py. Library saturation is 59.8% for adult 1, 74.2% for adult 2, and 89.5% for 3 dpf juveniles, and read depth is 24,583 mean reads per cell for adult 1, 37,173 for adult 2 and 58,191 for juveniles. Counts were analysed using Seurat (v.5)[98] and all cells with fewer than 500 or more than 20,000 unique molecular identifiers, fewer than 200 genes and mitochondrial content higher than 20% were filtered out. Our final dataset contains 29,513 cells (21,777 adult cells and 7,736 juvenile cells) expressing 19,367 genes. These datasets were merged, normalized, scaled and subjected to principal components analysis (using the top 50 principal components) followed by canonical correlation analysis integration were performed on the object layers. Our final resolution for clustering was set to 1, resulting in a total of 33 cell clusters (Rscript in Supplementary Information). Gene expression specificity such as the tau metrics or the coefficient of variation were computed as described[99].

SAMap (v.1.0.2) was applied between the *P. gotoi* datasets and the following species: the polyclad flatworm *Prostheceraeus crozieri*[39], the flatworm *S. mediterranea*[100], the oyster *Crassostrea gigas* (larval stages)[39], the earthworm *Eisenia andrei*[40], the annelid worm *Pristina leidyi*[38] and the fruit fly *Drosophila melanogaster*[40]. Cell-type annotation for all species can be found in Supplementary Table 8; for a list of genes shared between dataset per cell type, see Supplementary Table 9.

## In situ hybridization
Synthetic probes were designed for selected genes (Supplementary Table 7) flanked by T7 or SP6 adaptor (IDT). Riboprobes were then synthesized using the DIG labelling kit (Roche). In situ hybridization was performed as described[7,101,102] with a shortened Proteinase K digestion (2–3 min). Stained embryos and juveniles were imaged with a Zeiss AxioImager M1 using bright field Nomarski optics. Unfortunately, surveying the expression of other Hox genes than the one we investigate here in *P. gotoi* proved challenging as the high AT percentage and low-complexity stretches in their transcripts made it impossible to synthesize or clone a probe.

## Enzymatic Methyl-seq analysis
Using the same genomic DNA obtained for genome sequencing, 100 ng of *P. gotoi* gDNA were spiked in with unmethylated lambda phage and CpG methylated pUC19 control before fragmenting it to about 300 bp with a BioRuptor (Covaris). The resulting sheared DNA was then used to prepare an Enzymatic Methyl-seq library according to the manufacturer's instructions (NEBNext Enzymatic Methyl-seq Kit). The library was sequenced with a NovaSeq instrument using 150 paired-end reads to an average coverage of ~14× per CpG site. Reads were mapped to the reference genome, including also a phage lambda and pUC19 sequences, using BS-Seeker2 (v.2.1.8) with bowtie2 as back-end. Duplicate reads were removed with Sambamba (v.1.0.1), and methylation calls were obtained using CGmapTools (v.0.1.3). The lambda genome showed a non-conversion rate of 0.11%, and the pUC19 methylated control showed

97.88% global methylation at CpG methylation sites, as expected for this technology and validating the quality of the library. The rest of the data were read into R using the bsseq Bioconductor package, which was used to obtain methylation average per gene and genomic windows used for chromosome-level methylation. Bigwig files were constructed from the CGmapTools output, and were used as input for DeepTools2 to detect methylation levels on top of protein-coding genes and the repeat annotation (Extended Data Fig. 10). Methylation genes were obtained using human DNMT1, DNMT3A, TET1 and UHRF1 sequences as input for a BLASTP search against the predicted proteome of *P. gotoi*, and were subsequently analysed in the InterProScan web server (https://www.ebi.ac.uk/interpro). The top hits were manually inspected and contrasted with the phylogenetic trees obtained from the GeneRax search.

### Reporting summary

Further information on research design is available in the Nature Portfolio Reporting Summary linked to this article.

### Data availability

The genome assembly and associated data have been deposited to the NCBI database under BioProject PRJNA1169515. Functional genomics data have been deposited in the Gene Expression Omnibus (GEO) GSE279488. Processed data files including genome annotation have also been deposited at Zenodo (https://doi.org/10.5281/zenodo.13936459 (ref. 103)).

### Code availability

Code used is available at Zenodo (https://doi.org/10.5281/zenodo.13936459 (ref. 103)).

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

**Acknowledgements** F.M. and L.P. are supported by a Royal Society University Research Fellowship (URF\R1\191161), a Biotechnology and Biological Sciences Research Council research grant (BB/V01109X/1), a Leverhulme Trust research grant (RPG-2021-436) and a Japan Society for the Promotion of Science Kakenhi grant (JP 19K06620). E.P. is supported by a Newton International Fellowship from the Royal Society (NIF\R1\222125). Work at the Okinawa Institute of Science and Technology (OIST) Molecular Genetics Unit (D.S.R., D.G. and F.M.) and Marine Genomics Unit (N.S.) was supported by OIST internal funds. We acknowledge the OIST sequencing facility and its members for support, especially N. Arakaki and H. Goto. D.S.R. is a Chan Zuckerberg Biohub Investigator and is supported by the Marthella Foskett Brown Family Chair of Biological Sciences at the University of California, Berkeley. We thank the OIST Sequencing Section and acknowledge the support of OIST supercomputing. A.d.M. and J.M.M.-D. are funded by the Horizon 2020 Framework Programme (European Research Council Starting Grants action number 950230 to A.d.M. and 801669 to J.M.M.-D.). L.A.S. was funded by a QMUL PhD studentship and the ERC Starting Grant 950230. The authors thank H. Miyamoto for his help; M. Telford for insightful comments on the manuscript; K. Valendinova for help dealing with overdetermination; and Y. Le Parco for prior inspiration and discussions. The authors acknowledge the pioneering work of the late Q. Bone on chaetognaths and zooplankton.

**Author contributions** F.M., T.G. and N.S. conceived the study. D.G. and L.P. performed the single-cell RNA-sequencing experiments and analyses. L.P. carried out in situ hybridization, phylostratigraphic analyses and SAMap comparisons. J.M.M.-D. helped to set up and interpret in situ hybridization in chaetognaths. F.M., N.S. and D.S.R. sequenced, assembled and annotated the genome. T.G. generated the inbred line of *P. gotoi*. D.G. carried out ATAC–seq experiments and analyses. T.G., K.T.C.A.P. and F.M. collected and spawned adult chaetognaths to obtain embryos. E.P. analysed proximity ligation data and performed the gene family expansion analyses. L.A.S. constructed the Enzymatic Methyl-seq library. A.d.M. analysed the methylation gene toolkit and methylation signal. F.M. conducted comparative genomics and gene family analyses. F.M. and L.P. wrote the paper with input from all authors.

**Competing interests** D.S.R. is a shareholder and advisor to Cantata Bio. The other authors declare no competing interests.

**Additional information**
**Correspondence and requests for materials** should be addressed to Ferdinand Marlétaz.

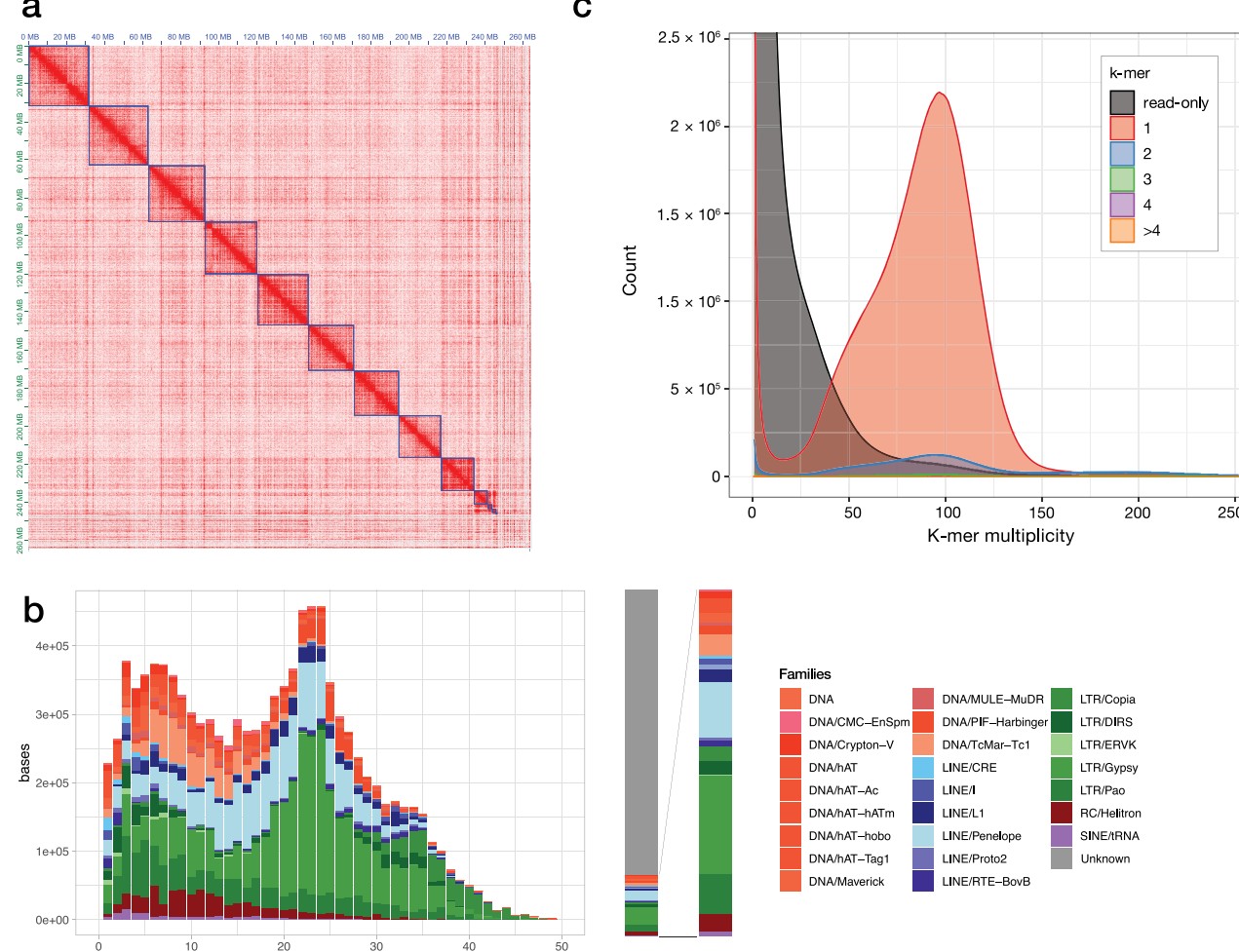

**Extended Data Fig. 1 | Genome assembly and repetitive landscape for *P. gotoi*. a**, Hi-C contact map representing binned contact density across the genome and highlighting the 9 chromosome-size scaffolds of *P. gotoi* consistent with reported karyotype in other chaetognath species. **b**, Repeat landscape summarising the fraction of regions diverging from consensus repeats at varying levels of divergence (Kimura 2-parameters) and the proportion of different repeats annotated in *P. gotoi*. **c**, k-mer spectrum assuming a 21-mer featuring the coverage at various k-mer multiplicity in the assembly and supporting polymorphism homogenisation.

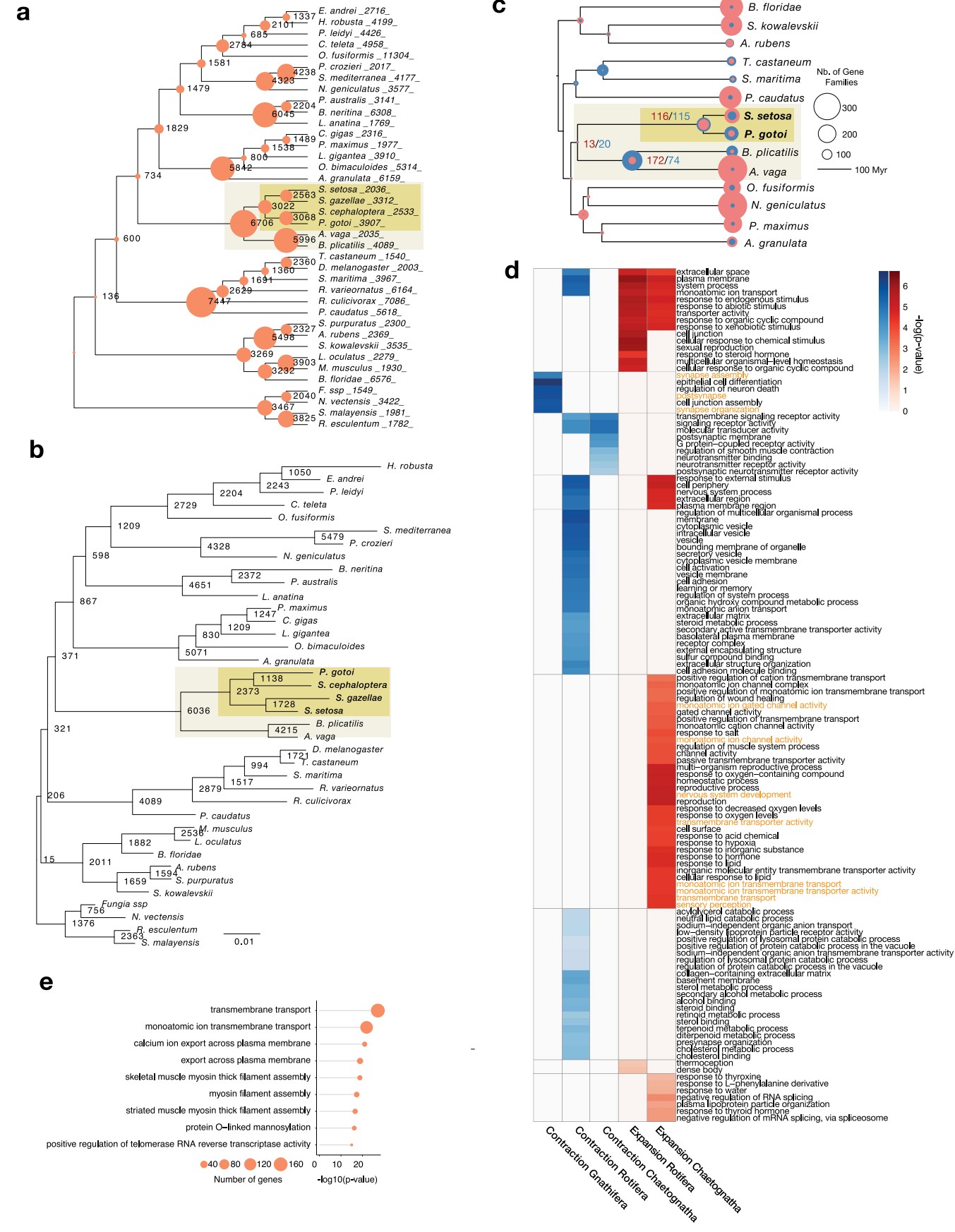

**Extended Data Fig. 2** | See next page for caption.

**Extended Data Fig. 2 | Evolution of gene complement. a**, Loss of PANTHER families inferred on the metazoan tree according to parsimony. **b**. Maximum-likelihood rates of gene loss inferred from GeneRax gene tree/species tree reconciliation. **c**, Number of expanded (red) and contracted (blue) gene families inferred in the CAFE analysis mapped on a time tree. **d**, Gene ontology enrichment for gene families expanded (red) or contracted (blue) at different gnathiferan nodes highlighting terms involved in neuronal function in yellow that appears reduced in the gnathiferan node and expanded secondarily in chaetognaths (Fisher's exact test, two-sided). **e**, Gene ontology enrichment calculated for genes belonging to families expanded in chaetognaths (Fisher's exact test, two-sided).

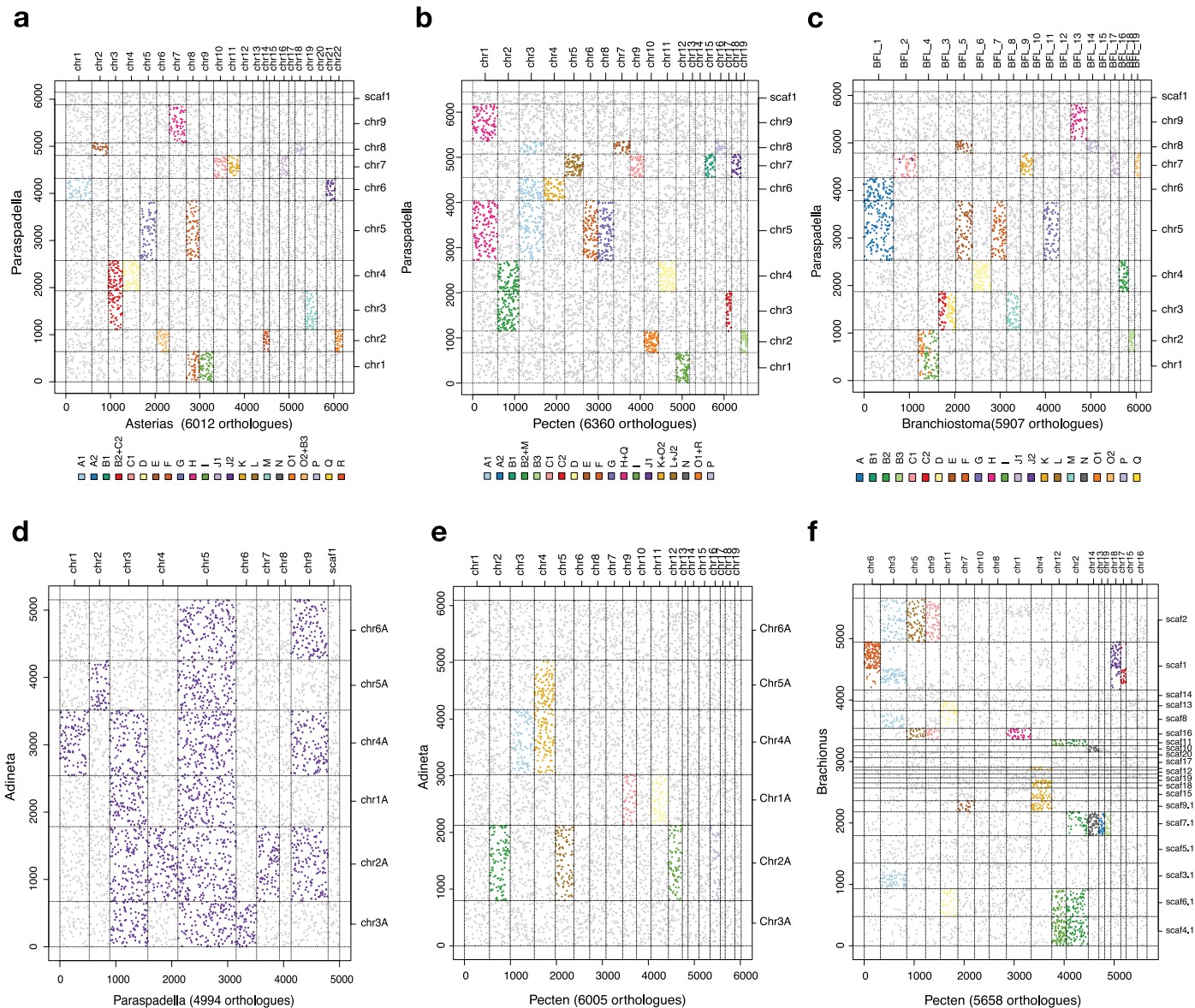

**Extended Data Fig. 3 | Synteny evolution in Gnathifera.** 'Oxford' synteny plots showing the pairwise synteny relationship between the chaetognath *P. gotoi*, the sea star *Asterias rubens* (**a**), the sea scallop *Pecten maximus* (**b**), the amphioxus *Branchiostoma Floridae* (**c**), the rotifer *Adineta vaga* (**d**). A direct comparison between rotifers (**e**, *Adineta saga*; **f**, *Brachionus manjacavas*) and sea scallop highlights the lack of direct correspondence and the loss of ALGs in rotifers. Orthologuous genes are computed as mutual-best-hits (MBH) and coloured according to the inferred ALGs. Dots in pairs of chromosomes that show a reciprocal enrichment using Fisher's exact test (p < 0.05, one-sided) are coloured according to their ALG assignments (**a**,**b**,**d**) or arbitrarily (**c**).

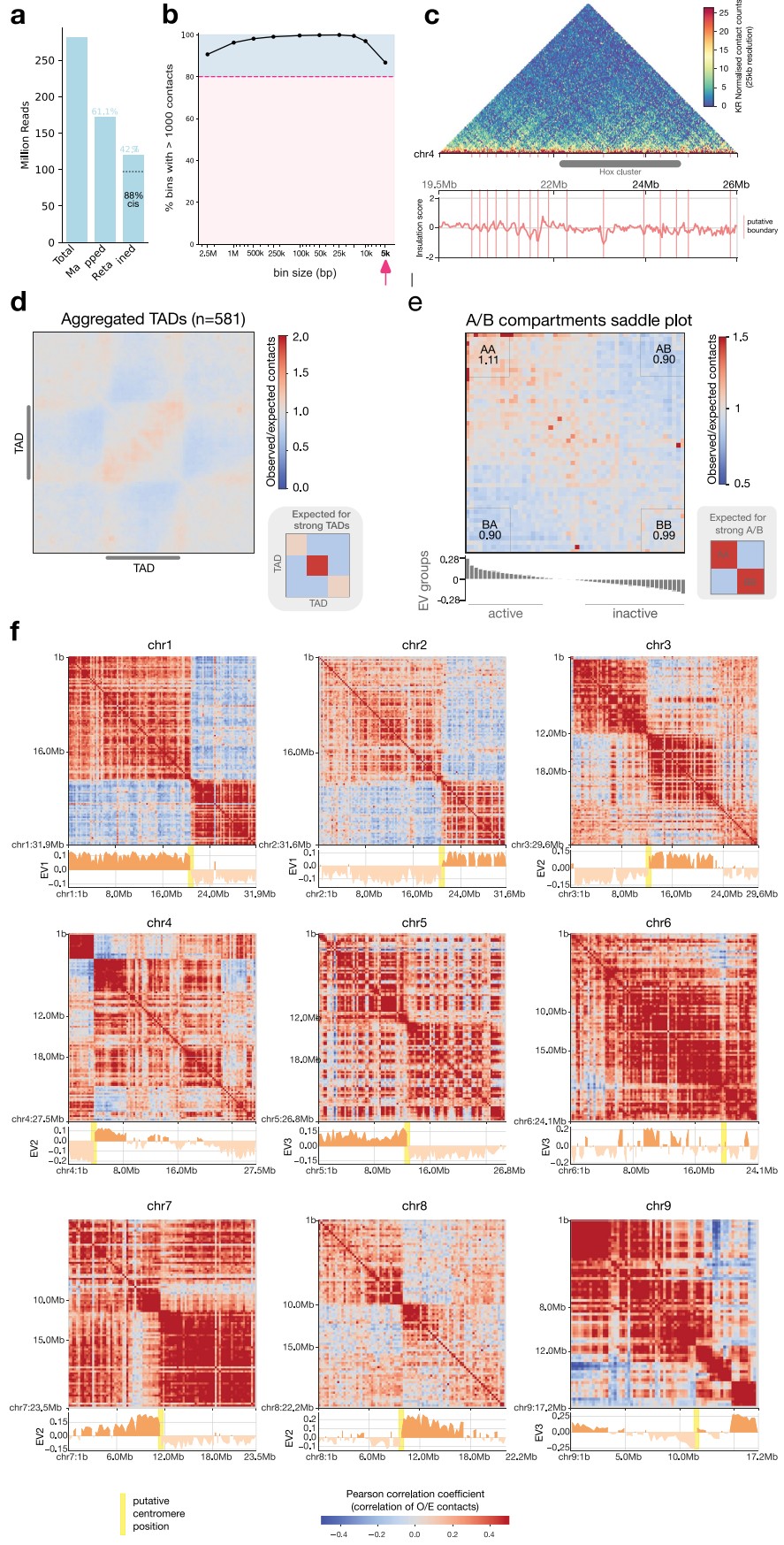

**Extended Data Fig. 4** | See next page for caption.

**Extended Data Fig. 4 | Limited compartimentalisation of *P. gotoi* chromatin. a**, Mapping statistics of the Hi-C library highlighting the number of mapped and inferred contacts used to generate contacts maps including those in *cis*. **b**, Resolution of Hi-C library estimated by computing the number of bins containing >1000 contacts at increasingly higher resolutions. **c**, Hi-C contacts map in the Hox cluster region showing insulation scores and putative TAD boundaries revealing the limited structuration of chromatin in the Hox region. **d**, TAD metaplot showing the limited domain structuration of *P. gotoi* chromatin (average within TADs observed/expected contact ratio = 1.05) as well as the expected pattern in case of strong TAD structuration shown as the bottom right inset. **e**, Saddle plot showing interactions strength across active and inactive regions with inset showing expected pattern for strong A/B compartimentalisation. **f**, Pearson correlation matrices and associated eigenvectors that best capture centromere position, calculated at 250 kb resolution for the 9 chromosomes of *P. gotoi*. Putative centromere positions are indicated (highest eigenvector value differential). Hi-C maps suggest a lack of A/B structures (except possibly for chr5).

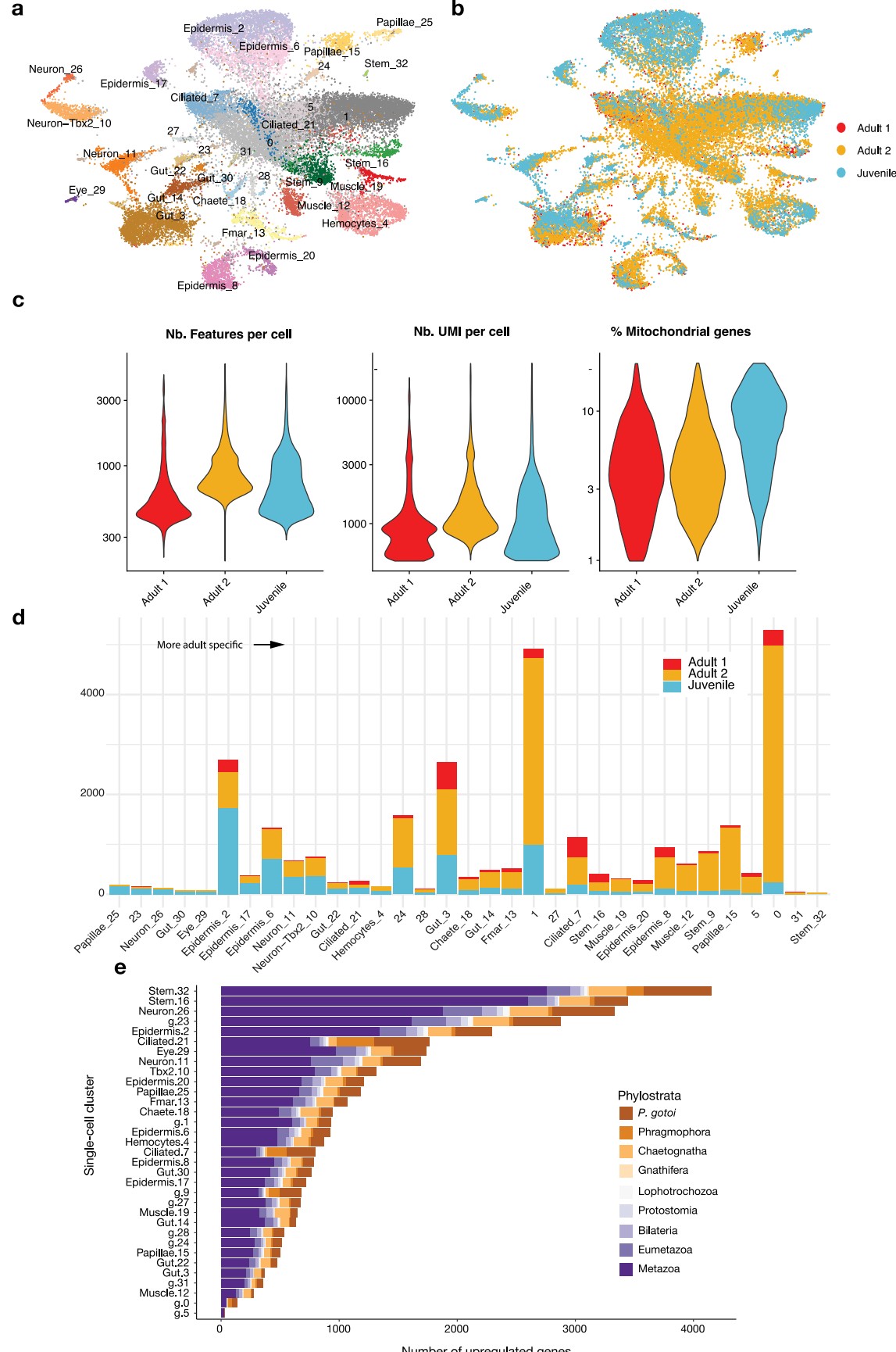

**Extended Data Fig. 5** | See next page for caption.

**Extended Data Fig. 5 | Single cell of juvenile and adult *P. gotoi*. a**, UMAP of all cells with annotated cluster, **b**, same UMAP with individual cells annotated from the sample origin. **c**, Quality control statistics across single-cell samples, including number of features (genes) per cell, number of reads per cell, and percentage of mitochondrial reads per cell. **d**, Number of cells derived from the 3 different samples in each single-cell cluster. Most cell types are shared across juveniles and adults except for one germline cell cluster (cluster 32) and cluster 31 which are specific to adults, while papillae cluster 25 and cluster 23 are mostly composed of juveniles cells. **e**, Number of genes from distinct phylostratigraphic origin in each single-cell cluster.

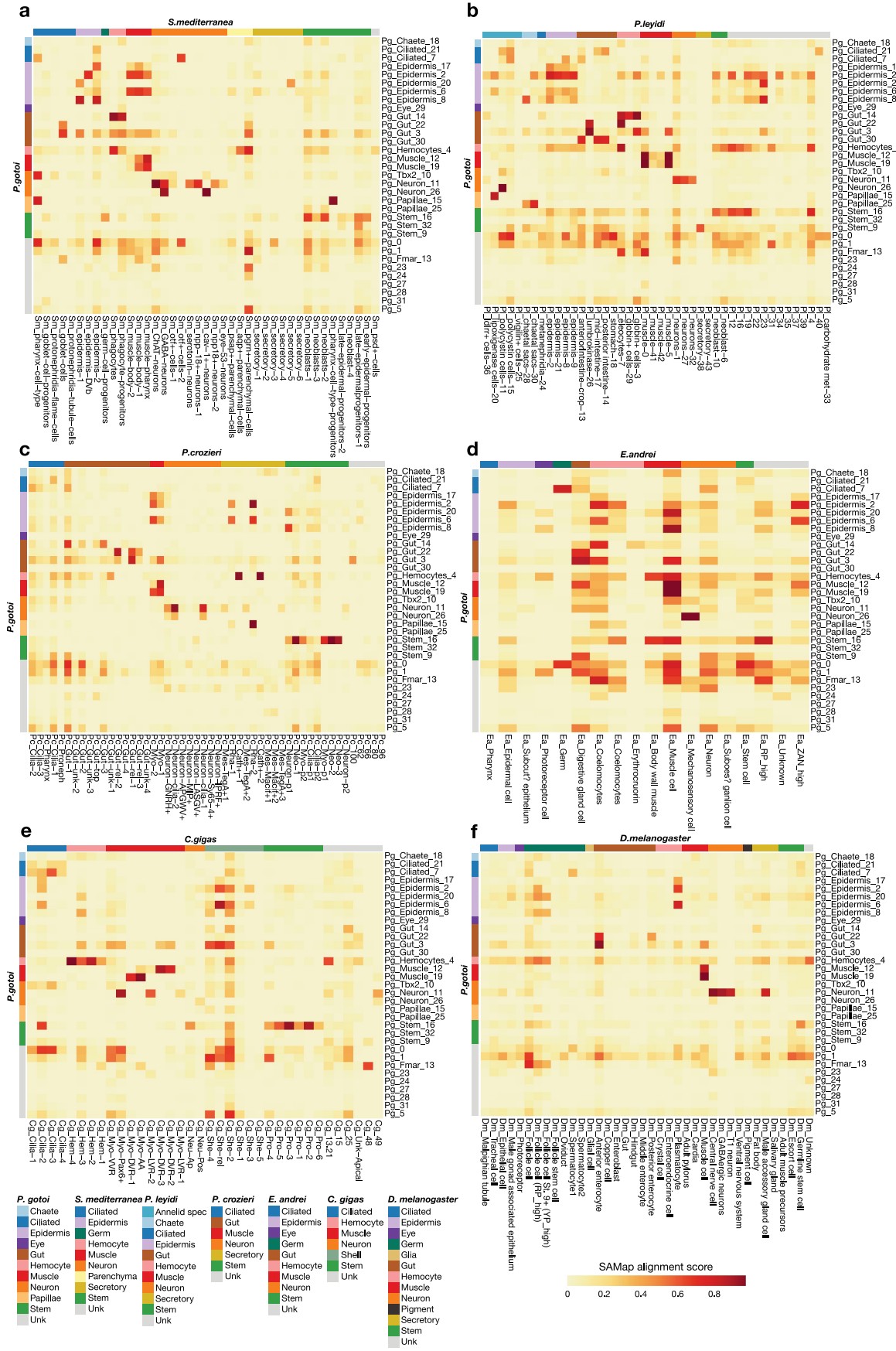

**Extended Data Fig. 6** | See next page for caption.

**Extended Data Fig. 6 | Pairwise SAMap comparison between selected scRNA-seq datasets and *P. gotoi*. a**, platyhelminth Schmidtea mediterranea from (García-Castro et al.[100]). b, annelid Pristina leyidi from (Álvarez-Campos et al.[38]). **c**, the larva of the flatworm Prostheceraeus crozieri from (Piovani et al., 2023)[39] **d**, earthworm Eisenia andrei (annelid) from (Li et al.[40]). **e**, the larva of the oyster Crassostrea gigas from (Piovani et al., 2023)[39] and f, the fruit fly Drosophila melanogaster (Li et al.[40]). The cell-type colour code corresponds to broad putatively homologous cell type categories. The scale bar indicates the alignment score which is defined as the average number of mutual nearest cross-species neighbours of each cell relative to the maximum possible number of neighbours.

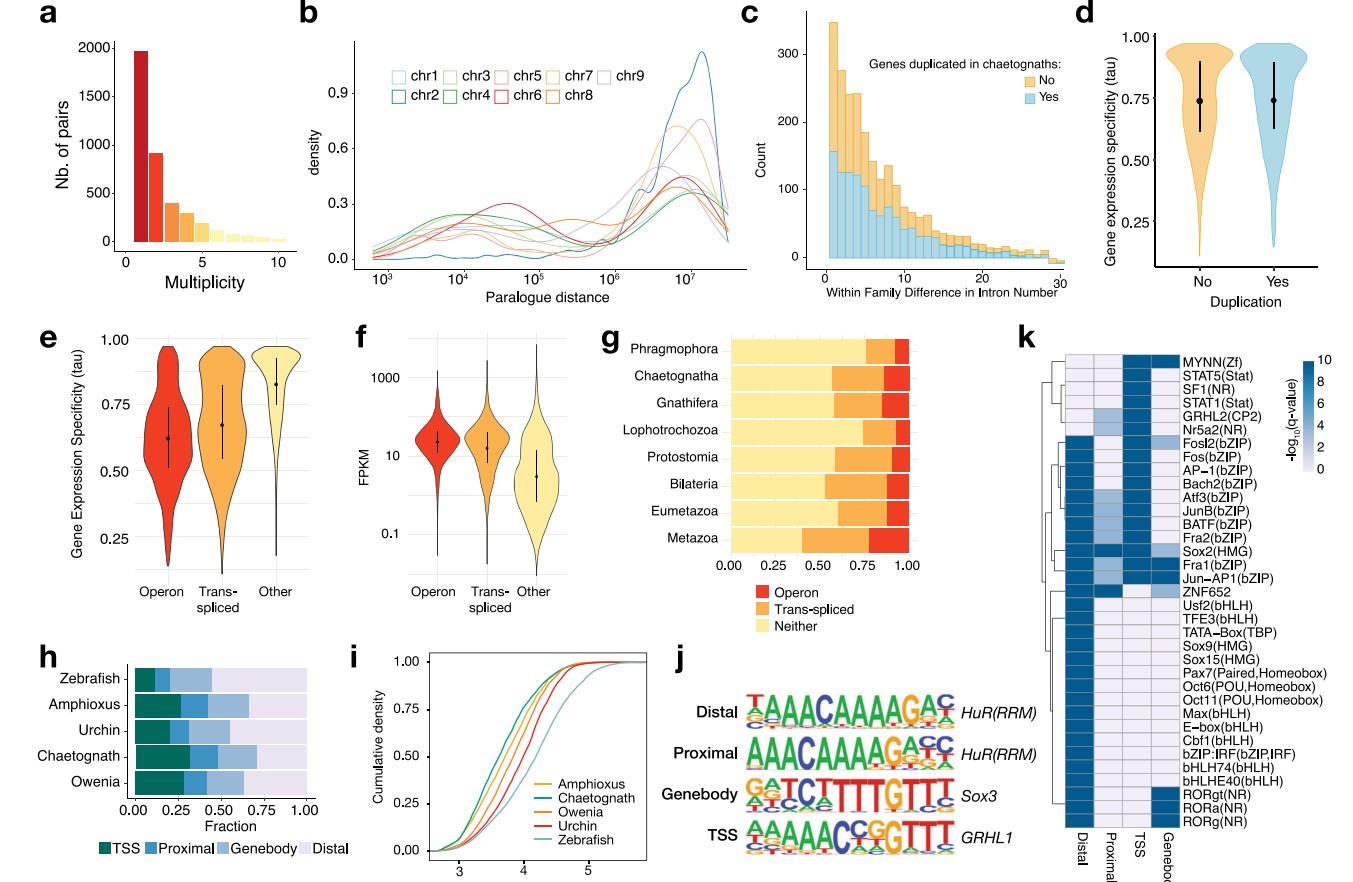

**Extended Data Fig. 7 | Genome duplication and regulation in *P. gotoi*.**
**a**, Number of paraloguous genes (multiplicity) derived from each chaetoganth-specific duplication. **b**, Distance between paralogue pairs belonging to a single chromosome. **c**, Difference between exon number of paralogues belonging to a given gene family for gene family containing chaetognath-specific duplicates or not. **d-e**, Gene expression specificity (tau) computed using averaged expression of main chaetognath cell types for chaetognath duplicates (**d**, n = 15060 and 4250 genes for duplicated and non-duplicated, respectively), as well as depending on their trans-splicing status (**e**, n = 3612, 6342 and 9356 genes for operon, trans-spliced and other genes, respectively). In **d-f**, violin plots with mean and quartiles as error bars.

**f**, Expression level (FPKM) of genes depending on their trans-splicing status (n = 3740, 6457 and 11885 genes for operon, trans-spliced and other genes, respectively); **g**, Distribution of genes of distinct trans-splicing status across phylostratigraphic levels. **h**, Number of Open Chromatin Regions OCR) assigned to various genomic locations (proximal is <5 kb) toward the genome. **i**, Distance to TSS of ATAC-based OCRs in a select species. **j**, De novo motifs detected by HOMER for OCR localised at different genomic positions. **k**, Enriched known motifs detected by HOMER in the different peak categories highlighting the conservation of some transcription factors properties, with bHLH and homeodomains preferentially targeting distal OCRs (one-sided hypergeometric test).

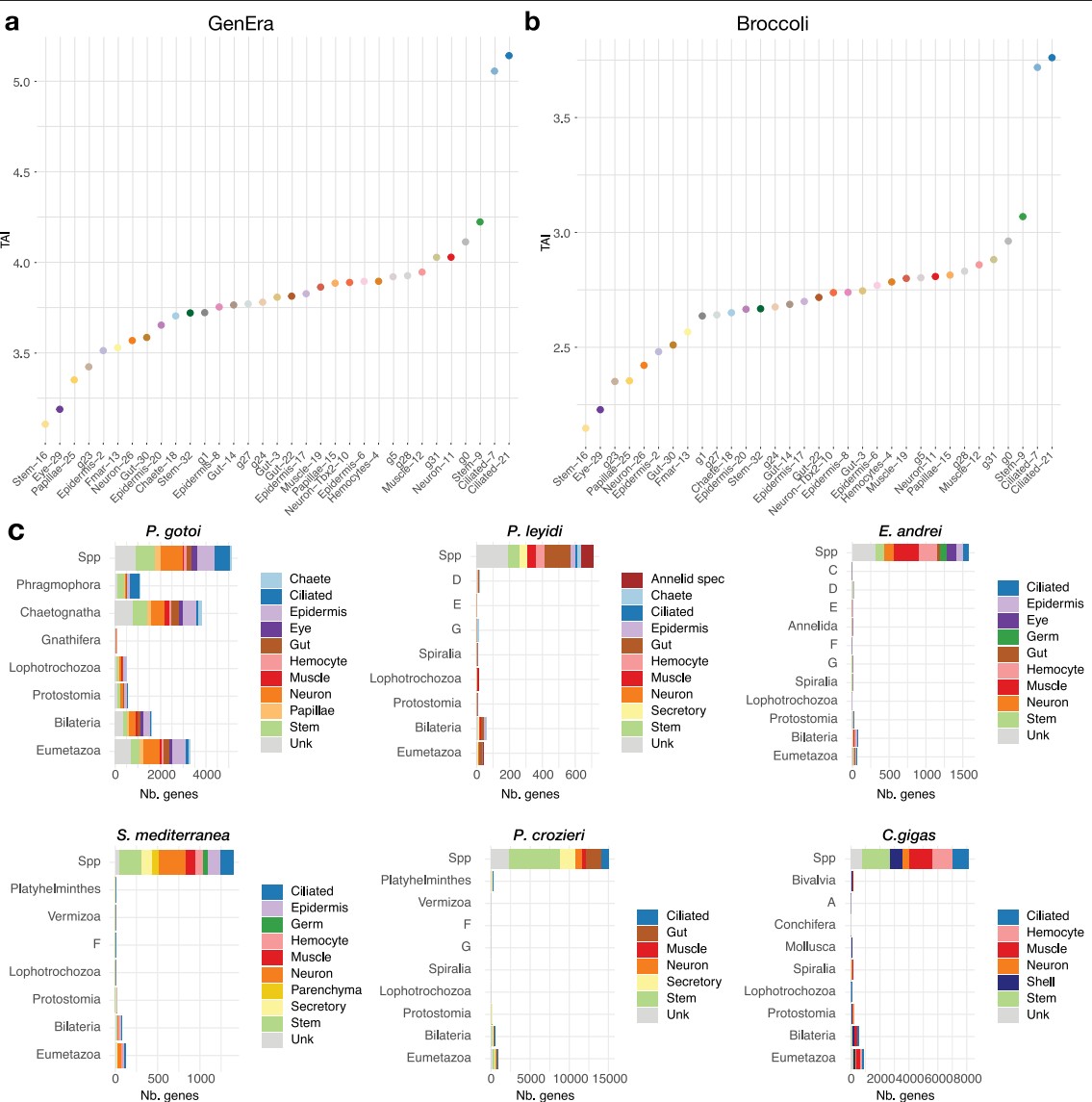

**Extended Data Fig. 8 | Phylostratigraphic analysis of single-cell gene expression clusters. a-b**, Transcriptome Age Index (AI) calculated using gene age derived from GenEra that relies on blast similirity search (**a**) and from the age of inferred Broccoli gene families (**b**). **c**, Phylostratigraphic distribution of cell-type marker genes in *P*. gotoi and other species considered according to the age of Broccoli gene families. **d**, Number of cell-types belonging to broad functional categories in the different species for which we compared single-cell datasets.

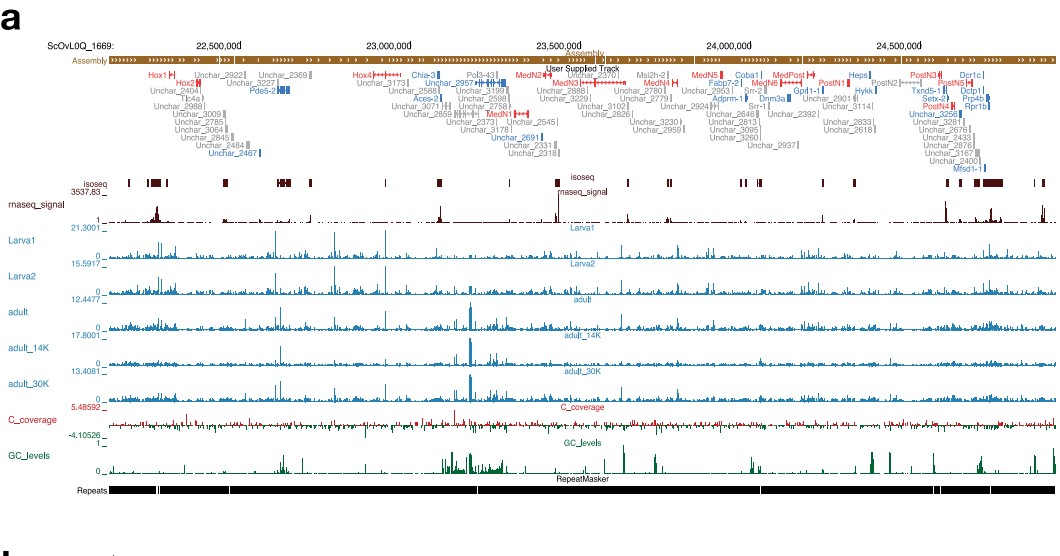

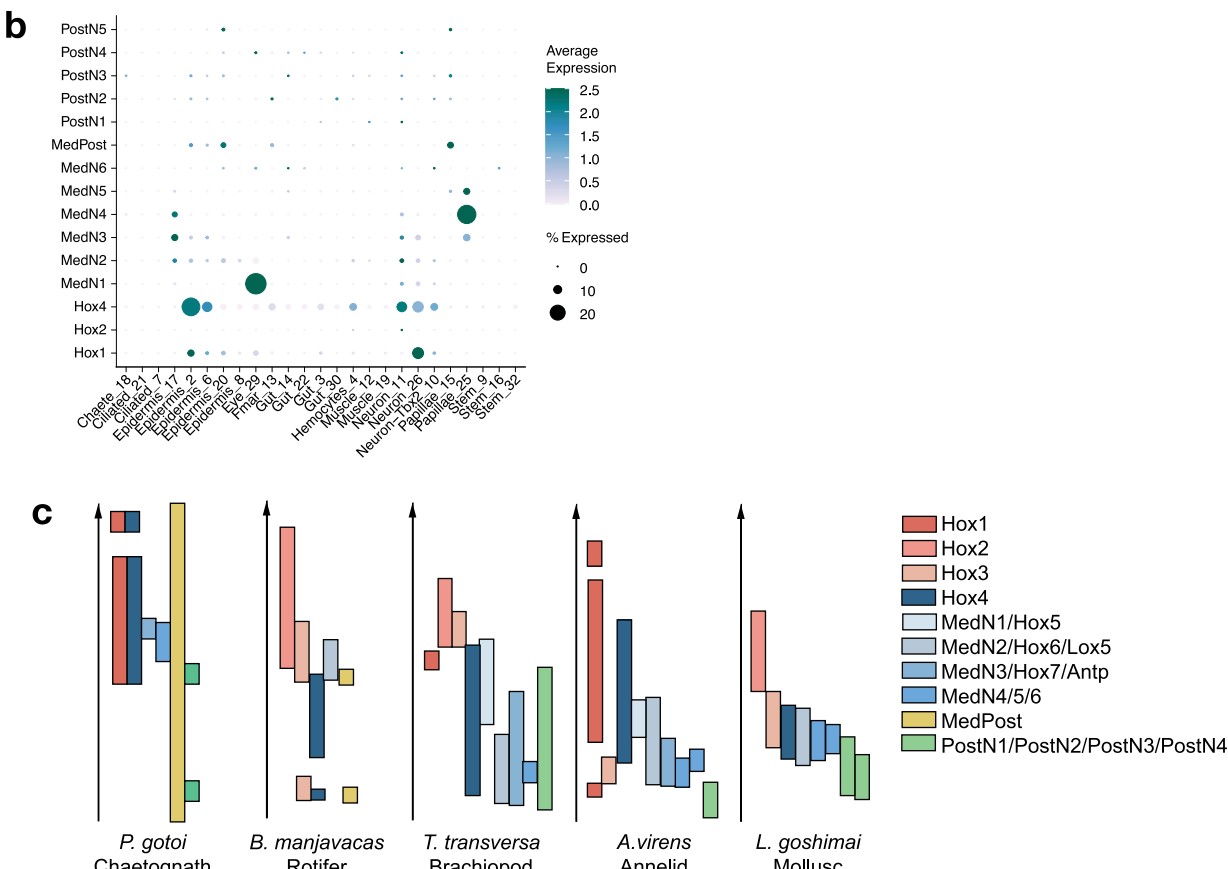

**Extended Data Fig. 9 | Hox cluster and Hox expression in *P. gotoi*. a**, Structure of *P. gotoi* hox cluster with hox genes labelled in red, high-confidence genes with high expression (FPKM > 5) and homology outside chaetognath in blue and other putative genes in grey. Such high-confidence genes include muti-exonic genes encoding for collagen-related or ion-channel peptides. The RNA-seq (dark red), the ATAC-seq signal (blue) and the methylated GC (GC levels) are also displayed. **b**, Dotplot of expression of Hox genes in single-cell clusters showing cluster-specific expression of some Hox genes. **c**, Collinearity schematics for selected spiralians from this work and from Fröbius et al.[18] and Gąsiorowski et al.[104].

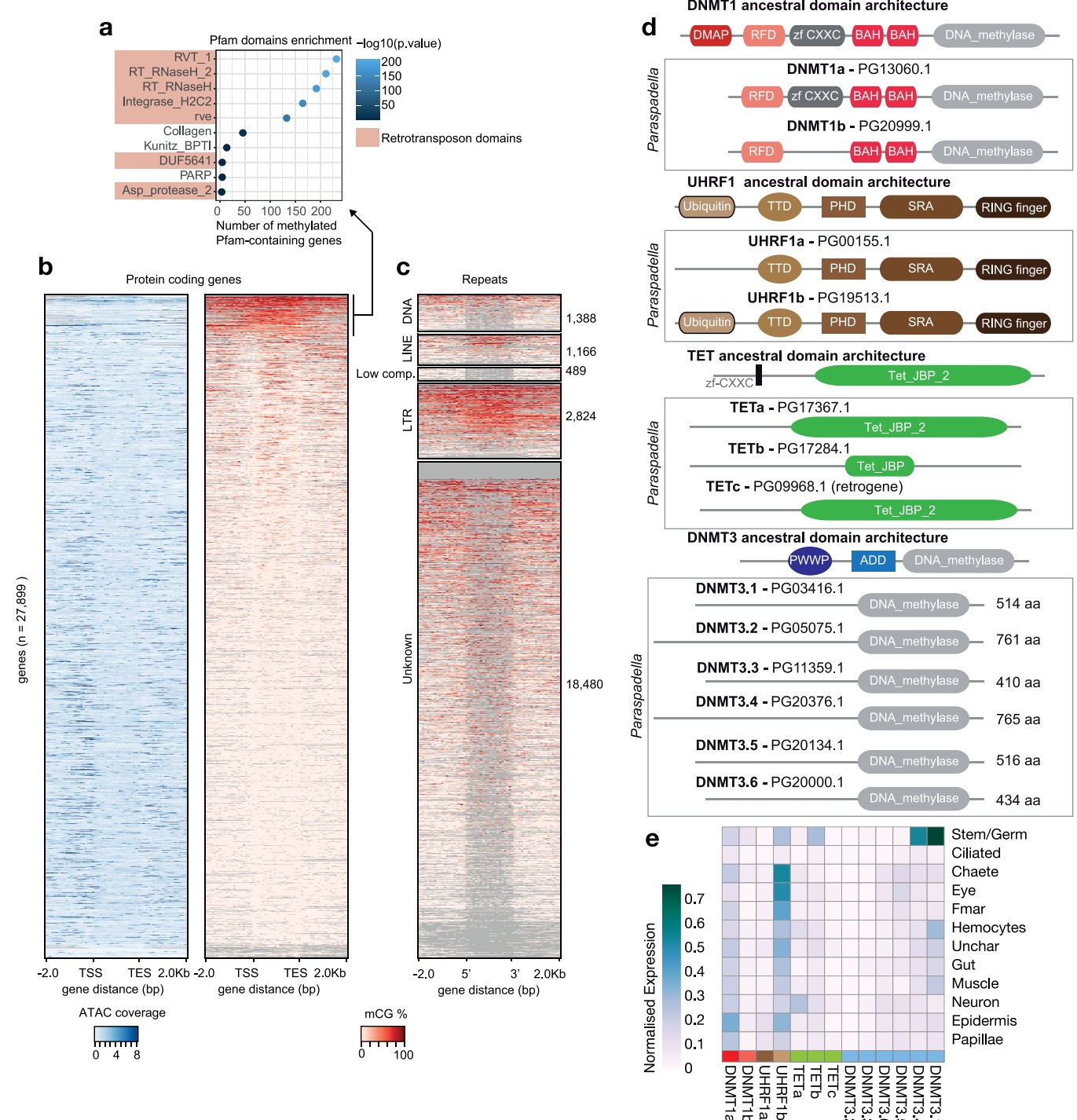

**Extended Data Fig. 10 | A simplified methylation toolkit controls transposable elements in *P. gotoi*. a**, Methylation is only found on transposon-derived sequences. a) Number of methylated genes in Paraspadella (mCG > 20%) that contains enriched Pfam domains (Two sided Fisher exact test p-value < 0.01). Only the top most abundant domains are shown, with all those known to be associated with retrotransposons shaded in red. **b**, Heatmap showing the ATAC-seq and Enzymatic Methyl-seq signal across all gene models, sorted by methylation intensity. **c**, Heatmap showing the methylation levels on the major types of repetitive elements in the *Paraspadella* genome. Only >500 bp repeats/inserts are displayed. Domain architecture of DNA methylation toolkit in *Paraspadella*. **d**, The domain architecture of DNMT1, UHRF1, TET and DNMT3 genes in an ancestral configuration and in *P. gotoi*, defined by PfamA domain presence/absence. RFD (PF12047), zf-CXXC (PF02008), BAH (PF01426), DNA_methylase (PF00145), TTD (PF12148), PHD (PF00628), SRA (PF02182), Ring Finger (PF00097), TET_JBP (PF12851). **e**, Expression of *P. gotoi* methylation toolkit in broad cell types.

# Reporting Summary

## Statistics

For all statistical analyses, confirm that the following items are present in the figure legend, table legend, main text, or Methods section.

| n/a | Confirmed | |
|-----|-----------|---|
| ☒ | ☐ | The exact sample size (*n*) for each experimental group/condition, given as a discrete number and unit of measurement |
| ☒ | ☐ | A statement on whether measurements were taken from distinct samples or whether the same sample was measured repeatedly |
| ☐ | ☒ | The statistical test(s) used AND whether they are one- or two-sided<br>*Only common tests should be described solely by name; describe more complex techniques in the Methods section.* |
| ☒ | ☐ | A description of all covariates tested |
| ☐ | ☒ | A description of any assumptions or corrections, such as tests of normality and adjustment for multiple comparisons |
| ☐ | ☒ | A full description of the statistical parameters including central tendency (e.g. means) or other basic estimates (e.g. regression coefficient) AND variation (e.g. standard deviation) or associated estimates of uncertainty (e.g. confidence intervals) |
| ☐ | ☒ | For null hypothesis testing, the test statistic (e.g. *F*, *t*, *r*) with confidence intervals, effect sizes, degrees of freedom and *P* value noted<br>*Give P values as exact values whenever suitable.* |
| ☒ | ☐ | For Bayesian analysis, information on the choice of priors and Markov chain Monte Carlo settings |
| ☒ | ☐ | For hierarchical and complex designs, identification of the appropriate level for tests and full reporting of outcomes |
| ☒ | ☐ | Estimates of effect sizes (e.g. Cohen's *d*, Pearson's *r*), indicating how they were calculated |

*Our web collection on statistics for biologists contains articles on many of the points above.*

## Software and code

Policy information about availability of computer code

| Data collection | No software used for data collection |
|---|---|
| Data analysis | Assembly: Meraculous (v2.2.2.5) , PBJelly (v15.8.24), Meryl (v1.1), Merqury (v1.3)<br>Annotation: STAR (V2.5.2b, v2.7.8a),  Stringtie (v1.3.3b),Trinity (v2.5.1) ,  GMAP (v2017-09-05) ,Exonerate (v2.2.0), Mikado (v1.2.1), Portcullis (v1.0.2), Trans-decoder (v5.5.0), RepeatModeller (2.0.1) , RepeatMasker (4.1.0) , PASA pipeline (v2.3.1), Eggnog-mapper (v2.1.5)<br>Gene family analyses: MMSeqs2 (r12-113e3), Broccoli (v1.1), MAFFT (v.7.471) , CLIPKIT (v1.1.6), IQTREE (v2.1.1), generax (v2.1.3), GenEra (v1.0.2),MMSeqs2 (r12-113e3)<br>Transcriptomics:  featureCount from subreads package (v1.6.3),<br>HiC: bwa mem (v0.7.17),pairtools (v1.0.2) , juicer_tools (v1.22.01) , fanc python package (v0.9.23)<br>ATAC: bowtie2 (v.2.4.2), Generich (v0.6) HOMER (v5)<br>bulk RNAseq: BBTools (https://jgi.doe.gov/data-and-tools/software-tools/bbtools/bb-tools-user-guide/),<br>scRNA-seq: Cell Ranger (v7/0/1) , SAMAP (v1.0.2 ), Seurat (v5)<br>Methylome: BS-Seeker2 (v2.1.8 ) , Bowtie2 (v2.4.2), Sambamba (v1.0.1), CGmapTools (v0.1.3), DeepTools2 (v2) |

For manuscripts utilizing custom algorithms or software that are central to the research but not yet described in published literature, software must be made available to editors and reviewers. We strongly encourage code deposition in a community repository (e.g. GitHub). See the Nature Portfolio guidelines for submitting code & software for further information.

## Data

Policy information about availability of data

All manuscripts must include a data availability statement. This statement should provide the following information, where applicable:

- Accession codes, unique identifiers, or web links for publicly available datasets
- A description of any restrictions on data availability
- For clinical datasets or third party data, please ensure that the statement adheres to our policy

Supplementary files and codes used for data analysis has been deposited in zenodo: https://doi.org/10.5281/zenodo.13936459
Functional genomics data has been deposited in GEO GSE279488
The genome assembly and associated data have been deposited under project PRJNA1169515
All datasets are publicly availably.

## Research involving human participants, their data, or biological material

Policy information about studies with human participants or human data. See also policy information about sex, gender (identity/presentation), and sexual orientation and race, ethnicity and racism.

| | |
|---|---|
| Reporting on sex and gender | N/A |
| Reporting on race, ethnicity, or other socially relevant groupings | N/A |
| Population characteristics | N/A |
| Recruitment | N/A |
| Ethics oversight | N/A |

Note that full information on the approval of the study protocol must also be provided in the manuscript.

# Field-specific reporting

Please select the one below that is the best fit for your research. If you are not sure, read the appropriate sections before making your selection.

☒ Life sciences ☐ Behavioural & social sciences ☐ Ecological, evolutionary & environmental sciences

For a reference copy of the document with all sections, see nature.com/documents/nr-reporting-summary-flat.pdf

# Life sciences study design

All studies must disclose on these points even when the disclosure is negative.

| | |
|---|---|
| Sample size | Sample size is not relevant to the current study. |
| Data exclusions | No data exclusions. |
| Replication | For ISH, at least 3 individuals were used for qualitative assessment of reproducibility. For single cell and atac-seq data, several individuals were pooled for each stage. |
| Randomization | Randomisation was not applicable to the study. |
| Blinding | Blinding was not relevant to our study as comparison were performed by computer software not influenced by investigator. |

# Reporting for specific materials, systems and methods

We require information from authors about some types of materials, experimental systems and methods used in many studies. Here, indicate whether each material, system or method listed is relevant to your study. If you are not sure if a list item applies to your research, read the appropriate section before selecting a response.

## Materials & experimental systems

| n/a | Involved in the study |
|-----|------------------------|
| ☒ | Antibodies |
| ☒ | Eukaryotic cell lines |
| ☒ | Palaeontology and archaeology |
| ☐ | ☒ Animals and other organisms |
| ☒ | Clinical data |
| ☒ | Dual use research of concern |
| ☒ | Plants |

## Methods

| n/a | Involved in the study |
|-----|------------------------|
| ☒ | ChIP-seq |
| ☒ | Flow cytometry |
| ☒ | MRI-based neuroimaging |

# Animals and other research organisms

Policy information about studies involving animals; ARRIVE guidelines recommended for reporting animal research, and Sex and Gender in Research

| | |
|---|---|
| Laboratory animals | No laboratory animals used |
| Wild animals | Adult hermaphrodite Paraspadella gotoi were collected in Amakusa (Kyushu prefecture, Japan) and maintained in natural seawater. After spawning, juvenile chaetognaths were fed with increasingly older nauplii of the copepod Tigriopus californicus to match their size until capable of feeding on adults. All animals were sacrificed for the experiment of the paper (DNA-extraction, fixation, single cell sequencing). |
| Reporting on sex | Animals are hermaphrodites |
| Field-collected samples | We did not collect any field samples. |
| Ethics oversight | No ethics oversight needed |

Note that full information on the approval of the study protocol must also be provided in the manuscript.

# Plants

| | |
|---|---|
| Seed stocks | *Report on the source of all seed stocks or other plant material used. If applicable, state the seed stock centre and catalogue number. If plant specimens were collected from the field, describe the collection location, date and sampling procedures.* |
| Novel plant genotypes | *Describe the methods by which all novel plant genotypes were produced. This includes those generated by transgenic approaches, gene editing, chemical/radiation-based mutagenesis and hybridization. For transgenic lines, describe the transformation method, the number of independent lines analyzed and the generation upon which experiments were performed. For gene-edited lines, describe the editor used, the endogenous sequence targeted for editing, the targeting guide RNA sequence (if applicable) and how the editor was applied.* |
| Authentication | *Describe any authentication procedures for each seed stock used or novel genotype generated. Describe any experiments used to assess the effect of a mutation and, where applicable, how potential secondary effects (e.g. second site T-DNA insertions, mosiacism, off-target gene editing) were examined.* |

