## [Peer Review file · Nature]

The genomic origin of the unique chaetognath body plan

Corresponding Author: Dr Ferdinand Marlétaz

Version 0:

Reviewer comments:

Referee #1

(Remarks to the Author)

Piovani et al. present a chromosome-scale genome and single-cell transcriptomics atlas for the chaetognath *Paraspadella gotoi*. Chaetognaths are one of the most mysterious animal lineages, with few species known, a highly modified morphology and, until recently, uncertain phylogenetic affinities. This study constitutes a major step forward in our understanding of this fascinating animals, revealing a highly rearranged genome structure, gene content, and cell type repertoires. The paper is well-written, effectively highlighting the main findings and providing compelling interpretations and discussions. I have no major concerns regarding the methodologies or computational analyses employed.

Main comments

- In the part about 3D genome organization, the authors briefly mention the absence of A/B compartmentalization and Topologically Associating Domains (TADs). If the resolution of the reported Hi-C maps permits (acknowledging that these were primarily generated to assemble the genome rather than to achieve high-resolution chromatin interaction profiles), it would be valuable to comment on the presence or absence of chromatin insulation more generally. Additionally, discussing whether chromatin loops are observed—or absent—would enrich the analysis. A deeper discussion of the implications of the observed or unobserved genome architectural features in these organisms would further contextualize these findings.

- The scRNA-seq data analysis and interpretation could be enriched in multiple ways:

1. First, it would be helpful to present all scRNA-seq data in a gene-by-single-cell expression heatmap or similar visualization. This would provide a more immediate impression of the diversity of cell states captured in the dataset.

2. Second, the authors could extend to cross-species comparisons by describing more explicitly key genes supporting cell type similarities (particularly transcription factors).

3. Third, the comparative scRNA-seq analysis focuses on broad cell type definitions (myocytes, neurons, stem/germ) and it would be important to explicitly define how these were defined/grouped. Particularly in the case of “ciliary” cells, as many different cell types are ciliated. Reciprocally other, well-defined, cell types like hemocytes are omitted, while

4. Finally, I'm intrigued by the lack of global similarity between chaetognath neuronal cells and neurons in other bilaterians (given the strong transcriptional similarities very often observed between neurons across phyla). Is there any evidence that chaetognath neurons are highly derived? Do they express genes typically associated with neurons (pre/post-synaptic scaffold, ion channels, RNA binding proteins, certain TFs, etc.)?

- While the study highlights TE methylation in chaetognaths, this pattern is also observed in vertebrates, nematodes, and annelids (e.g., <https://genomebiology.biomedcentral.com/articles/10.1186/s13059-024-03346-z>). Thus, the chaetognath DNA methylation pattern appears less exceptional in terms of TE methylation and more so in the loss of coupling to H3K36me3 and gene body methylation. However, given our limited knowledge of DNA methylation patterns in many other animal lineages, the broader context remains unclear. Similarly, the correlation between a lack of gene body methylation and trans-splicing, while intriguing, seems weak. In fact, there are compelling alternative hypothesis linking the coevolution of DNA methylation and DNA repair systems (<https://www.nature.com/articles/s41588-018-0061-8>), rather than gene

regulation mechanisms. So, overall, I am not sure one can call this a “reversion to pre-animal DNA methylation”.

- The generated ATAC-seq data for adult and larval stages is underutilized in the paper, which represents a missed opportunity. What is the genomic distribution of cis-regulatory elements (CREs) in the Chaetognath genome? How frequent are non-promoter (intergenic or intronic) CREs and how distant from the nearest TSS? How do the enriched motifs in the CREs (representing putative transcription factor binding sites) compare to those known for bilaterian species? Generally these motifs are highly conserved for most TFs, but it would be exciting to check if this is the case also for Chaetognaths.

Additional comments

- The methods section would benefit from including additional QC metrics for the functional genomics assays. For example, report the percentage of trans-contacts and mappability for Hi-C experiments, FRiP scores and saturation levels (e.g., percentage of duplicates) for ATAC-seq datasets, and sequencing saturation metrics for scRNA-seq libraries.

- Provide detailed information about the Hi-C experiments directly in the manuscript, rather than referring to a previously published study.

- In panel 5b it would be better to show the “canonical” bilaterian protein structural domain architecture in comparison to the “simplified” version in *Paraspadella*.

- Show scRNA-seq stats in SFig 2c in log-scale.

- It would help if *Paraspadella* cell types in SFig. 6 were shown in the same order in each heatmap.

Referee #2

(Remarks to the Author)

Chaetognaths are an important but understudied phylum of marine invertebrate predators recently shown to be spiralian, closely related to rotifers. In the current manuscript, Piovani et al. combine a new chromosome-scale genome of the chaetognath *Paraspadella gotoi* with multi-omics analysis, including single-cell RNA-sequencing data and Methyl-seq analysis, to study the evolutionary origins of the chaetognath phylum. Due to their mesoplanktonic nature, obtaining sufficient high-quality DNA for comprehensive genomic analysis has been challenging. The authors should be commended for overcoming this difficulty and successfully generating a chromosome-level genome. Using these genomic resources, the authors demonstrate extensive gene family loss and the emergence of new gene families in the chaetognath lineage, with new gene families being recruited to novel cell types. They also reveal several interesting aspects of the chaetognath genome, including extensive tandem duplication without whole-genome duplication (WGD), the loss of centromere proteins, and methylation of transposable elements but not protein-coding genes.

Overall, the authors present a substantial and valuable dataset, including a chromosome-scale genomic resource for chaetognaths, which holds significant potential for advancing our understanding of the evolution of spiralian and gnathiferans. The manuscript is highly original and of broad interest to researchers in fields such as invertebrate zoology, genome structure evolution, cell type evolution, and methylation. However, the manuscript would benefit from a more coherent narrative to effectively connect the various datasets and provide a comprehensive and informative story. This enhancement would help the manuscript resonate more with readers and align with the standards of Nature. In addition to structural concerns, several other issues mainly related to phylogenetic interpretation, genome structure evolution, and gene family evolution are outlined below.

Major comments

1. The introduction would be more engaging for Nature readers if it emphasised the broader implications of the findings rather than detailing technical approaches. Highlighting how this study enhances our understanding of the evolution of chromosomal structures and gene regulation in early branching spiralian would make the introduction more impactful. Additionally, the manuscript would benefit from greater coherence among datasets. Establishing a clearer linkage, such as demonstrating how the genomic data, single-cell transcriptomics, and expression analyses collectively support key evolutionary insights, would strengthen the narrative. Currently, the manuscript reads as a collection of distinct data points that would gain from a stronger, unified storyline.

2. Despite the last author's expertise in spiralian phylogeny, there are significant phylogenetic inaccuracies that need to be addressed. The manuscript shows fundamental phylogenetic issues and a lack of comprehensive understanding, which raises concerns about the robustness of the gene family gain and loss analysis—especially as a reliable phylogenetic tree is essential for meaningful interpretation. One example is Fig. 1b, where *P. australis* and *L. anatina* (not *L. anatica*) are shown as sister groups, with this combined group depicted as the sister group to *B. neritina*, which has not been reported in the literature. Additionally, errors such as labelling Lophophorata as Platyhelminthes require correction. The inconsistencies between Fig. 1b and Fig. 4a should also be resolved to maintain coherence throughout the manuscript and improve the reliability of the analyses.

3. Phylogenetic terms such as "Spiralia" and "Lophotrochozoa" should be clearly defined in the text, and more precise labelling in Fig. 1b would enhance reader comprehension. While the manuscript provides extensive discussion on the rotifer genome, placing greater emphasis on the chaetognath-specific data from this study would create a more balanced narrative and add depth to the analysis. Given that rotifers exhibit rapid chromosomal evolution, the availability of a chaetognath genome presents an excellent opportunity to reconstruct gnathiferan genome evolution, as their synteny is relatively more tractable. This point will be elaborated upon in a later comment.

4. In the gene family analysis, more attention should be given to the top GO terms related to biological function, such as insulin-like receptors and SMAD proteins. These pathways are significant as insulin signalling controls body size, and SMAD proteins play roles in TGF-beta signalling. The authors should carefully annotate which components of these pathways are missing, particularly for SMAD proteins, which have distinct roles (e.g., receptor-regulated SMADs such as Smad1/2/3/5/8, co-SMADs like Smad4, and inhibitory SMADs such as Smad6/7). Noting that Smad2/3 mediates TGF- β /Nodal signalling and Smad1/5 is involved in BMP signalling would provide insights into how these pathways evolved in chaetognaths.

5. For the loss of CenH3 genes, which ranked among the top findings, the authors should clarify the process used to confirm the true absence of these genes. Absence in gene annotations alone is not sufficient; additional evidence such as BLAST searches of the genome sequence should be provided. Ideally, identifying microsyntenic regions where these genes are found in closely related species and confirming their absence in the chaetognath genome would strengthen this conclusion.

6. One advantage of having a chromosome-scale genome is the opportunity to explore genome structure evolution through synteny analysis. It is notable that this aspect has not been fully explored in the manuscript. Simakov et al. (2022) proposed an ancestral spiralian karyotype with 20 linkage groups based on data from molluscs and annelids (all sharing H \otimes Q, J2 \otimes L, K \otimes O2, and O1 \otimes R fusion events). This study could provide the first evidence confirming these findings in a more early-branching spiralian lineage. The authors are encouraged to discuss how chaetognaths share spiralian fusion-with-mixing events, expanding on this section to make the findings more accessible.

7. It appears that the authors used a colour scheme for ancestral linkage groups similar to that in Martín-Zamora et al. (2023), but Fig. 1e in the current manuscript does not match Fig. 1d of Martín-Zamora et al. Clarifying whether different versions of the Pecten genome were used would be helpful. The authors should ensure comparability across studies by using the same genome assembly version. Notable differences include:

- Pecten Chr 1 (H \otimes Q) appears as Chr 18 in Fig. 1e, showing fission into two chromosomes.
- Pecten Chr 4 (J2 \otimes L) corresponds to Chr 16 in Fig. 1e and stays intact with fusion-with-mixing.
- Pecten Chr 5 (K \otimes O2) corresponds to Chr 19 in Fig. 1e, showing fission followed by fusion-with-mixing.
- Pecten Chr 10 (O1 \otimes R) corresponds to Chr 3 in Fig. 1e, remaining intact with fusion-with-mixing.

Discussing these findings and comparing them with other studies on genome architecture would emphasise their significance and better integrate them into the broader context of spiralian genomics. Additionally, consider moving Fig. 3d to accompany Fig. 1e for improved visual coherence.

8. Following the previous comment, the synteny analysis raises several points that need clarification. The comparison between *Paraspadella* and *Pecten* appears significantly different when viewed in the ribbon plot (Fig. 1e) compared to the dot plot (Supplementary Fig. 3b). It would be beneficial to clarify what gene sets are represented in each plot and the criteria used for their selection. The assumption might be that the genes shown in the ribbon plot represent the subset of orthologues with significant enrichment in Fisher's exact test (coloured genes in Supplementary Fig. 3b), but this does not seem to hold true. For instance, ALG Q is missing in the dot plot but appears prominently in the ribbon plot (e.g., on chromosomes 9 and s1). The main text mentions that ALGs A2, B2, L, N, and Q were not detected in *Paraspadella*, yet these are visible on specific chromosomes in Fig. 1e (e.g., A2 on chromosome 1, B2 on chromosome 4, and Q on chromosomes 9 and s1). Clarification is needed on why these ALGs appear in the figure if they are not statistically enriched, and vice versa. Furthermore, while the text states that the spiralian fusion-with-mixing H \otimes Q is not detected, it seems present on chromosomes 9 and s1 in Fig. 1e. An additional consideration is that a fourth fusion (O1 \otimes R) has been identified in other spiralian (Simakov et al. 2022). Determining whether this fusion is present in chaetognaths and discussing its relevance would provide further insight into the evolutionary context of the data.

9. The integration of the single-cell atlas and Hox gene expression data could be expanded to emphasise evolutionary implications. Specifying which cell types express Hox genes and discussing their relevance to spiralian evolution would help readers understand how lineage-specific traits are reflected in single-cell data.

10. Including a schematic illustration of Hox gene expression in Fig. 4 would be beneficial, particularly to highlight any spatial collinearity similar to that seen in other species. This could be presented as a cartoon illustration comparing expression collinearity with rotifers and other outgroups, such as annelids and molluscs, and incorporating data from additional spiralian species for broader context would add value. Such comparisons could align well with the Hox gene expression patterns shown in the single-cell dataset. In addition, arranging the *in situ* data so that the anterior of the worm is oriented to the left would ensure consistency with Fig. 2d and other bilaterian studies.

11. The manuscript in its current form lacks a clear connection between lineage-specific genes and novel cell types. Strengthening this aspect would help underline how these new gene families contribute to the development of unique chaetognath cell types.

12. The identification of the MedPost gene positioned between central and posterior Hox genes is intriguing, though not

unexpected, as its close relationship to these Hox groups has been noted in transcriptomic studies. However, the manuscript leaves the identity of central Hox genes largely unresolved. Given that the authors successfully recovered the Hox cluster with their high-quality genome assembly, including a gene tree (and potentially trees for all Hox genes) would strengthen the study and provide valuable insights into Hox gene evolution in this lineage. Determining whether MedPost is more closely related to central or posterior Hox genes would be particularly informative. Additionally, the methods section should detail how Hox genes were identified and verified, even if the process is straightforward. Combining phylogenetic analysis with motif analysis would be especially useful for identifying certain Hox genes that possess lineage-specific diagnostic peptides, such as the PG6 motif (KS(I/L)ND in gnathiferans) adjacent to the homeodomain. Given the availability of high-quality Hi-C data, it would also be informative to investigate whether there is a topologically associating domain (TAD) correlated with the unique structure of the Hox cluster.

13. The proposed order of evolutionary events—suggesting that the gnathiferan lineage underwent genomic simplification involving gene loss and disruption of ancestral linkages, followed by the mobilisation of lineage-specific duplicated and novel genes—could be elaborated. Alternative explanations, such as simultaneous gene loss and gain or initial gene gain followed by loss, should be briefly discussed. The authors should specify the strongest evidence for their proposed order and consider precedents in other lineages.

14. Fig. 3f is not referred to in the main text, and its figure legend lacks clarity on the conclusions being drawn. Including a reference in the main text and clarifying its relevance would improve the manuscript.

Minor comments

Text (the authors are encouraged to use continuous line numbers):

Page 2 Line 23 – Typo: "ressources" should be "resources."

Page 2 Line 48 and throughout – Supplementary subfigures are referred to out of order, which can be distracting. Consider reordering figure panels or adjusting the text references for consistency.

Page 4 Line 27 – Define the acronym "TAD" when it is first mentioned.

Page 6 Line 8 – The phrase "as well as could be" is awkward. Consider rephrasing for clarity.

Page 6 Line 40 – Add a brief explanation of what chaete cells are for readers unfamiliar with the term.

Page 7 Line 23 – References to main text panels appear out of order. Reorganise for a smoother reading experience.

Page 8 Line 9 – Supplementary Fig. 8 is referred to before Fig. 7. Adjust the order for clarity.

Page 8 Line 39 – The sentence "ciliary cells appear among the ones with the most recent transcriptome" is unclear. Add context to explain what phylostratigraphic gene ages mean and what this result implies.

Page 10 Line 28 – The phrase "suffered duplications" implies a negative connotation. Use a more neutral term, such as "experienced duplications."

Page 12 – There are several references to non-existent figures (e.g., 4l, 4k) and incorrect mentions of Figure 4, which is the Hox figure. Correct these references.

Page 15 Line 35 – Define "dph" when first used.

Main figures:

Fig. 1c – Clarify if "neg. reg." is "negatively regulated." If not, please describe it.

Fig. 1d – Explicitly state what the grey boxes represent. Also, correct the order of references in the legend (e.g., e and d).

Fig. 1e – Ensure consistent labelling, such as using "Chr s1" uniformly. Additionally, clarify the rationale behind naming it "s1," as this is not well explained.

Fig. 1e and throughout – Correct the typo "Adinetta" to "Adineta."

Fig. 1e legend – Consider stating explicitly that these are bilaterian linkage groups.

Fig. 1f legend (Line 18) – The reference to "Supplementary Figure 3" should be "Supplementary Figure 4." Terms like "A/B compartmentalisation" and "trans-splicing" should be defined when first used to prevent readers from needing to search for definitions.

Fig. 2a legend – Mention explicitly that the data includes both hatchling and adult cells.

Fig. 2d – Add cell type labels directly to the in situ images for better panel clarity.

Fig. 4a – Label species names to provide more context on Hox cluster organisation. Including the Hox cluster organisation in the Ecdysozoa outgroup would further enrich the analysis.

Supplementary figures:

Supplementary Fig. 1 – Italicize *P. gotoi*.

Supplementary Fig. 2 – Highlight chaetognaths in plots a, d, and e for easier identification. Clarify what the numbers in plots d and e represent. Correct species names (e.g., *L. anatina* instead of *L. anatida*, *P. australis* instead of *P. austratlis*). Use the correct formatting for *Fungia* sp. instead of *Fungia* ssp. (do not italicise "sp.").

Supplementary Fig. 3 – Italicise genus names throughout and ensure consistent terminology when labelling *Pecten* as either scaffold or chromosome (note that the correct version of the *Pecten* genome should be used).

Supplementary Fig. 4 – The caption should say "Supplementary Figure 4," not "Supplementary 4." Add more detail to part a to help readers unfamiliar with the figure's significance.

Supplementary Fig. 5 – Consider ordering the plots by a useful criterion (e.g., adult-specific ones grouped together) rather than alphabetically.

Supplementary Fig. 6 – Add labels and units to the legend in the plot.

Supplementary Fig. 7 – Correct the typo "infered" to "inferred."

Supplementary Fig. 8 – The term "multiplicity" is somewhat unclear; consider rephrasing or elaborating on its meaning for better understanding.

Methods:

Clarify which type of Hi-C technology was used.

Specify the read depth per cell for the single-cell data.

Correct "SAMAP" to "SAMap."

Remove references to the acoeel worm *H. miamia* if the data is not included, and note that acoeels are not flatworms.

Referee #3

(Remarks to the Author)

I enjoyed reading the manuscript by Piovani and collaborators, which provides novel insights into the origin and evolution of the phylum Chaetognatha and a first look into the evolutionary patterns underlying the diversification of the Gnathifera. Chaetognaths are striking beasts that present a unique body plan, and this study looks into genomic, regulatory, and cellular aspects to understand their emergence and distinctive features. The results point to accelerated genomic evolution in the gnathiferans and chaetognaths, with a mix of novel patterns but also gene loss, although it is still early days for gnathiferans as their sampling is still scarce. I believe the datasets generated and the analyses done reflect the state-of-the-art in the field and provide original and novel insights on the evolution of the clade, I'd like to congratulate the authors for the enormous effort to put together these datasets given how hard it is to work with these organisms.

I believe there are no major issues with the data, analyses or conclusions, and I can only happily recommend the publication of this manuscript. I do have some minor questions and suggestions, mostly aimed at making the manuscript to the general audience of a journal such as Nature.

- 1) I noticed the authors did not refer to or compare their results against the two recent papers from Prof Timothy G Barraclough lab (Oxford) on rotifer genomes (Wilson et al Trends in Genetics 2024; Nowell et al Nat Comms 2024). I understand this rotifer genome data came out very recently and I would not expect the authors to add it to their analyses. However, I'd like to encourage the authors to incorporate the findings of these papers in their discussion section.
- 2) Annotation of gene loss: the methods mention eggNOG mapper and goatools, but I couldn't find which species was used to annotate lost genes. Given that these are absent in chaetognaths/gnathiferans, I think the text should clarify how these were identified.
- 3) I think the finding on CENP genes being lost in gnathiferans is interesting. That said, I'd like to ask why there is no more elaboration on the role of regulators of the insulin-like receptor signalling pathway and SMAD protein signal transduction, given the biological function of these genes in development and the fact that more gene families have been lost for these categories than any others.
- 4) Fig 1b, please review clades labelling, Brachiozoa/Phoronozoa are mistakenly labelled as Platyhelminthes, and the flatworms branch has no label.
- 5) Fig 1b and across the manuscript, I find that the use of species abbreviations gets in the way of accessibility. For example, Figure 2c is hard to follow unless you are familiar with the species of invertebrates for which single-cell data is available. At least the first time these species are named, the full genus name should be used, and Fig 1b seems the perfect place for this.
- 6) Fig 1e, Adinetta has only one letter t.
- 7) Figure 3 could use some more/better labelling. For example, Fig 3a could indicate what is the meaning of the numerical value indicated in each of the curves. Fig 3d, what is the meaning of the ball size and the grey colour?
- 8) Figure 3b, I'd like the manuscript to elaborate on why the matrix is not symmetrical. I understand the matrix represents the number of paralogs pairs shared between chromosomes, and this should be the same value no matter in which direction the comparison is made (e.g., like in the example given for WGD).
- 9) Could the authors expand on what's going on with chromosome 5? It seems like a very "hot" chromosome!
- 10) Fig 3f, is this truly "phylostratigraphic" distribution (as in one way BLAST from one single anchor species against one representative for each of any other lineages) or just good old phylogenetic bracketing?
- 11) Page 8, lines 6-7 and sentence in line 8. The duplicate multiplicity or the multiple duplicability, makes this hard to follow. I wonder if these sentences could be streamlined to make them more accessible.
- 12) Page 8, some recent papers have highlighted the possibility of orthology assignment artefacts caused by the fast-evolving nature of some lineages, such as chaetognaths. This would cause an excess of new genes, as well as of lost ones. This is a significant limitation in comparative genomics analyses that should be mentioned in this study. Please, see Natsidis P et al iScience (2021), Weisman et al PLoS Biol (2020), and Guijarro-Clarke et al Nature Ecology & Evolution (2020)
- 13) Page 8, line 44, the sentence has a teleological whiff, please rephrase.

- 14) Fig 4 could use a legend, for example, what are the grey boxes in the Hox genes figure?
- 15) Page 10, line 38. I am no expert on methylation, but is the lack of gene body methylation really a “unique” case of reversion? How does this relate to genome size?
- 16) Fig 5b, I have checked different displays, and I have a hard time finding the difference between shades.
- 17) Trans-splicing findings are fascinating. Due to its prevalence in the chaetognaths genome (50% of the genes), I understand that most of the genes that are trans-spliced in chaetognaths are not in other taxa, is that correct? How many of the chaetognath-specific genes are trans-spliced, or is this a “feature” of older genes?
- 18) Page 13, line 22, I believe the sentence should be “provides clues to the origin” instead of “provides clues in the origin”.

Referee #4

(Remarks to the Author)

Review of manuscript by Piovani et al. “The genomic origin of the unique chaetognath body plan”

Summary of key results, originality, and significance:

Piovani et al. present an impressive genomic study of the enigmatic phylum Chaetognatha, integrating a range of disciplines and frontier methodologies to investigate its origin at the genomic, regulatory, and cellular levels. The phylogenetic position of this old and ecologically important pelagic group was long debated, with the group showing both deuterostome and protostome characteristics. However, its position within Spiralia has now been established through several recent phylogenomic studies. In the context of this phylogenetic position, Piovani et al. now provide a timely and highly relevant, deeper insight into the evolutionary history and function of the aberrant genomic and morphological traits of Chaetognatha.

The study integrates information from organismal to cellular level, from chromosomal to functional genomic levels, and boldly propose new links between geno- and phenotypes. Overall, the study presents a wealth of original data on this understudied group, which allows for highly interesting comparisons across Metazoa and for erecting new hypotheses – not only on the origin of Chaetognatha but on general genomic evolutionary mechanisms underlying adaptations of animal body plans. In sum, the study has the novelty and potential of high relevance and vast impact on a broad readership from many different scientific disciplines.

Conclusions, data and methodology:

The many different analyses generally represent state of the art (or beyond) within their disciplines and the breadth and depth of results are impressive (e.g. on centromeric structures, gene loss and duplications, and specific gene functions), especially considering that this non-model organism has not previously been studied in any such details. Still, I have a few concerns on central parts of the work (see below under methods/results) and on the phrasing of some of the conclusions.

Piovani et al. keeps referring to Gnathifera as a clade including Chaetognatha, although most phylogenomic studies (including Piovani et al.) shows Chaetognatha to be sister to Gnathifera (see and cite Laumer et al. 2019, <https://doi.org/10.1098/rspb.2019.0831>).

Although it can seem as barely semantics, there are very good arguments in Bekkouche & Gasiorowski 2022 (<https://doi.org/10.1080/14772019.2022.2109217>) for instead using the name Chaetognathifera for this common clade. I strongly recommend the authors to replace the name Gnathifera with Chaetognathifera when referring to this clade throughout the manuscript. This will also make it easier for the reader to follow when they are referring to Gnathifera ‘sensu stricto’ vs Chaetognathifera.

The Piovani et al. references to Chaetognathifera evolution seems a bit one-sided and B&G 2022 should as a minimum be discussed and cited, since it convincingly positions Amiskwia next to extant Chaetognatha (in Cuculophora) and thoroughly reviews the morphological characters and their evolution, such as the likely lack of homology between Chaetognatha grasping spines (by the authors called ‘hooks’?) and gnathiferan jaws. Instead Piovani et al. multiple times refer (only) to the Vinther & Parry 2019 paper, which first was followed by a more detailed fossil study (Caron & Cheung 2019) and later was shown to include multiple erroneous character codings and likely misinterpretations of the evolutionary history (e.g., Bekkouche & Gasiorowski 2022).

Moreover, Piovani et al. base some of their conclusions on the assumption that the Chaetognathifera lineage has underwent genomic miniaturization before the expansion/duplications found in Chaetognatha. It is unclear to me how they can draw this conclusion so solidly from their data when they mainly compare to a highly aberrant (according to themselves) bdelloid rotiferan genome and miss data from several of the other gnathiferan lineages (as well as other Spiralian lineages). This should be further explained as some of the main conclusions rest on these results.

Related to this, Chaetognathifera is now regarded (also by the authors) as a clade sister to the remaining Spiralia and to my understanding it is still equally possible that the Chaetognathiferan ancestor may have possessed ancestral Spiralian genomic properties rather than a miniaturized design. The authors do mention this caution in the conclusion, but less throughout the manuscript. It seems uncouthly to give this uncertainty a bit more weight in the discussion, and then put less emphasis on the proposed pre-miniaturization/losses in Chaetognathifera in the conclusion - without devaluating the main conclusions/hypotheses on the genomic evolution of Chaetognatha.

Likewise, the text in between closely link the small body size of gnathiferan taxa to their genome size, and gene loss to genome miniaturization, dependencies that are not always clear or established (e.g., also substantial gene loss at other nodes, suppl. Fig. 2e,d). Furthermore, the calculated gene losses for the Chaetognathifera lineage likely depend on the limited number of included gnathiferans and might suffer from the missing Gnathostomulida and other minor spiralian phyla, and potentially be affected by the divergent position (and paraphyly) of Platyhelminthes. This is apparently not discussed at present and might deserve more attention. More detailed comments on specific analyses are given below.

The centromere discussion on page 4 could benefit from including more comparative information on rotifers. Topology of Figure 4 needs further explanation, the Parenchymia and Tetraneuralia clades on Figure 4 are quite controversial.

Specific comments to methodology, data:

Contamination: One potentially major issue is that the authors seemingly do not perform any decontamination steps. This should be standard for all new genome assemblies but particularly because this assembly is primarily an Illumina based assembly. Using Blobtools could be an option. Since the animals were lab bred and fed exclusively on *Tigriopus californicus*, it would be easy to check specifically for contamination from food sources in addition to the standard Blobtools decontamination pipeline. It is recommended to perform this step prior to duplicate purging and scaffolding. There is ample genomic data for *T. californicus* on SRA that can be used for this. Were the animals starved before extraction? Maybe a little more detail is needed here in the methods.

Potentially uncollapsed heterozygosity and strange Kmer spectrum: The authors accounted for heterozygosity in the experimental design by inbreeding for five generations. However, even if the loss of heterozygosity from inbreeding was "perfect", after five generations of full sib mating, you'd still expect there to be ~3% of the heterozygosity of the original parent (50% loss heterozygous sites per generation for five generation = 0.5^5). Further, percent heterozygosity of the genome predicted was not reported. The authors admit chaetognaths are expected to have high heterozygosity so there might not be enough generations to effectively control it in a pool of 20 individuals. Because of this issue, the lack of a haplotype purging step is problematic. The assembler was set up to merge haplotypes but assemblies generally still require/benefit from a devoted haplotype purging step. There are some concerns regarding the kmer spectrum in supplementary figure 1. There is no heterozygous peak but there is a long "tail" where the peak is expected (~50x coverage). You'd expect the count of the single copy kmers to approach zero whereas in Piovani et al., the count levels off at 1/2 the count of the homozygous peak. This might indicate that the kmer size is incorrect (maybe play around with this value a little) or more worryingly, there are uncollapsed haplotypes that were scaffolded in tandem. Purge_dups are recommended on the decontaminated assembly prior to scaffolding. If there are uncollapsed haplotypes that made it into the final assembly, this could seriously affect some of the major conclusions of the paper (i.e. gene duplication/gain, hox gene analysis).

Analysis of gene gain and loss: The section describing the use of GenEra for phylostratigraphy should be expanded on to include the parameters used for the program.

In Supplementary Figure 2A, CAFÉ was used, however there are no methods to report what parameters were used for the input tree (ultrametric tree production, fossil calibrations, etc) or for the running of the program (i.e. gamma rate or error correction).

In situ's and Hox genes: Page6, l39-l44 authors compare "chaete" cell type with "epidermal" cell type both expressing paralogues of chitin synthase and chitinase. Chs2-5 is located close to the grasping spines in Figure2d. But it is less clear in the text that Vwf-2 was used as a marker of the "epidermal" cell type expressing chitin synthase and chitinase. This should be stated more clearly in the text. Based on the presence of several epidermal cells, and literature on the pluristratified epidermis in chaetognath, the Vwf2 gene, showing faint and uniform signal, might not be the best to label the epidermis cells expressing chitin synthase and chitin. Instead, the epidermal marker Chs2-3 might better serve as comparison with "chaete" cells marked with Chs2-5 and the argument of a spine synthesis function more likely. In Figure 4 the authors show Hox genes in situ performed on juvenile (in legend)/ hatchling (in text). Authors should clarify the stage as it is inferred from the method that a juvenile would be a 3dph. Moreover, the relevance of the life stage used to show Hox expression is seemingly not justified, neither by morpho-developmental nor gene expression data. Please elaborate. Could the authors use the RNAseq dataset to provide a temporal expression of the Hox at gastrula, hatchling and 3dph that could support the choice of life stage for In situ's? Piovani et al. claim that MedPost is expressed in papillae peripheral sensory neurons, showing a new function, and they do show in Supl. Figure 9 that MedPost is expressed in cluster cell Papillae-15. However, Supl. Figure 9 also shows that MedPost is expressed in epidermis_20. The authors should provide a full view of Pxdn-9 in situ in Fig. 2 to support the claim that MedPost signal is in papillae peripheral sensory neurons throughout the body and explain the seemingly double signal in "epidermis-20" cluster cells.

Adinetta should be Adineta (spelled wrong throughout figures)

Version 1:

Reviewer comments:

Referee #1

(Remarks to the Author)

The authors have comprehensively addressed all my concerns and suggestions. This is an excellent study and I fully endorse its publication in Nature.

Referee #2

(Remarks to the Author)

The authors have thoroughly addressed our previous comments in this revision. We commend their success in resolving the mystery of their unique body plan through an integrative genomic and multi-omic approach. We have no further major concerns, but would like to offer a few minor corrections and suggestions to improve clarity and presentation.

Line 47 – lophotrochozoans -> lophotrochozoan

Line 54 – bodyplans -> bodyplans

Line 86 – it says “rotifers and to a lesser extent chaetognaths” but the numbers show that it should be to a greater extent in chaetognaths.

Line 142 – domain -> domains

Line 222 – the abbreviation is no longer in the figure

Line 304 – there is at least one further case of Hox gene duplication: the zen/shx gene of lepidopterans, which can have up to 165 copies: See Mulhair et al. 2023 (<https://genome.cshlp.org/content/33/1/32.full>) and Mulhair et al. 2024 (<https://www.sciencedirect.com/science/article/pii/S1084952122003573>).

Referee #3

(Remarks to the Author)

I enjoyed reading the revised manuscript by Piovani and collaborators, and I'd like to thank the authors for addressing my concerns.

About annotating lost genes in chaetognaths, I'd usually prefer to use a closer relative to minimise the impact of the divergence time. The amphioxus-chordates-deuterostomes nodes may have their own gene losses obfuscating the annotation. In the immortal words of Philip H. Pope (1921), "it's a long way from amphioxus" (apologies, this is my best comeback to the Haldane quote). That said, I understand that the “closest” genomes available are very messy, and cephalochordates are a reasonable compromise.

I believe the manuscript has significantly improved and I support its publication.

Referee #4

(Remarks to the Author)

I would like to congratulate Piovani et al. for having done an amazingly thorough and comprehensive revision as well as a well-argued and detailed response to reviewer letter. I believe their impressive and highly interesting work shines even brighter now :-). Main figures have been very nicely revised and now reads clearly. Additional analyses (and additional suppl. figures), additional methodological explanations, and text rephrasing further supported their findings and refined their conclusions. The few raised concerns rebutted, are well argued for in their response letter. I have no further comments, and can only highly recommend publication of this very interesting and intriguingly integrative study.

A few minor typos caught:

line 47: lophotrochozoans to lophotrochozoan

line 54: bodyplans to body plans

line 87 and 88: '1456 and 2165, respectively' '116 and 172' - switch order of figures to reflect previous line giving order rotifer, chaetognath, respectively

Please see below our detailed response to referees. Please note that per journal guidelines, we renamed **Supplementary Figures** as **Extended Data Figures** and introduced in parallel 3 new **Supplementary Figures** for more targeted panels describing confirmatory analyses.

Referee #1

Piovani et al. present a chromosome-scale genome and single-cell transcriptomics atlas for the chaetognath *Paraspadella gotoi*. Chaetognathes are one of the most mysterious animal lineages, with few species known, a highly modified morphology and, until recently, uncertain phylogenetic affinities. This study constitutes a major step forward in our understanding of this fascinating animals, revealing a highly rearranged genome structure, gene content, and cell type repertoires. The paper is well-written, effectively highlighting the main findings and providing compelling interpretations and discussions. I have no major concerns regarding the methodologies or computational analyses employed.

We thank the reviewer for their positive appreciation of work. We did our best to answer their comments, see below.

Main comments

- In the part about 3D genome organization, the authors briefly mention the absence of A/B compartmentalization and Topologically Associating Domains (TADs). If the resolution of the reported Hi-C maps permits (acknowledging that these were primarily generated to assemble the genome rather than to achieve high-resolution chromatin interaction profiles), it would be valuable to comment on the presence or absence of chromatin insulation more generally. Additionally, discussing whether chromatin loops are observed—or absent—would enrich the analysis. A deeper discussion of the implications of the observed or unobserved genome architectural features in these organisms would further contextualize these findings.

As noted by the reviewer, we attempted to exploit Hi-C data generated for genome assembly to improve the characterisation of chromatin architecture in chaetognaths. While characterisation of A/B compartments is usually possible using medium to low resolution Hi-C, identification of TADs, and even more so loops requires higher resolution data. In the original version of this manuscript, we used 25kb resolution matrix to compute the distribution of insulation scores and selected the strongest boundaries (boundary score > 0.5) to derive an aggregate plot (**Extended Data Fig. 4d** in revised version). This plot reveals limited intra-domain contacts, which is not indicative of the presence of TADs.

Given the relatively small size of the chaetognath genome (257Mb), it could be argued that smaller domains might be present but not captured with our analysis performed at 25kb resolution. We had previously estimated the maximum resolution of our contact map using a state-of-the-art method (Rao et al., 2014) (**Extended Data Fig 4b**), which suggested the data could be used at finer resolutions (10kb and 5kb). Upon visual inspection, 10-5kb contact maps appear however very sparse (see *Figure R1*). Discrepancies between inferred resolution using the Rao et al. method (developed for mammalian genomes) and visual inspection might be due to the low structuration of the chaetognath genome. Of note, a resolution of 10kb-5kb would be required to call loops, but our contact map cannot be used at these resolutions, and, when attempting to call loops using SIP (Rowley et al., 2020) with default parameters, we recovered, as anticipated, zero significant loop.

Figure R1: Contact map of Hi-C data at 3 resolution (25, 10 and 5kb) in the Hox region.

We agree with the reviewer that the presence of loops and/or TADs in the chaetognath genome would suggest that these structures were already present in the bilaterian ancestor. We however cannot confidently comment on the presence of loops and remain cautious in our proposition for an absence of TADs. As the reviewer points out, our Hi-C map was primarily generated for genome scaffolding - not to achieve high-resolution interaction profiles, we therefore refrain from suggesting an evolutionary scenario for the evolution of TADs or loops. We have added more contextualisation of our results in the discussion as follows: *“The absence of detectable TADs in chaetognaths would require further confirmation using higher resolution methods (e.g. micro-C⁸⁹). Such sequence insulated chromatin domains, which have recently been shown to be a bilaterian innovation⁶⁷, could hence be less widespread than anticipated.”*

- The scRNA-seq data analysis and interpretation could be enriched in multiple ways:

1. First, it would be helpful to present all scRNA-seq data in a gene-by-single-cell expression heatmap or similar visualization. This would provide a more immediate impression of the diversity of cell states captured in the dataset.

We agree with the reviewer that the scRNA-seq analysis could be improved and have followed most of their suggestions (see below); however we feel that a gene-by-single-cell expression heatmap is less legible than a dotplot, particularly to the wider audience. Dotplots are commonly used for single cell representation as they reduce noise and are able to capture both expression level and percentage of cells expressing the gene which is useful to represent general vs specific markers as needed for **Figure 2b**.

Figure R2: Heatmap of cell type marker gene expression.

2. Second, the authors could extend to cross-species comparisons by describing more explicitly key genes supporting cell type similarities (particularly transcription factors).

Following the reviewer suggestion we have identified key co-expressed genes across cell types of different animals and have added a few sentences in the text (see below); we also now provide all shared genes in **Supplementary Table 9**. As shown in **Figure 2c** we find an abundance of shared orthologous genes, however among these we find only a handful of TFs. *P. gotoi* neurons share the expression of the TF Bc11a with both flatworms (*P. crozieri* and *S. mediterranea*) and the fruit fly *D. melanogaster*. Neurons of the fruit fly also co-express the TFs Cot1, Csde1 and Meis1 with *P.gotoi*. Finally we find the TF Yboxh-2 is co-expressed among stem cells of *P.gotoi* and both flatworms, the fruit fly *D.melanogaster* and the pacific oyster *C.gigas*. We believe that this lack of conserved transcription factors co-expression could be in part explained by technical differences between datasets, specifically varying gene depth, as TFs tend to have lower expression levels (see **EDF 7c**) and differences in stages compared (*C.gigas* and *P.crozieri* are larval datasets). However, most single cell studies find very little overlap in transcription factors expression at these evolutionary distances even when datasets are generated using the same technology and life stages (see for example (Li et al., 2022; Piovani et al., 2023; Seb e-Pedr os et al., 2018)). To further confirm that the lack of TF co-expression was not specific to chaetognath cell types we decided to perform another SAMap comparison, this time between the annelid *E. andrei* and the flatworm *S. mediterranea* and similarly found that although some of their cell types (namely hemocytes, stem cells, muscle cells and neurons) co-express

several genes there are very few TFs (see **Supplementary Figure 2** and **Supplementary Table 9**). Specifically, neurons and hemocytes don't share any TFs, stem cells co-express Hmg2-2, Hmg2-3 and Smca5 and muscle cells co-express Nfkb1-5. Accordingly, we made extensive edits to the single cell section (**Ancestral and novel cell-types in chaetognaths**).

3. Third, the comparative scRNA-seq analysis focuses on broad cell type definitions (myocytes, neurons, stem/germ) and it would be important to explicitly define how these were defined/grouped. Particularly in the case of "ciliary" cells, as many different cell types are ciliated. Reciprocally other, well-defined, cell types like hemocytes are omitted, while

We thank the reviewer for their input, we now provide all the cell type labels and their broad annotation for the comparative scRNAseq datasets in **Supplementary Table 8**. We also thank the reviewer for drawing attention to hemocytes which we had accidentally overlooked. They are now included in updated **Figure 2c**.

4. Finally, I'm intrigued by the lack of global similarity between chaetognath neuronal cells and neurons in other bilaterians (given the strong transcriptional similarities very often observed between neurons across phyla). Is there any evidence that chaetognath neurons are highly derived? Do they express genes typically associated with neurons (pre/post-synaptic scaffold, ion channels, RNA binding proteins, certain TFs, etc.)?

We thank the reviewer for their comment, however we didn't mean to say that all of chaetognath's neurons are highly derived (in terms of gene expression profile). In fact we find that general neuronal markers such as Elav4-1, calcium and sodium channels (Cac1a-2, Cac1a-3, Cac1g-2, Scna-3, Scna-4) several neurotransmitter receptors (glycine and glutamate Gira1-2, Gira2-3, Gria2-1, Grik2-4, Grik2-11) and synaptic genes (Syt4, Acha9-1) are expressed in neurons from clusters 11 and 26. These two neuronal clusters also map well to neurons of several other species with which they share expression of several orthologous genes and even a few TFs (see answer above for more details, **Figure 2c** for number of shared genes per species and **Supplementary Table 9** for a full list). However, other sensory cell types identified appear to be chaetognaths specific, namely the peripheral sensory papillae (papillae 15 and 25) and those associated with ciliary sense organs (Neuron_Tbx2 and Fmar_13).

"Among ancestral cell-types, we identified two neuronal clusters (clusters 11 and 26) which express the neuronal marker Elav, several synaptotagmins, calcium (e.g. Cac1a-2) and sodium (e.g. Scna-3) channels, a dopamine-synthesizing enzyme (Ddc) and can be distinguished by distinct glutamate receptors (Grik2, Grm) and degrading enzymes (Glna) (Figure 2b). From their shared neuronal marker Cac1a-2 expression, they are located both in the ventral and cephalic ganglion and as well as in some peripheral neurons, an abundant cell type in the chaetognath nervous system 47 (Figure 2d). These neuronal clusters map well to that of other protostomes with which they share the expression of the TFs Bc11a, Cot1, Csde1 and Meis and the splicing factor Rfox3 (Figure 2c, Supplementary Table 9)."

"As for chaetognath-specific cell types, those that appear unrelated to any other bilaterians cell types in our cross-species comparison, they are associated with phylum-specific sensory and structural roles. Among these we find two peripheral sensory papillae cell types (expressing neuronal genes such as synaptotagmins and acetylcholine receptors Acha7-1) (clusters 15 and 25) positioned around the mouth and interspersed in the epidermis throughout the body (see marker gene Pxdn-953 in Figure 2d). We also recover two cell types (clusters 10 and 13) (clusters 10 and 13) that are associated with ciliary sense organs, mechanoreceptors distributed on the body of the animal that constitute their primary sensory system for swimming and predation, as indicated by expression of Tbx2 and Fmar-5 in these structures (Figure 2b,d)⁵³"

- While the study highlights TE methylation in chaetognaths, this pattern is also observed in vertebrates, nematodes, and annelids (e.g., <https://genomebiology.biomedcentral.com/articles/10.1186/s13059-024-03346-z>). Thus, the chaetognath DNA methylation pattern appears less exceptional in terms of TE methylation and more so in the loss of coupling to H3K36me3 and gene body methylation. However, given our limited knowledge of DNA methylation patterns in many other animal lineages, the broader context remains unclear. Similarly, the correlation between a lack of gene body methylation and trans-splicing, while intriguing, seems weak. In fact, there are compelling alternative hypothesis linking the coevolution of DNA methylation and DNA repair systems (<https://www.nature.com/articles/s41588-018-0061-8>), rather than gene regulation mechanisms. So, overall, I am not sure one can call this a “reversion to pre-animal DNA methylation”.

We appreciate the reviewer’s suggestion and agree that the phrasing “reversion to pre-animal DNA methylation” was maybe too strong. We have revised the text to indicate that chaetognaths have lost gene body methylation — a state that, based on current knowledge, likely represents the ancestral condition in animals — and have acquired TE-focused methylation.

Additionally, we searched the genomes of chaetognaths and ctenophores for ALKBH2/ALKBH3 orthologues, which are considered essential for DNA repair in the context of DNA methylation. Both lineages possess well-conserved ALKBH2 and ALKBH3 sequences, setting them apart from many nematodes. We found no major loss of DNA repair functions in the chaetognath lineage; hence, there is not a strong link between reduced DNA repair and the methylation pattern observed in *Paraspadella*. We now discuss both possibilities in the manuscript.

Finally, while the link between trans-splicing and gene body methylation is not very strong, we observe that in three independent lineages (chaetognaths, nematodes, and ctenophores) predominant trans-splicing is associated with the loss of gene body methylation. Both mechanisms seem to demarcate stably expressed genes (“housekeeping”), so it is definitively a possibility they might have similar functions. However, we now discuss this possibility in a more nuanced manner by also considering tunicates and hydrozoans, which exhibit both trans-splicing and gene body methylation, and suggest that further studies across diverse animal phyla may help clarify this potential relationship.

- The generated ATAC-seq data for adult and larval stages is underutilized in the paper, which represents a missed opportunity. What is the genomic distribution of cis-regulatory elements (CREs) in the Chaetognath genome? How frequent are non-promoter (intergenic or intronic) CREs and how distant from the nearest TSS? How do the enriched motifs in the CREs (representing putative transcription factor binding sites) compare to those known for bilaterian species? Generally these motifs are highly conserved for most TFs, but it would be exciting to check if this is the case also for Chateognaths.

We omitted some results from ATAC-seq analysis that did not seem highly insightful, but we are happy to provide some more details. We originally included a graph in the original manuscript showing the distance to TSS for various species (**Supplementary Fig. 8h** which is now **Extended Data Fig. 7i**). In our revision, we added another plot displaying the fraction of open chromatin regions (OCR) localised at TSS, proximal (<5kb), distal or intronic (**Extended Data Fig. 7h**). While we find that chaetognaths possess a lower fraction of non-TSS than other bilaterians, they nevertheless have >25% of distal OCR suggesting the presence of *bona fide* enhancers. As suggested, we investigated TFBS enrichment in these different OCR categories,

both as *de novo* and using previously characterised TFBS (presented in **Extended Data Fig. 7i** and **7j**). We found that chaetognaths show comparable TFBS enrichment to those of other bilaterians, instance bZIP binding sites appear located closer to the TSS than bHLH which are enriched in distal OCR. We also performed some *de novo* TFBS detection which detected TFBS similar to some known in other animals (Grainyhead GRHL1, Six3 in TSS and genebody) but also enigmatic motifs with limited similarity in more distal regions. This could indicate some level of lineage specific evolution and divergence at the transcription factor level. We added the following sentences in the results : “However, we identified distal regulatory regions and conserved TFBS binding properties with that of other animals (**Extended Data Fig. 7h-k**), suggesting at least partial conservation of their cis-regulatory logic.”

Additional comments

- The methods section would benefit from including additional QC metrics for the functional genomics assays. For example, report the percentage of trans-contacts and mappability for Hi-C experiments, FRiP scores and saturation levels (e.g., percentage of duplicates) for ATAC-seq datasets, and sequencing saturation metrics for scRNA-seq libraries.

In line with reviewer’s recommendations, we added a new panel (a) in **Extended Data Fig. 4** displaying the mapping statistics for Hi-C data as well as the percentage of cis interactions detected.

We also computed more statistics for ATAC-seq including mapping rates, duplication rates and FRiP that are presented in a new **Supp. Table 10**. Finally, we now specify in the methods that single cell libraries saturation is 74.2% for Adult2, 59.8% for Adult1 and 89.5% for juvenile (see below).

“Library saturation is 59.8% for Adult1, 74.2% for Adult2, and 89.5% for juvenile and read depth is 24,583 mean reads per cell for Adult1, 37,173 for Adult2 and 58,191 for juvenile.”

- Provide detailed information about the Hi-C experiments directly in the manuscript, rather than referring to a previously published study.

As request, we included additional information about the Hi-C procedure: *“Briefly, animals was crosslinked in 1% PFA, and chromatin subsequently extracted, immobilised on SPRI beads, washed and digested with DpnII as previously described (Rao et al., 2014). After end-labelling, proximity ligation was carried out using T4 DNA ligase and cross-linking reversed using Proteinase K, removed from the beads and the DNA fragments were purified again on SPRI beads. Sequencing library was constructed using the NEB Ultra library preparation kit (New England Biolabs)”*

- In panel 5b it would be better to show the “canonical” bilaterian protein structural domain architecture in comparison to the “simplified” version in *Paraspadella*.

We have now included the ancestral protein domain architectures for DNA methylation enzymes, yet due to space limitations, these can be seen in **Extended Data Fig. 10d**, as we

could not find a nice way to display so many gene families/architectures in a single panel effectively in the main figure.

- Show scRNA-seq stats in SFig 2c in log-scale.

We adjusted the Supplementary figure (**Extended Data Fig. 5c**) accordingly (see below)

- It would help if Paraspadella cell types in SFig. 6 were shown in the same order in each heatmap.

The Paraspadella cell types were shown in the same order across heatmaps (to the right in the y-axis, labelled Pg_X) but the labelling was confusing due to an error with the annotation of *D. melanogaster* now fixed. We apologise for the confusion and hope the revised figure reads better (now **Extended Data Fig. 5**).

Referee #2

Chaetognaths are an important but understudied phylum of marine invertebrate predators recently shown to be spiralian, closely related to rotifers. In the current manuscript, Piovani et al. combine a new chromosome-scale genome of the chaetognath *Paraspadella gotoi* with multi-omics analysis, including single-cell RNA-sequencing data and Methyl-seq analysis, to study the evolutionary origins of the chaetognath phylum. Due to their mesoplanktonic nature, obtaining sufficient high-quality DNA for comprehensive genomic analysis has been challenging. The authors should be commended for overcoming this difficulty and successfully generating a chromosome-level genome. Using these genomic resources, the authors demonstrate extensive gene family loss and the emergence of new gene families in the chaetognath lineage, with new gene families being recruited to novel cell types. They also reveal several interesting aspects of the chaetognath genome, including extensive tandem duplication without whole-genome duplication (WGD), the loss of centromere proteins, and methylation of transposable elements but not protein-coding genes.

Overall, the authors present a substantial and valuable dataset, including a chromosome-scale genomic resource for chaetognaths, which holds significant potential for advancing our understanding of the evolution of spiralian and gnathiferans. The manuscript is highly original and of broad interest to researchers in fields such as invertebrate zoology, genome structure evolution, cell type evolution, and methylation. However, the manuscript would benefit from a more coherent narrative to effectively connect the various datasets and provide a comprehensive and informative story. This enhancement would help the manuscript resonate more with readers and align with the standards of Nature. In addition to structural concerns, several other issues mainly related to phylogenetic interpretation, genome structure evolution, and gene family evolution are outlined below.

Major comments

1. The introduction would be more engaging for Nature readers if it emphasised the broader implications of the findings rather than detailing technical approaches. Highlighting how this study enhances our understanding of the evolution of chromosomal structures and gene regulation in early branching spiralian would make the introduction more impactful. Additionally, the manuscript would benefit from greater coherence among datasets. Establishing a clearer linkage, such as demonstrating how the genomic data, single-cell transcriptomics, and expression analyses collectively support key evolutionary insights, would strengthen the narrative. Currently, the manuscript reads as a collection of distinct data points that would gain from a stronger, unified storyline.

We appreciate the reviewer's concerns and we did our best to rephrase our introduction to emphasize the impact of genome regulation and genome organisation on body plan evolution, particularly in early branching spiralian. Particularly, we now added "*Understanding the genomic events which led to the emergence of such an intriguing body plan represents therefore a key challenge to understanding the origin of animals itself. Recent studies have showed how genomic and regulatory changes have sometimes (e.g. vertebrates^{24,25}, annelids⁷) but not always (e.g. echinoderms²⁶) accompanied the emergence of novel bodyplans.*"

2. Despite the last author's expertise in spiralian phylogeny, there are significant phylogenetic inaccuracies that need to be addressed. The manuscript shows fundamental phylogenetic issues and a lack of comprehensive understanding, which raises concerns about the robustness of the gene family gain and loss analysis—especially as a reliable phylogenetic tree is essential for meaningful interpretation. One example is Fig. 1b, where *P. australis* and *L. anatina* (not *L. anaitida*) are shown as sister groups, with this combined group depicted as the sister group to *B. neritina*, which has not been reported in the literature. Additionally, errors such as labelling Lophophorata as Platyhelminthes require correction. The inconsistencies between Fig. 1b and Fig. 4a should also be resolved to maintain coherence throughout the manuscript and improve the reliability of the analyses.

We apologise for labelling mistakes in some of the panels, and we agree with the reviewer that the topology in which bryozoans (*B. neritina*) and phoronids (*P. australia*) are grouped together in regard to brachiopods (*L. anatina*) would indeed be more consistent with previous papers on spiralian phylogeny (Laumer et al., 2019; Marlétaz et al., 2019). We therefore ran again our gene family analysis (inference of gene family gain, losses and also duplication using ML generax reconciliation) using such a revised topology and we show that it has nearly no impact on the outcome in terms of gene family gain, loss and duplication at deeper nodes. **Figure 1b** now features the same topology as **Figure 1a** and revised numbers as well as **Extended Data Figure 3d and 3e**.

Incidentally, to evaluate the impact of spiralian relationships on our reconstructions (also respond to reviewer #3), we also perform a similar gene family analysis using an alternative topology presenting a deeper branching for platyhelminths and bryozoans (the so-called 'Rouphozoa' topology in Marlétaz *et al.* 2019, also consistent with results of Laumer *et al.* 2019). We observed that this alternative topology does not change the main conclusion highlighted in this study, namely the loss of genes at the origin of Gnathifera and the burst of novelty and duplications in the chaetognath lineage. We included a **Supplementary Fig. 1** to highlight the results of this.

Finally, we also apologised for the mislabelling of platyhelminthes and lophophorates in **Figure 1** that has been rectified, we also attempted to add more labels.

3. Phylogenetic terms such as "Spiralia" and "Lophotrochozoa" should be clearly defined in the text, and more precise labelling in Fig. 1b would enhance reader comprehension. While the manuscript provides extensive discussion on the rotifer genome, placing greater emphasis on the chaetognath-specific data from this study would create a more balanced narrative and add depth to the analysis. Given that rotifers exhibit rapid chromosomal evolution, the availability of a chaetognath genome presents an excellent opportunity to reconstruct gnathiferan genome evolution, as their synteny is relatively more tractable. This point will be elaborated upon in a later comment.

We agree with the review that a precise phylogenetic nomenclature is crucial and that Spiralia and Lophotrochozoa have sometimes been used interchangeably. Here, we refer to Spiralia as the superclade encompassing Gnathifera and Lophotrochozoa (the descendants of the last common ancestor of molluscs and annelids in the topology of **Figure 1b**). We added labels in **Figure 1b** and rephrase a sentence in the introduction: "*After years of sometimes contradictory molecular analyses^{10,18,19}, chaetognaths have been assigned to the clade Gnathifera which represents the sister-group to the rest of lophotrochozoans animals within spirilians^{5,6} (Figure 1).*"

Our original intention was to avoid decrease the legibility of **Figure 1b** by overcrowding it with too many labels and texts, in this revised version, we added some key labels, and also as also suggested by reviewer #3 included the full species name to increase clarity and legibility.

Regarding rotifer, our intention was not to discuss their genomic evolution, but rather to compare chaetognaths with them, as the only other representative of the gnathiferan lineage for which good quality genomic data is available so far. Notably, we rephrased the following sentence: "*To characterise genomic evolution in the gnathiferan lineage, we compared our chaetognath genome and transcriptomes with data available in other spirilians*" to decrease the emphasis on rotifers as suggested by the reviewer.

4. In the gene family analysis, more attention should be given to the top GO terms related to biological function, such as insulin-like receptors and SMAD proteins. These pathways are significant as insulin signalling controls body size, and SMAD proteins play roles in TGF-beta signalling. The authors should carefully annotate which components of these pathways are missing, particularly for SMAD proteins, which have distinct roles (e.g., receptor-regulated SMADs such as Smad1/2/3/5/8, co-SMADs like Smad4, and inhibitory SMADs such as Smad6/7). Noting that Smad2/3 mediates TGF- β /Nodal signalling and Smad1/5 is involved in BMP signalling would provide insights into how these pathways evolved in chaetognaths.

We thank the reviewer for suggesting to pay a closer look at the genes related to enriched functional terms. By doing that, we actually realised that for both these terms, the two noted enrichments were not really pertinent and were triggered by the approach used to perform a GO analysis on lost genes, which was actually to use genes belonging to lost gene families in a sister

species known not to have undergone extensive gene losses (in this case, we used the GO annotation of *Branchiostoma floridae* to do that). However, while examining specific genes associated with these terms, we found several lineage-specific expansion in amphioxus of genes associated with both SMAD and insulin which triggered this significant association. This was notably the case for CILP gene (cartilage intermediate layer protein, OG_5509) which in vertebrates has an interplay with both SMAD and insulin.

We therefore decided to revise this analysis for increased robustness to such effects and this time performed a GO enrichment based on a set of non-redundant GO terms assigned to each reconstructed gene family. We recovered a clear signal for the loss of centromeric assembly factors, and some new alternative terms (neuropeptide signalling and response to exogenous dsRNA). We now list the corresponding gene families as well genes in mouse and drosophila associated with these terms in **Supplementary Table 4**. We therefore also replaced the panel **Figure 1c** with the one below.

We specified in the method section : “For GO enrichment associated with family losses, we assigned non-redundant GO terms for each family from the emapper annotation obtained for selected species specified in **Supplementary Table 2**.”

5. For the loss of CenH3 genes, which ranked among the top findings, the authors should clarify the process used to confirm the true absence of these genes. Absence in gene annotations alone is not sufficient; additional evidence such as BLAST searches of the genome sequence should be provided. Ideally, identifying microsyntenic regions where these genes are found in closely related species and confirming their absence in the chaetognath genome would strengthen this conclusion.

The absence of CenH3 was first recovered by examining the results of our GO enrichment for gene family analysis, which included multiple genomes and transcriptomes (4 chaetognaths and 2 rotifers). We confirmed that the absence of CenH3 was not related to a gene family reconstruction artefact by checking the gene family assignment of the best blast hit of the urchin CenH3 against these different proteomes, for which we recovered other histones H3 (H3.2) as best hits and no other histone-related gene or CenH3-like gene. We then performed a tblastn against the genomic sequence of chaetognath and rotifer to confirm that the gene had not been missed by the gene prediction and confirmed its absence. We added the corresponding statement in the method: “Gene gains and losses were inferred following parsimony principles and using the ete3 python library to process phylogenetic trees. For relevant lost families, we assessed whether the best blastp match in relevant species belonged to another gene family to disclose potential gene fragmentation related to evolutionary acceleration. Gene absence was also corroborated by tblastn against the genome of *P. gotoi*.”

6. One advantage of having a chromosome-scale genome is the opportunity to explore genome structure evolution through synteny analysis. It is notable that this aspect has not been fully explored in the manuscript. Simakov et al. (2022) proposed an ancestral spiralian karyotype with 20 linkage groups based on data from molluscs and annelids (all sharing H⊗Q, J2⊗L, K⊗O2, and O1⊗R fusion events). This study could provide the first evidence confirming these findings in a more early-branching spiralian lineage. The authors are encouraged to discuss how

chaetognaths share spiralian fusion-with-mixing events, expanding on this section to make the findings more accessible.

We completely agree with the reviewer about this being a highly relevant point, and our initial manuscript was maybe too succinct in describing spiralian specific fusions. However, given the fact that chaetognaths experienced a number of lineage-specific fusions and have a somehow derived karyotype, we wanted to stay cautious. We revised our text and included some extra panels to make it more clear what we think can be and cannot be inferred. Particularly, we have unfortunately not been able to detect BLG L and Q in the chaetognath using dotplot comparisons and fisher enrichment tests with 3 'focal' genomes: the seastar (*A. rubens*), the amphioxus (*B. floridae*) and the pecten (*P. maximus*).

We also added the comparison between *Paraspadella* and amphioxus as **Extended Data Fig. 3c**, particularly as it displays evidence for the presence of $K \otimes O2$. We also (see below) replaced the comparison between *Paraspadella* and *Pecten* in **Figure 1e** to help visualise the fusion events. We also examine the synteny of the contiguous but non-chromosome scale rotifer *Brachionus manjacacas* (**Extended Data Fig. 3d**) and show that they also experienced extensive fusions.

- From comparison with seastar, Q (ARU20) is not statistically detected. BFL3 ($Q \otimes C2$) matches PG09 but this might be due to the presence of C2 which is also detected in the comparison with pecten. So, $H \otimes Q$ cannot be detected in chaetognaths.
- Presence of $O1 \otimes R$ is likely from *Asterias* (on chr2)
- $K \otimes O2$ similarly is there from comparison with amphioxus notably.
- $J2 \otimes L$: L not detected ($J2 \otimes L$ from *Pecten* matching chr6 but no L detected there from amphioxus or *Asterias*).

To clarify, we now state in the manuscript:

“Among the 4 fusions-with-mixing previously reported in spiralian, namely $H \otimes Q$, $J2 \otimes L$, $K \otimes O2$, and $O1 \otimes R$ (Lewin et al., 2024; Martín-Zamora et al., 2023; Simakov et al., 2022), we identified the presence of $K \otimes O2$ and $O1 \otimes R$ in chaetognath, therefore indicating that they are likely shared by all lophotrochozoans. We were not able to detect Q and L which could suggest either that the fusions $J2 \otimes L$ and $H \otimes Q$ took place after the divergence between gnathiferans and spiralian (particularly as we detected H and J2 alone), or simply that there is not enough syntenic signal to detect them.”

7. It appears that the authors used a colour scheme for ancestral linkage groups similar to that in Martín-Zamora et al. (2023), but Fig. 1e in the current manuscript does not match Fig. 1d of Martín-Zamora et al. Clarifying whether different versions of the *Pecten* genome were used would be helpful. The authors should ensure comparability across studies by using the same genome assembly version. Notable differences include:

- *Pecten* Chr 1 ($H \otimes Q$) appears as Chr 18 in Fig. 1e, showing fission into two chromosomes.
- *Pecten* Chr 4 ($J2 \otimes L$) corresponds to Chr 16 in Fig. 1e and stays intact with fusion-with-mixing.
- *Pecten* Chr 5 ($K \otimes O2$) corresponds to Chr 19 in Fig. 1e, showing fission followed by fusion-with-mixing.
- *Pecten* Chr 10 ($O1 \otimes R$) corresponds to Chr 3 in Fig. 1e, remaining intact with fusion-with-mixing.

Discussing these findings and comparing them with other studies on genome architecture would emphasise their significance and better integrate them into the broader context of spiralian genomics. Additionally, consider moving Fig. 3d to accompany Fig. 1e for improved visual coherence.

We apologise for the confusion across different versions of the *Pecten* genome, notably we used in the original version of this manuscript the annotation provided by (Kenny et al., 2020). As we now realise this introduces unfortunate discrepancies with other publications and we thus revised our analyses using the latest deposited *P. maximus* genome (xPecMax1.1: GCF_902652985.1) and the annotation of NCBI which appears in line with (Martín-Zamora et al., 2023). We now present the update comparison in panel **Extended Data Figure 3b and d** and also introduced the comparison with the amphioxus *B. floridae* as **panel 3c**.

8. Following the previous comment, the synteny analysis raises several points that need clarification. The comparison between *Paraspadella* and *Pecten* appears significantly different when viewed in the ribbon plot (Fig. 1e) compared to the dot plot (Supplementary Fig. 3b). It would be beneficial to clarify what gene sets are represented in each plot and the criteria used for their selection. The assumption might be that the genes shown in the ribbon plot represent the subset of orthologues with significant enrichment in Fisher's exact test (coloured genes in Supplementary Fig. 3b), but this does not seem to hold true. For instance, ALG Q is missing in the dot plot but appears prominently in the ribbon plot (e.g., on chromosomes 9 and s1). The main text mentions that ALGs A2, B2, L, N, and Q were not detected in *Paraspadella*, yet these are visible on specific chromosomes in Fig. 1e (e.g., A2 on chromosome 1, B2 on chromosome 4, and Q on chromosomes 9 and s1). Clarification is needed on why these ALGs appear in the figure if they are not statistically enriched, and vice versa. Furthermore, while the text states that the spiralian fusion-with-mixing $H \otimes Q$ is not detected, it seems present on chromosomes 9 and s1 in Fig. 1e. An additional consideration is that a fourth fusion ($O1 \otimes R$) has been identified in other spiralian (Simakov et al. 2022). Determining whether this fusion is present in chaetognaths and discussing its relevance would provide further insight into the evolutionary context of the data.

We agree that the representation in **Figure 1e** could give the impression that the $H \otimes Q$ fusion is present and could be seen as misleading. This was caused by the fact that we displayed as links 1:1 orthologues between *Pecten* and *Paraspadella* and the links were colored according to the ALG annotation in *Pecten*, even if *Paraspadella* had no significant representation of the ALG. We decided to modify this panel and use amphioxus as the outgroup as it doesn't have the putative spiralian-specific fusions in order to better display the fusions and avoid any confusion. Below is the new panel **Figure 1e**.

9. The integration of the single-cell atlas and Hox gene expression data could be expanded to emphasise evolutionary implications. Specifying which cell types express Hox genes and discussing their relevance to spiralian evolution would help readers understand how lineage-specific traits are reflected in single-cell data.

We thank the reviewer for their comment and have now included a more thorough description of Hox gene expression in cell types reflecting both scRNA-seq data and ISH (see **Extended Data Fig. 9b** and below text). We find that Hox genes in our scRNA-seq data show expression in more than one cluster ranging from epidermal cell types, neurons and peripheral papillae sensory neurons (for specifics see **EDF 9b**). This is expected as Hox genes specify larger domains often encompassing different tissues (and cell types) for this reason ISH is a better tool to localise their specific expression. In fact, ISH reveals that anterior Hox show a broad expression in the nervous system, while median and posterior appear staggered similarly to what is observed in other lineages. Interestingly MedPost appears expressed in the papillae peripheral sensory neurons which is a chaetognath's specific cell type. We have now changed the text as below to better reflect these observations.

*“Hox expression in our single cell sequencing datasets shows expression in several cell types, mainly epidermal, neuronal and papillae cell types (**Extended Data Fig. 9b**). To further localise their expression we performed in situ hybridisation on six Hox genes in *P. gotoi* juveniles (**Figure 4b,c** and **Supplementary Table 7**). While anterior Hox genes show a broad expression in the chaetognath nervous system (paired ventral and cephalic ganglions for Hox1 and Hox4), median and posterior Hox genes show a more classical staggered expression in this ventral ganglion that appears coherent with paralogy group (MedN3 more anterior than MedN4, and PostN4 in the posterior-most part of the ganglion)²¹ (**Extended Data Fig. 9c**). PostN4 presents a dual domain, both in the posterior-most part of the ventral ganglion and in the postanal region of the juvenile (**Figure 4b**). Surprisingly, MedPost is localised in papillae peripheral sensory neurons interspersed in the epidermis (**Figure 4b, Extended Data Fig. 9b**). The expression of MedPost in sensory neurons throughout the body is different from that observed in rotifers where it is*

expressed specifically in the postanal components of the nervous system (Extended Data Fig. 9c). In chaetognaths, posterior Hox genes are expressed in this region; these genes have been lost in rotifers²⁰. The cell-type specific expression of MedPost suggests that acquisition of new expression domains could have participated in the emergence of novelties at the bodyplan or cell type level”

We therefore postulate that from an evolutionary standpoint Hox genes have a dual role as seen in other animals (vertebrates, butterfly): an ancestral expression in AP patterning, and newly acquired domains underlying the evolution of novel structures, possibly at the sensory and cell type level (likewise for instance to the cerebral structures in vertebrates) (see below).

“Similarly to what has been observed in other lineages, tandemly duplicated Hox genes acquired cell-type specific expression domains, while others maintained their broad patterning roles (Figure 4)⁸¹”

10. Including a schematic illustration of Hox gene expression in Fig. 4 would be beneficial, particularly to highlight any spatial collinearity similar to that seen in other species. This could be presented as a cartoon illustration comparing expression collinearity with rotifers and other outgroups, such as annelids and molluscs, and incorporating data from additional spiralian species for broader context would add value. Such comparisons could align well with the Hox gene expression patterns shown in the single-cell dataset. In addition, arranging the in situ data so that the anterior of the worm is oriented to the left would ensure consistency with Fig. 2d and other bilaterian studies.

We have implemented all the reviewer suggestions and have added the following: i) a schematic illustration of Hox gene expression in **Figure 4**, ii) collinearity schematics in figure **EDF9** following the orthology groups from **Supplementary Figure 3** iii) have turned the worms so they now all have their heads pointing leftwards. We generally revised **Figure 4** for clarity to also match reviewer #4 comments.

11. The manuscript in its current form lacks a clear connection between lineage-specific genes and novel cell types. Strengthening this aspect would help underline how these new gene families contribute to the development of unique chaetognath cell types.

We tried to emphasise the importance of expression of lineage-specific genes such as duplicated or newly evolved genes in cell-types that appear specific to chaetognaths. Particularly we revised the last paragraph of the section previously entitled ‘*Rapid gene complement turnover (without WGD)*’ that we renamed ‘*Gene complement turnover contributes to cell-type innovation*’ to highlight the connection suggested by the reviewer. We added this sentence in the last paragraph:

“We therefore hypothesise that after extensive gene loss in the gnathiferan lineage, particularly affecting gene families involved in sensory and nervous functions, the evolution of cell types in chaetognaths relied on newly evolved genes as well as lineage-specific tandemly duplicated genes. This reshuffling of the gene repertoire therefore likely played an instrumental role in the establishment of their distinctive body plan.”

12. The identification of the MedPost gene positioned between central and posterior Hox genes is intriguing, though not unexpected, as its close relationship to these Hox groups has been noted in transcriptomic studies. However, the manuscript leaves the identity of central Hox genes largely unresolved. Given that the authors successfully recovered the Hox cluster with their high-quality genome assembly, including a gene tree (and potentially trees for all Hox genes) would strengthen the study and provide valuable insights into Hox gene evolution in this lineage. Determining whether MedPost is more closely related to central or posterior Hox genes would be particularly informative. Additionally, the methods section should detail how Hox genes were identified and verified, even if the process is straightforward. Combining phylogenetic analysis with motif analysis would be especially useful for identifying certain Hox genes that possess lineage-specific diagnostic peptides, such as the PG6 motif (KS(I/L)ND in gnathiferans) adjacent to the homeodomain. Given the availability of high-quality Hi-C data, it would also be informative to investigate whether there is a topologically associating domain (TAD) correlated with the unique structure of the Hox cluster.

We followed the reviewer suggestion, and performed a focused phylogenetic analysis of Hox genes, which is included as **Supplementary Figure 3**. We based our identification of Hox genes on previously characterised chaetognath Hox, particularly in Papillon *et al.* 2003 and Matus *et al.* 2006. We found that some of the median Hox genes could be assigned to a broader paralogy group: MedN1 appears to be related to Hox5 (not unexpectedly), MedN2 appears related to Lox5, which is further confirmed by the presence of a divergent but identifiable Lox5 peptide as defined by (de Rosa *et al.*, 1999), MedN3 is less clearly positioned, but the genes MedN4,5,6 are clustered in the tree and thus appear to represent lineage specific duplicates. Interestingly, our tree suggests a possible affinity of Lox2/Lox4 to these genes (with mild support), which could indicate that these median genes were independently expanded in chaetognaths (MedN4,5,6), spiralian (Lox2/4) and even ecdysozoans (Ubx/Abd-A). Position of MedPost genes remains elusive, as they are generally fast evolving and based on diagnostic positions rather than phylogeny. We edited **Figure 4** to add subsequent details about the paralogy assignment of identified chaetognath Hox genes and also slightly edited the text to recognise that median expansion mostly affect PG8.

We added to the methods: “Hox genes were annotated based on similarity search toward previously annotated chaetognath Hox genes (Matus *et al.*, 2007; Papillon *et al.*, 2003), inspection of diagnostic residues and phylogenetic analysis of selected bilaterian homeodomain genes (**Supplementary Figure 3**) using IQ-TREE with a LG4X model and 1000 ultra-fast bootstraps.”

A contact map of the Hox region is included as **Extended Data Fig. 4c** in which we do not observe chromatin domains or boundaries within the Hox cluster.

13. The proposed order of evolutionary events—suggesting that the gnathiferan lineage underwent genomic simplification involving gene loss and disruption of ancestral linkages, followed by the mobilisation of lineage-specific duplicated and novel genes—could be elaborated. Alternative explanations, such as simultaneous gene loss and gain or initial gene

gain followed by loss, should be briefly discussed. The authors should specify the strongest evidence for their proposed order and consider precedents in other lineages.

Regarding gene gains and loss, our gene family reconstruction makes it possible to place such event on a tree, and even if there is always a possibility of simultaneous convergent loss, Regarding the accelerated genome evolution and fusions, we agree that such events could have taken place convergently as we could not identified shared fusions between chaetognath and rotifers. We adjusted the discussion to consider the possibility of convergent events: *“To account for this unusual pattern, we propose that the gnathiferan lineage underwent an accelerated genomic evolution involving a loss of genes as well as a disruption of ancestral linkages, either as a single event or as multiple lineage specific events. Such loss of genomic traits could plausibly have been coupled with morphological and possibly body size reduction from their progenitors, even if uncertainty remains regarding the complexity of the last bilaterian ancestor^{82,83}”*

14. Fig. 3f is not referred to in the main text, and its figure legend lacks clarity on the conclusions being drawn. Including a reference in the main text and clarifying its relevance would improve the manuscript.

We added this panel in a late version of the figure as we felt it was quite striking, and we forget to reference it, we now referenced it where it should have been : *“Interestingly, such an incorporation of phylum-specific genes among cell-type markers is only observed in chaetognaths compared to five other species investigated in which most marker genes are either ancient or species-specific (Figure 3f and EDF 7c)”*

Minor comments

Text (the authors are encouraged to use continuous line numbers):

Page 2 Line 23 – Typo: "ressources" should be "resources."

Corrected

Page 2 Line 48 and throughout – Supplementary subfigures are referred to out of order, which can be distracting. Consider reordering figure panels or adjusting the text references for consistency.

Overall we adjusted the figure to reflect the order of citation when possible.

Page 4 Line 27 – Define the acronym "TAD" when it is first mentioned.

We added *“ topologically associating domain (TAD)”*

Page 6 Line 8 – The phrase "as well as could be" is awkward. Consider rephrasing for clarity.

We rephrased as *“From their shared neuronal marker Cac1a-2 expression, they are located both in the ventral and cephalic ganglion and as well as in some peripheral neurons, an abundant cell type in the chaetognath nervous system⁴⁷ (Figure 2d).”*

Page 6 Line 40 – Add a brief explanation of what chaete cells are for readers unfamiliar with the term.

We rephrased and swapped the order of the sentence to improve clarity:

“The expression of chitin synthase (Chs2-5) near chaetognath grasping spines indicates these are likely ensuring the growth and synthesis of the grasping spines (cluster 18) while the marker gene Vwf-2 (clusters 2,6 and 17) and lfea-2 (clusters 8,20) show expression in the epidermal layer (**Figure 2d**). The presence of multiple epidermal cell types is consistent with the presence of a complex pluristratified epidermis in chaetognaths, a condition rarely found outside of vertebrates⁵⁴. All of these structural cell-types express various chitin synthase and chitinases, with alternative paralogues present in the ‘chaete’ cells (clusters 18, Chs2-5, Chi10-1, Chi10-3 and Chi10-4) and epidermal cells (clusters 2,6 and 17; Chs2-3, Chit1-7, Chit1-8).”

Page 7 Line 23 – References to main text panels appear out of order. Reorganise for a smoother reading experience.

We adjust the order of panels to improve legibility.

Page 8 Line 9 – Supplementary Fig. 8 is referred to before Fig. 7. Adjust the order for clarity.

We assume that the order of figures .

Page 8 Line 39 – The sentence "ciliary cells appear among the ones with the most recent transcriptome" is unclear. Add context to explain what phylostratigraphic gene ages mean and what this result implies.

We rephrased as follow : “We estimated the phylostratigraphic age of gene sets specific to each cell types by leveraging gene phylogenetic age and we noticed that ciliary cells appear among the ones with the most recent profile”

Page 10 Line 28 – The phrase "suffered duplications" implies a negative connotation. Use a more neutral term, such as "experienced duplications."

Accordingly, we replaced by “underwent gene duplications”

Page 12 – There are several references to non-existent figures (e.g., 4l, 4k) and incorrect mentions of Figure 4, which is the Hox figure. Correct these references.

Thanks for pointing out this discrepancy, we relabeled to the corresponding panels h and i, and reorder the panel in the figure to improve legibility

Page 15 Line 35 – Define "dph" when first used.

We defined as “ days post hatchling (dph)”

Main figures:

Fig. 1c – Clarify if "neg. reg." is "negatively regulated." If not, please describe it.

This GO term is not appearing anymore.

Fig. 1d – Explicitly state what the grey boxes represent. Also, correct the order of references in the legend (e.g., e and d).

This GO term is not appearing anymore. We specified “centromeres inferred from HiC data (highlighted as grey box).”

Fig. 1e – Ensure consistent labelling, such as using "Chr s1" uniformly. Additionally, clarify the rationale behind naming it "s1," as this is not well explained.

We updated the Figure 1 with a new version using more homogeneous labelling, s1 does not appear anymore.

Fig. 1e and throughout – Correct the typo "Adinetta" to "Adineta."

Corrected

Fig. 1e legend – Consider stating explicitly that these are bilaterian linkage groups.

We added for clarity : “to their bilaterian ancestral linkage group as defined by a comparison with amphioxus.”

Fig. 1f legend (Line 18) – The reference to "Supplementary Figure 3" should be "Supplementary Figure 4." Terms like "A/B compartmentalisation" and "trans-splicing" should be defined when first used to prevent readers from needing to search for definitions.

We corrected the reference to the figure, thanks for pointing this out!

Regarding A/B compartment, it is difficult to define but to improve clarity we added “large-scale A/B compartments and local topological associated domains” and also added a citation to Lieberman-Aiden *et al.* 2009 which first described these structures.

Fig. 2a legend – Mention explicitly that the data includes both hatchling and adult cells.

We added “from pooled juvenile and adult libraries” for clarity as suggested by the reviewer (see response to reviewer #4).

Fig. 2d – Add cell type labels directly to the in situ images for better panel clarity.

We added the cluster number to each figure.

Fig. 4a – Label species names to provide more context on Hox cluster organisation. Including the Hox cluster organisation in the Ecdysozoa outgroup would further enrich the analysis.

Provided there is an extensive literature on the topic, and some other reviewers complain about how bulky this panel was, we could rely on the classical classification of Hox paralogy groups in bilaterians as defined by (Gąsiorowski *et al.*, 2023).

Supplementary figures:

Supplementary Fig. 1 – Italicize *P. gotoi*.

Done

Supplementary Fig. 2 – Highlight chaetognaths in plots a, d, and e for easier identification. Clarify what the numbers in plots d and e represent. Correct species names (e.g., *L. anatina* instead of *L. anatida*, *P. australis* instead of *P. austratlis*). Use the correct formatting for *Fungia* sp. instead of *Fungia* ssp. (do not italicise "sp.").

We corrected species names and added some shading and bold to highlight chaetognaths and gnathiferans in the phylogenetic trees (consistently with Figure 1).

Supplementary Fig. 3 – Italicise genus names throughout and ensure consistent terminology when labelling *Pecten* as either scaffold or chromosome (note that the correct version of the *Pecten* genome should be used).

We used a more recent *Pecten* genome version and italicised species names.

Supplementary Fig. 4 – The caption should say "Supplementary Figure 4," not "Supplementary 4." Add more detail to part a to help readers unfamiliar with the figure's significance.

We updated Supp. Figure 4 in line with the comments of reviewer #1, we added “Mapping statistics of the Hi-C library highlighting the number of mapped and inferred contacts used to generate contacts maps including those in cis-” to better explain the processing of the Hi-C data.

Supplementary Fig. 5 – Consider ordering the plots by a useful criterion (e.g., adult-specific ones grouped together) rather than alphabetically.

We have ordered them by increasing adult-specific percentage.

Supplementary Fig. 6 – Add labels and units to the legend in the plot.

Done

Supplementary Fig. 7 – Correct the typo "infered" to "inferred."

Done

Supplementary Fig. 8 – The term "multiplicity" is somewhat unclear; consider rephrasing or elaborating on its meaning for better understanding.

We rephrased as follow: "Number of paralogous genes (multiplicity) derived from each chaetognath-specific duplication."

Methods:

Clarify which type of Hi-C technology was used.

We added the following details regarding Hi-C methodology: "A proximity ligation library was constructed from ~20 adults collected at the same locality near Amakusa. Briefly, animals were crosslinked in 1% PFA, and chromatin subsequently extracted, immobilised on SPRI beads, washed and digested with DpnII as previously described (Rao et al., 2014). After end-labelling, proximity ligation was carried out using T4 DNA ligase and cross-linking reversed using Proteinase K, removed from the beads and the DNA fragments were purified again on SPRI beads. Sequencing library was constructed using the NEB Ultra library preparation kit (New England Biolabs) and sequenced on a NovaSeq6000 instrument."

Specify the read depth per cell for the single-cell data.

We added in the methods: "read depth is 24,583 mean reads per cell for Adult1, 37,173 for Adult2 and 58,191 for juvenile"

Correct "SAMAP" to "SAMap."

Done

Remove references to the acoel worm *H. miamia* if the data is not included, and note that acoels are not flatworms.

Apologies, we did not use in the final version of the comparisons, reference to acoel have been indeed removed.

Referee #3:

I enjoyed reading the manuscript by Piovani and collaborators, which provides novel insights into the origin and evolution of the phylum Chaetognatha and a first look into the evolutionary patterns underlying the diversification of the Gnathifera. Chaetognaths are striking beasts that present a unique body plan, and this study looks into genomic, regulatory, and cellular aspects to understand their emergence and distinctive features. The results point to accelerated genomic evolution in the gnathiferans and chaetognaths, with a mix of novel patterns but also gene loss, although it is still early days for gnathiferans as their sampling is still scarce. I believe the datasets generated and the analyses done reflect the state-of-the-art in the field and provide

original and novel insights on the evolution of the clade, I'd like to congratulate the authors for the enormous effort to put together these datasets given how hard it is to work with these organisms.

I believe there are no major issues with the data, analyses or conclusions, and I can only happily recommend the publication of this manuscript. I do have some minor questions and suggestions, mostly aimed at making the manuscript to the general audience of a journal such as Nature.

We are really grateful for the reviewer's enthusiasm and appreciation for our work!

1) I noticed the authors did not refer to or compare their results against the two recent papers from Prof Timothy G Barraclough lab (Oxford) on rotifer genomes (Wilson et al Trends in Genetics 2024; Nowell et al Nat Comms 2024). I understand this rotifer genome data came out very recently and I would not expect the authors to add it to their analyses. However, I'd like to encourage the authors to incorporate the findings of these papers in their discussion section.

We thank the reviewer for pointing out these very interesting studies, unless we missed something, they don't appear to introduce new rotifer genes, but transcriptomic analysis. We added the following in the discussion: "Interestingly, rotifers also acquired new genes albeit through the distinct process of horizontal gene transfer which were involved in the evolution of new traits regarding resistance to desiccation or ionizing radiation (Nowell et al., 2024; Wilson et al., 2024)"

2) Annotation of gene loss: the methods mention eggnoG mapper and goatools, but I couldn't find which species was used to annotate lost genes. Given that these are absent in chaetognaths/gnathiferans, I think the text should clarify how these were identified.

Our original approach was to perform the enrichment by using the functional annotation of lost gene families in amphioxus, as this species is known to have undergone limited gene loss from the ancestral bilaterian gene complement. However as explained above, we realised that this procedure was prone to artefacts in case of lineage specific expansion, and we decided to revise this analysis using non-redundant GO annotation for each family based on the curation of eggnoG annotation for multiple species. See revised **Figure 1c**. We now added in the methods: "For GO enrichment associated with family losses, we assigned non-redundant GO terms for each family from the emapper annotation obtained for selected species specified in Supplementary table 2 and computed enrichment at the family level."

3) I think the finding on CENP genes being lost in gnathiferans is interesting. That said, I'd like to ask why there is no more elaboration on the role of regulators of the insulin-like receptor signalling pathway and SMAD protein signal transduction, given the biological function of these genes in development and the fact that more gene families have been lost for these categories than any others.

As stated for point 2), we revised these analyses by tweaking the GO enrichment calculation, which excluded the SMAD and insulin-like (both were related to an expansion of CILP gene in amphioxus). We however now observed among lost genes the neuropeptide signalling and processing of exogenous dsRNA. A detailed list of corresponding genes in *Drosophila* and mouse is provided in **Supplementary Table 10**.

4) Fig 1b, please review clades labelling, Brachiozoa/Phoronozoa are mistakenly labelled as Platyhelminthes, and the flatworms branch has no label.

We apologised for this mistake in labelling the tree in **Figure 1b**, this has now been adjusted.

5) Fig 1b and across the manuscript, I find that the use of species abbreviations gets in the way of accessibility. For example, Figure 2c is hard to follow unless you are familiar with the species

of invertebrates for which single-cell data is available. At least the first time these species are named, the full genus name should be used, and Fig 1b seems the perfect place for this.

We followed the reviewer's suggestion and listed the full species name in **Figure 1b**, we also added more labels for main clades discussed in the text. Regarding **Figure 2c**, we also included the colloquial name below the species name to improve clarity and refer in the legend to **Figure 1b** for full species name.

6) Fig 1e, Adinetta has only one letter t.

Corrected

7) Figure 3 could use some more/better labelling. For example, Fig 3a could indicate what is the meaning of the numerical value indicated in each of the curves. Fig 3d, what is the meaning of the ball size and the grey colour?

Thanks for the suggestion, we added a caption for the number of paralogues per duplicated family (“#paralogue pairs per family”) in panel c (previously a) and we added a legend indicating that the bubble size corresponds to the number of shared orthologues between chromosomes. We also added in the legend: “Each bubble represents the number of shared orthologues between a pair of chromosomes, which is displayed in color (see Figure 1e) if the enrichment is significant (fisher test) and otherwise in grey.”

8) Figure 3b, I'd like the manuscript to elaborate on why the matrix is not symmetrical. I understand the matrix represents the number of paralog pairs shared between chromosomes, and this should be the same value no matter in which direction the comparison is made (e.g., like in the example given for WGD).

The matrix is not symmetrical because we chose to count each gene once among pairs of paralogues, and there are many cases when there are more than 2 paralogues for a given gene that originated in chaetognaths. We added : “Each duplicated gene was only counted in one paralogue pair to avoid giving excess weight to large paralogue families, hence, the matrix is asymmetrical.”

9) Could the authors expand on what's going on with chromosome 5? It seems like a very “hot” chromosome!

The chromosome 5 is indeed a very dynamic chromosome with a very high rate of tandem gene duplications. Interestingly, our Hi-C analysis shows that while most chaetognath chromosomes do not show A/B compartment, the chromosome 5 exhibits a typical checkerboard pattern that is consistent with A/B type of 3D organisation.

10) Fig 3f, is this truly “phylostratigraphic” distribution (as in one way BLAST from one single anchor species against one representative for each of any other lineages) or just good old phylogenetic bracketing?

We used GenEra to infer phylostratigraphy for *P. gotoi* which is implementing a ‘canonical’ phylostratigraphic approach and not inferred age of a gene family which could have residual homology with one another (what the reviewer refers as ‘phylogenetic bracketing’ as far as we understood).

11) Page 8, lines 6-7 and sentence in line 8. The duplicate multiplicity or the multiple duplicability, makes this hard to follow. I wonder if these sentences could be streamlined to make them more accessible.

We rephrased as follow: “We observe a single 4DTV peak for genes that belong to duplicated subfamilies showing distinct numbers of paralogues indicating that they likely originated in a single event”. We do not use the term ‘multiplicity’ anymore as we agree with that reviewer that it could be confusing. We also adjusted the label of the panel as ‘#paralogue pairs per family’.

12) Page 8, some recent papers have highlighted the possibility of orthology assignment artefacts caused by the fast-evolving nature of some lineages, such as chaetognaths. This would cause an excess of new genes, as well as of lost ones. This is a significant limitation in comparative genomics analyses that should be mentioned in this study. Please, see Natsidis P et al iScience (2021), Weisman et al PLoS Biol (2020), and Guijarro-Clarke et al Nature Ecology & Evolution (2020)

We are conscious of these limitations, and that’s why we attempted to rely on multiple distinct criteria, from ‘phylogenetic bracketing’ from gene family reconstruction, strict phylostratigraphy (‘GenEra’) but also Panther families to estimate gene losses. Interestingly, we also do not observe a corresponding burst of gene family novelties in other fast evolving lineages such as for instance platyhelminthes or rotifers that would be consistent with the such a gene family ‘detachment’ effect. We included references and rephrased to put emphasis on such as artefacts. This section now reads: “In *P. gotoi*, these families make up for 1,832 genes or 8.3% of all predicted genes; to ascertain these numbers, we further confirmed that 1,307 of these genes do not have detectable homology outside chaetognaths in a direct phylostratigraphy assignment, and we observed PFAM domain in only 397 of them (Supplementary Table 1). These distinct lines of evidence support that this observation is not a reconstruction artefact such as those pointed out by several recent studies (Guijarro-Clarke et al. 2020; Natsidis et al. 2021).”

13) Page 8, line 44, the sentence has a teleological whiff, please rephrase.

As Haldane once said “Teleology is like a mistress to a biologist: he cannot live without her but he's unwilling to be seen with her in public.” Apologies for that.

We then rephrased: “We therefore hypothesise that after extensive gene loss in the gnathiferan lineage, particularly affecting gene families involved in sensory and nervous functions, the evolution of cell types in chaetognaths relied on these newly evolved genes. This reshuffling of the gene repertoire thus likely played an instrumental role in the establishment of the distinctive body plan of chaetognaths.”

14) Fig 4 could use a legend, for example, what are the grey boxes in the Hox genes figure?

We added this in the legend: “Distribution of Hox genes in lophotrochozoans assuming the phylogeny of (5) highlighting the expansion of different paralogy groups in distinct spiralian lineages. Anterior genes are coloured in shades of red/pink, medians in blue, medpost in yellow and posterior in green, each shade represent a paralogy group. Grey boxes indicate an unknown gene complement. A horizontal line crossing Hox genes indicates evidence for physical linkage.” More generally, the figure has been edited for clarity.

15) Page 10, line 38. I am no expert on methylation, but is the lack of gene body methylation really a “unique” case of reversion? How does this relate to genome size?

There are quite a lot of lineages that have lost DNA methylation, and consequently are not performing gene body methylation (e.g. dipterans, rotifers, some beetles, myxozoans). The chaetognath case is remarkable in the sense that they kept DNA methylation but do not use it for gene bodies and have repurposed this for transposable elements, and this correlates with a protein domain configuration simplification in the major enzymes of this pathway. This pattern has only been reported in a few animal lineages, including some nematodes and ctenophores. We adjusted the text in both results and discussion to clarify this and now state that: “Noticeably, trans-splicing is present in ctenophores, chaetognaths, and nematodes^{68,70}, the three animal phyla where gene body methylation has been repurposed for transposable element targeting. This observation suggests a recurrent evolutionary process in which trans-splicing compensates for the loss of gene body methylation. However, other evolutionary forces may also drive the loss of gene body methylation, such as the DNA repair burden imposed by DNA methyltransferases (DNMTs)⁷⁰.”

The relationship between gene body methylation and genome size remains unclear, largely because we have few examples of this type of genomic simplification. Although complete loss of cytosine methylation is common, it does not consistently correlate with smaller genomes. For instance, in nematodes—where some species exhibit TE-methylation and others have no methylation at all—there is no obvious link. *Trichinella spiralis* and *Romanomermis culicivorax* retain TE-methylation with genomes of approximately 55 Mb and 322 Mb respectively, whereas *C. elegans*, with a genome around 100 Mb, shows no cytosine methylation. Similarly, among chaetognaths there is extensive variation in genome size, with *Paraspadella* possessing a relatively small genome of about 200 Mb. Only by analysing the genomes and epigenomes of additional chaetognaths will we be able to determine whether exclusive TE methylation leads to genome expansion or, conversely, effectively controls TEs and streamlines the genome.

16) Fig 5b, I have checked different displays, and I have a hard time finding the difference between shades.

The differences in shades were here to mark the fact that some paralogues retain the ancestral domain composition while others underwent domain simplification. We have now tried to make these shades more apparent. For instance, all TET copies lack zf-CXXC and are therefore colored as a lighter green. We rephrased the figure legend as follow: “Paralogues with simplified domain architectures are coloured in a lighter shade than those that retained the ancestral state.”

17) Trans-splicing findings are fascinating. Due to its prevalence in the chaetognaths genome (50% of the genes), I understand that most of the genes that are trans-spliced in chaetognaths

are not in other taxa, is that correct? How many of the chaetognath-specific genes are trans-spliced, or is this a “feature” of older genes?

The review is indeed raising a very interesting point. We found that genes from all phylostrata can be operonic and trans-splicing, as shown in **Extended Data Fig. 7g** (formerly **8g**), however, the oldest genes (metazoan and older) have a higher fraction of trans-spliced genes, likely related to the increased of trans-splicing of housekeeping genes that are very broadly conserved. We added the sentence “Trans-spliced genes are also not restricted to a phylogenetic origin and belong to all phylostrata (**Extended Fig. 7g**).” in the corresponding section.

18) Page 13, line 22, I believe the sentence should be “provides clues to the origin” instead of “provides clues in the origin”.

Corrected.

Referee #4:

Review of manuscript by Piovani et al. “The genomic origin of the unique chaetognath body plan”

Summary of key results, originality, and significance:

Piovani et al. present an impressive genomic study of the enigmatic phylum Chaetognatha, integrating a range of disciplines and frontier methodologies to investigate its origin at the genomic, regulatory, and cellular levels. The phylogenetic position of this old and ecologically important pelagic group was long debated, with the group showing both deuterostome and protostome characteristics. However, its position within Spiralia has now been established through several recent phylogenomic studies. In the context of this phylogenetic position, Piovani et al. now provide a timely and highly relevant, deeper insight into the evolutionary history and function of the aberrant genomic and morphological traits of Chaetognatha.

The study integrates information from organismal to cellular level, from chromosomal to functional genomic levels, and boldly propose new links between geno- and phenotypes. Overall, the study presents a wealth of original data on this understudied group, which allows for highly interesting comparisons across Metazoa and for erecting new hypotheses – not only on the origin of Chaetognatha but on general genomic evolutionary mechanisms underlying adaptations of animal body plans. In sum, the study has the novelty and potential of high relevance and vast impact on a broad readership from many different scientific disciplines.

Conclusions, data and methodology:

The many different analyses generally represent state of the art (or beyond) within their disciplines and the breadth and depth of results are impressive (e.g. on centromeric structures, gene loss and duplications, and specific gene functions), especially considering that this non-model organism has not previously been studied in any such details. Still, I have a few concerns on central parts of the work (see below under methods/results) and on the phrasing of some of the conclusions.

We thank the reviewer for their positive appreciation of our work and suggestions to improve the manuscript. We attempted in this revision to clarify the points of concerns from the reviewer.

Piovani et al. keeps referring to Gnathifera as a clade including Chaetognatha, although most phylogenomic studies (including Piovani et al.) shows Chaetognatha to be sister to Gnathifera (see and cite Laumer et al. 2019, <https://doi.org/10.1098/rspb.2019.0831>).

Although it can seem as barely semantics, there are very good arguments in Bekkouche & Gasiorowski 2022 (<https://doi.org/10.1080/14772019.2022.2109217>) for instead using the name Chaetognathifera for this common clade. I strongly recommend the authors to replace the name Gnathifera with Chaetognathifera when referring to this clade throughout the manuscript. This will also make it easier for the reader to follow when they are referring to Gnathifera 'sensu stricto' vs Chaetognathifera.

We understand the rationale behind proposing Chaetognathifera as a clade name as there is a possibility that classical gnathiferan clades (rotifers, micrognathozoa, gnathostomulids) could be monophyletic in regard to chaetognaths, even if some molecular analysis suggested that chaetognaths could be more closely related for instance to gnathostomulids. However, we feel that in general, it is simpler for the wider audience (including students) to use Gnathifera, a clade name that has recently been adopted in textbook (see for instance in the 4th edition of the *Invertebrates* by Brusca, Giribet & Moore, 2022). The usage of molecular phylogeny to re-evaluate the position of existing clades has often caused the reuse of older names (e.g. spiralian include non-spiral cleaving lineages such as brachiopods or sipunculids now included in molluscs). We also find that the usage of agglomerative terms, such as Chaetognathifera can be confusing, and is better avoided by abiding by cladist principles that clades should be defined by a synapomorphy. In the case of Gnathifera, there is such a synapomorphy, namely the presence of hardened mouthpieces, Bekkouche & Gasiorowski argue that jaws of chaetognaths and gnathiferans *stricto sensu* should not be homologised as they are external and “form not a pincer structure, but instead a grasping apparatus”. We find that this view is rather restrictive, as differences in size and ecology between the lineages, especially in the absence of molecular definition of these jaws. Moreover, the *Timorebestia* fossil is interpreted by Park et al. (2024) as a chaetognath and has internal jaw apparatus, which would therefore relate the jaw apparatus of chaetognaths to that of other gnathiferans. However, we keep an open mind about these issues, we added the following statement to emphasise the possibility that chaetognaths could be the sister-group of other gnathifera : “The internal relationships between rotifers, gnathostomulids and chaetognath remain ambiguous, both from a molecular^{5,6} and morphological²³ standpoint, with chaetognaths most likely positioned as sister-group to other gnathiferans²³..”

The Piovani et al. references to Chaetognathifera evolution seems a bit one-sided and B&G 2022 should as a minimum be discussed and cited, since it convincingly positions *Amiskwia* next to extant Chaetognatha (in Cuculophora) and thoroughly reviews the morphological characters and their evolution, such as the likely lack of homology between Chaetognatha grasping spines (by the authors called 'hooks'?) and gnathiferan jaws. Instead Piovani et al. multiple times refer (only) to the Vinther & Parry 2019 paper, which first was followed by a more detailed fossil study (Caron & Cheung 2019) and later was shown to include multiple erroneous character codings and likely misinterpretations of the evolutionary history (e.g., Bekkouche & Gasiorowski 2022).

As discussed below, we agree that Bekkouche & Gasiorowski (2022) is raising very interesting points, we now referenced Caron & Cheung 2019 and B&G 2022 and nuanced our discussion of character evolution:

“The paucity of Cambrian fossils for other gnathiferans such as rotifers or gnathostomulids, and the relative large body size of fossils such as Amiskwia interpreted as stem group

*chaetognaths*²² or more convincingly *gnathostomulids*⁸⁶ does not provide paleontological support for small body-size gnathiferan ancestor. The internal relationships between rotifers, *gnathostomulids* and *chaetognath* remain ambiguous, both from a molecular^{5,6} and morphological²³ standpoint, with *chaetognaths* most likely positioned as sister-group to other *gnathiferans*²³. Further characterisation of genomes and gene expression in other *gnathiferan* lineages (e.g. *micrognathozoans* or *gnathostomulids*) would help to further resolve organismal and genomic evolution in the *gnathiferan* lineage.”

Moreover, Piovani et al. base some of their conclusions on the assumption that the Chaetognathifera lineage has undergone genomic miniaturization before the expansion/duplications found in Chaetognatha. It is unclear to me how they can draw this conclusion so solidly from their data when they mainly compare to a highly aberrant (according to themselves) bdelloid rotiferan genome and miss data from several of the other *gnathiferan* lineages (as well as other Spiralian lineages). This should be further explained as some of the main conclusions rest on these results.

Related to this, Chaetognathifera is now regarded (also by the authors) as a clade sister to the remaining Spiralia and to my understanding it is still equally possible that the Chaetognathiferan ancestor may have possessed ancestral Spiralian genomic properties rather than a miniaturized design. The authors do mention this caution in the conclusion, but less throughout the manuscript. It seems uncouthly to give this uncertainty a bit more weight in the discussion, and then put less emphasis on the proposed pre-miniaturization/losses in Chaetognathifera in the conclusion - without devaluating the main conclusions /hypotheses on the genomic evolution of Chaetognatha.

We would like to emphasise that this was merely an interpretation of our observations (we stated “we propose...”). We now insist more explicitly that such divergence was related “either as a single event or as multiple lineage specific events” to account for the fact that we do not assign genomic character evolution to a specific node.

Our observations are based on a comparison of the data actually available so far. We have incorporated the genome of the two available rotifers *Brachionus* and *Adineta* in our gene family analyses, which represent the two main rotifer lineages (monogonont and bdelloids) split by significant divergence. The *Adineta* genomes is the only one scaffolded at chromosome scale (Simion *et al.* 2021) and that’s why we used it primarily for macrosyntenic comparisons. We wouldn’t necessarily call it aberrant, but we agree with the reviewer that it might not be representative of all rotifer genomes. We actually also in this revised version investigated the conservation of ancestral linkages in the genome of *Brachionus manjacavas* which is not chromosome-scale but is fairly contiguous, and we found that while it seems less rearranged than *A. vaga* (more ALGs can be detected), it shows no syntenic conservation with *chaetognaths*, and also underwent extensive genomic fusions.

This plot has now been included as panel **Extended Data Figure 3f**.

However, regarding the gene family analysis, we believe that our inference of genes lost in the Gnathifera (or Chaetognathifera as the reviewer prefers) based on proteomes of from several chaetognath and these two rotifers is sufficient to detect the occurrence of gene loss at the origin of gnathiferans.

Likewise, the text in between closely link the small body size of gnathiferan taxa to their genome size, and gene loss to genome miniaturization, dependencies that are not always clear or established (e.g., also substantial gene loss at other nodes, suppl. Fig. 2e,d). Furthermore, the calculated gene losses for the Chaetognathifera lineage likely depend on the limited number of included gnathiferans and might suffer from the missing Gnathostomulida and other minor spiralian phyla, and potentially be affected by the divergent position (and paraphyly) of Platyhelminthes. This is apparently not discussed at present and might deserve more attention. More detailed comments on specific analyses are given below.

As stated above, we were merely hypothesising an episode of miniaturisation as one of the factors that could have contributed to derived genome evolution in this lineage. Nevertheless, to assess the impact of the topology of the spiralian tree on the reconstructed gene gain and loss events, we attempted similar inferences of gene gains, losses and duplications assuming the 'Rouphozoa' topology where platyhelminth and ectoprocts are sister-group of other spiralians (see below) and we did not observe that it had a major impact on our observations regarding gene family evolution, and particularly, the presence of a burst of gene gain and duplication in chaetognaths and the occurrence of gene losses in the gnathiferan lineage.

This panel is now included in Supplementary Figure 1.

The centromere discussion on page 4 could benefit from including more comparative information on rotifers. Topology of Figure 4 needs further explanation, the Parenchymia and Tetraneuralia clades on Figure 4 are quite controversial.

We added more details regarding centromeres in rotifers, which have only been discussed in *A. vaga*: “Unlike chaetognath, the rotifer *A. vaga* likely possesses holocentromeres^{29,41,42}, suggesting that centromeres have broadly diverged in the gnathiferan lineage and have been replaced in chaetognaths by neocentromeres making use of alternative molecular components.”

Regarding Figure 4, we used the topology of Marlétaz et al. 2019 as specified but we agree with the reviewer that some of these clades are not fully accepted and that the presentation of the figure was putting an unnecessary emphasis on them. We revised and simplified Figure 4 and removed clade names to avoid any confusion.

Specific comments to methodology, data:

Contamination: One potentially major issue is that the authors seemingly do not perform any decontamination steps. This should be standard for all new genome assemblies but particularly because this assembly is primarily an Illumina based assembly. Using Blobtools could be an option. Since the animals were lab bred and fed exclusively on *Tigriopus californicus*, it would be easy to check specifically for contamination from food sources in addition to the standard Blobtools decontamination pipeline. It is recommended to perform this step prior to duplicate purging and scaffolding. There is ample genomic data for *T. californicus* on SRA that can be used for this. Were the animals starved before extraction? Maybe a little more detail is needed here in the methods.

We agree with the reviewer that decontamination is important, and we apologise for not providing more details regarding this in this version of manuscript. We originally ran some contamination checks including a Blobtools analysis (see below) which were not included (apologies) but are now fully referenced in the text, and indicated only a few contigs as putative contaminants which were excluded from the initial assembly. However, given the isolated phylogenetic position and our finding regarding presence of novel genes, homology-based decontamination easily gives

contentious results as some chaetognath genes will match chordates and others will match arthropods (which are very abundant in sequence databases).

Figure R3: Blobtools analysis of initial contigs of the *P. gotoi* assembly (arrow pointing to removed contigs)

We appreciate the recommendation of the reviewer regarding the genome of *Tigriopus*. We double-checked and realised that the species of crustacean with which the *P. gotoi* were fed was *Tigriopus japonicus* and not *californicus* (we adjusted the text accordingly, again sorry for the confusion). We then performed an alignment between *P. gotoi* and the available *T. japonicus* genome using minimap2 and only recovered a few very short alignments (<5kb) that do not indicate a contamination of *T. japonicus* sequence in the *P. gotoi* assembly.

Finally, we would also like to stress the fact that the phylostratigraphic analysis using GenEra provides a list of ‘discordant’ genes that could be genomic contamination, but only a limited number are flagged as such (288 genes in total) and only 62 of these genes are listed among chaetognath novel genes (62 out of 1832).

Potentially uncollapsed heterozygosity and strange Kmer spectrum: The authors accounted for heterozygosity in the experimental design by inbreeding for five generations. However, even if the loss of heterozygosity from inbreeding was “perfect”, after five generations of full sib mating, you’d still expect there to be ~3% of the heterozygosity of the original parent (50% loss heterozygous sites per generation for five generation = 0.5^5). Further, percent heterozygosity of the genome predicted was not reported. The authors admit chaetognaths are expected to have high heterozygosity so there might not be enough generations to effectively control it in a pool of 20 individuals. Because of this issue, the lack of a haplotype purging step is problematic. The assembler was set up to merge haplotypes but assemblies generally still require/benefit from a devoted haplotype purging step. There are some concerns regarding the kmer spectrum in supplementary figure 1. There is no heterozygous peak but there is a long “tail” where the peak is expected (~50x coverage). You’d expect the count of the single copy kmers to approach zero whereas in Piovani et al., the count levels off at $\frac{1}{2}$ the count of the homozygous peak. This might indicate that the kmer size is incorrect (maybe play around with this value a little) or more worryingly, there are uncollapsed haplotypes that were scaffolded in tandem. Purge_dups are recommended on the decontaminated assembly prior to scaffolding. If there are uncollapsed haplotypes that made it into the final assembly, this could seriously affect some of the major conclusions of the paper (i.e. gene duplication/gain, hox gene analysis).

We now provide further details regarding the way we processed heterozygosity. We particularly indicated in the method section the heterozygosity predicted from remapping reads on the assembly is 1.49% and the heterozygosity estimated using Genoscope2 (assuming a diploid assembly) is 0.66%.

We agree with the reviewer that using a tool such as purge_dups can be a good approach to remove residual haplotypes when DNA is sequenced from a single individual with two divergent alleles. Generally, there are two distinct approaches to deal with highly heterozygous genomes, one is to sequence a single individual, run the assembly in stringent conditions to split haplotypes as best as possible, and then purge the allelic variation, the other is to try to merge both alleles at the assembly stage. In our case, as multiple individuals were mixed and crossed for several generations, we expected that residual polymorphism would correspond to relatively low frequency alleles. We tried both approaches (splitting and merging) and found that attempting to merge polymorphisms was providing the best assembly, as it likely took advantage of the overall reduction of polymorphism. Hence, we ran the meraculous assembler in a mode aimed at merging alleles (mode 2) and indeed recovered an haploid assembly validated by a number of QCs:

- busco score does not show a high residual rate of duplication - even it could be biased by the chaetognath specific duplication (C:94.3% [S:91.3%, D:3.0%], F:1.3%, M:4.4%).
- the distribution of per-base coverage indicates a single-peak (the slight skew to the left is likely due the uneven representation of alleles in the sequenced population and the final assembly).
- during the Hi-C scaffolding process, we did not observe residual haplotypes, which are usually clearly observable as brought together by the HiC scaffolding tools.
- k-mer plot shows that most k-mer are not presented in >1 copies in the assembly.

When we performed an assembly in 'split' mode, we did not recover an assembly with two distinct k-mer coverage peaks, but increased the representation of low frequency alleles, without a clear per-base coverage distribution that would have enabled an efficient application of `purge_dups`. Finally, we attempted run `purge-dups` to the contigs of the present assembly, it was very difficult to define the coverage threshold due to the single coverage peak, and attempting to set them up manually at half coverage led to very limited effect (<3Mb of sequenced removed - confirming that our assembly is haploid).

Figure R4: per-base coverage in the primary contigs of *P. gotoi* assembly

Regarding k-mer spectrum, we realised that we mistakenly showed the 'cumulated' k-mer frequency instead of showing the regular k-mer frequency which is somehow confusing and more difficult to interpret. More generally, we interpret the presence of an elbow in the k-mer spectrum as indicative of the presence of low frequency residual polymorphisms because a singular half coverage peak would only appear in case of a diploid individual. The included version of the k-mer plot shows the current assembly is devoid of low frequency k-mer and double copy k-mers.

The phrasing of the methods might have added to the confusion: we say that the inbred line was reconstructed from a single individual, but as chaetognaths are cross-fertilising hermaphrodites and are known in the wild to store sperm from several mates, there are at least 4 alleles in the initial allelic pool: 2 of the parent, and ≥ 2 from the sperm used to internally fecund the eggs. This would be consistent with the absence of a low coverage peak in the k-mer spectrum.

The reviewer also raises the possibility that residual haplotypes could have been assembled in tandem and that this could explain our results on gene duplication, particularly for the Hox. We think that the actual divergence between the gene sequence (see for instance Hox tree in **Supplementary Figure 3**) as well as the presence of shared duplicates in other highly diverging chaetognath species (represented by transcriptomes, see **Figure 1**) indicates that this is not the case. To confirm this, we also generated a dotplot (from a lastz alignment) for the Hox cluster region, which shows that it does not contain any tandem repeats:

Figure R5: Lastz self-alignment of Hox cluster region (delimited by red lines)

Analysis of gene gain and loss: The section describing the use of GenEra for phylostratigraphy should be expanded on to include the parameters used for the program.

We used GenEra defaults parameters and blasted against the ncbi nr database. We added the following to the methods:

“For complementary phylostratigraphic analyses, we used GenEra (v1.0.2) with default options using the ncbi NR database for the blast search.”

In Supplementary Figure 2A, CAFÉ was used, however there are no methods to report what parameters were used for the input tree (ultrametric tree production, fossil calibrations, etc) or for the running of the program (i.e. gamma rate or error correction).

Apologies for not provided the necessary details on the CAFÉ analysis, we now added the following in the discussion:

“For the CAFÉ analysis, a dated tree was obtained by extracted phylogenetic markers described in ref. ⁵ for the selected proteomes included in the analyses. An ultrametric dated tree was reconstructed using Phylobayes (v4.1e) assuming a CAT+GTR model, a CIR relaxed clock model, birth-death priors on divergence times (0.5) and the following calibrations (lower-upper bounds in Myr): euarthropods (531-514), ecdysozoans (636-519), Gnathifera (636-518.5), bilaterians (636-521) and deuterostomes (635-516)¹⁰⁵⁻¹⁰⁷. CAFE (v5.1) was then applied using gene families previously defined from Broccoli assuming a gamma model with one rate category (k=1) and no error model¹⁰⁸.”

In situ's and Hox genes: Page6, l39-l44 authors compare "chaete" cell type with "epidermal" cell type both expressing paralogues of chitin synthase and chitinase. Chs2-5 is located close to the grasping spines in Figure2d. But it is less clear in the text that Vwf-2 was used as a marker of the "epidermal" cell type expressing chitin synthase and chitinase. This should be stated more clearly in the text. Based on the presence of several epidermal cells, and literature on the pluristratified epidermis in chaetognath, the Vwf2 gene, showing faint and uniform signal, might not be the best to label the epidermis cells expressing chitin synthase and chitin. Instead, the epidermal marker Chs2-3 might better serve as comparison with "chaete" cells marked with Chs2-5 and the argument of a spine synthesis function more likely.

We apologise for the unclarity of this section. We actually didn't mean to compare epidermal cells and chaetal cells, we were simply describing the expression of the markers we choose for each cluster (Chs2-5 and Vwf-2) and pointing out that both epidermal and chaetal cells express chitin synthase and chitinases. We have now rephrased the text as below and hope that this is clearer.

"The expression of chitin synthase (Chs2-5) near chaetognath grasping spines indicates these are likely ensuring the growth and synthesis of the grasping spines (cluster 18) while the marker gene Vwf-2 (clusters 2,6 and 17) and lfea-2 (clusters 8,20) show expression in the epidermal layer (Figure 2d). The presence of multiple epidermal cell types is consistent with the presence of a complex pluristratified epidermis in chaetognaths, a condition rarely found outside of vertebrates⁵⁴. All of these structural cell-types express various chitin synthase and chitinases, with alternative paralogues present in the 'chaete' cells (clusters 18, Chs2-5, Chi10-1, Chi10-3 and Chi10-4) and epidermal cells (clusters 2,6 and 17; Chs2-3, Chit1-7, Chit1-8)."

In Figure 4 the authors show Hox genes in situ performed on juvenile (in legend)/ hatchling (in text). Authors should clarify the stage as it is inferred from the method that a juvenile would be a 3dph. Moreover, the relevance of the life stage used to show Hox expression is seemingly not justified, neither by morpho-developmental nor gene expression data. Please elaborate.

Could the authors use the RNAseq dataset to provide a temporal expression of the Hox at gastrula, hatchling and 3dph that could support the choice of life stage for In situ's?

We thank the reviewer for spotting the inconsistency in stage description, the newly hatched animals have been called juveniles in the past (Yasuda et al., 1997), we have now changed the text to clarify that in situs were performed on 3dph (juvenile) in both legend and text. We didn't perform ISH at earlier stages as the eggshell stops the probes from reaching the animal and treatment with sodium thioglycolate as described in (Takada et al., 2002) didn't work in our experience.

Piovani et al. claim that MedPost is expressed in papillae peripheral sensory neurons, showing a new function, and they do show in Supl. Figure 9 that MedPost is expressed in cluster cell Papillae-15. However, Supl. Figure 9 also shows that MedPost is expressed in epidermis_20. The authors should provide a full view of Pxdn-9 in situ in Fig. 2 to support the claim that MedPost signal is in papillae peripheral sensory neurons throughout the body and explain the seemingly double signal in "epidermis-20" cluster cells.

We thank the reviewer for their comment, indeed what we meant to say is that MedPost is expressed in both papillae peripheral sensory neurons (which is apparent from both the *in situ* and is confirmed by the signal in the scRNA data) as well as in some epidermal cells. This is quite common with hox genes which often specify larger domains that sometimes encompass more than one cell type. To clarify these points we have changed the text as below and also provide a full worm for Pxdn-9 in **Figure 2**.

“Hox expression in our single cell sequencing datasets shows scattered expression in several cell types, mainly epidermal, neuronal and papillae cell types (**Extended Data Fig. 9b**). To further characterise their expression we performed *in situ* hybridisation on six Hox genes in *P. gotoi* juveniles (**Figure 4b** and **Supplementary Table 7**).”

“Similarly to what have been observed in other lineages, some tandemly duplicated Hox genes acquired cell-type specific expression domains, while others maintained their broad patterning roles”

Adinetta should be Adineta (spelled wrong throughout figures)

This has been corrected

References cited

- de Rosa R, Grenier JK, Andreeva T, Cook CE, Adoutte A, Akam M, Carroll SB, Balavoine G. 1999. Hox genes in brachiopods and priapulids and protostome evolution. *Nature* **399**:772–776. doi:10.1038/21631
- Gąsiorowski L, Martín-Durán JM, Hejnal A. 2023. The evolution of hox genes in Spiralia Hox Modules in Evolution and Development. Boca Raton: CRC Press. pp. 177–194. doi:10.1201/9781003057215-9
- Kenny NJ, McCarthy SA, Dudchenko O, James K, Betteridge E, Corton C, Dolucan J, Mead D, Oliver K, Omer AD, Pelan S, Ryan Y, Sims Y, Skelton J, Smith M, Torrance J, Weisz D, Wipat A, Aiden EL, Howe K, Williams ST. 2020. The gene-rich genome of the scallop *Pecten maximus*. *Gigascience* **9**. doi:10.1093/gigascience/giaa037
- Laumer CE, Fernández R, Lemer S, Combosch D, Kocot KM, Riesgo A, Andrade SCS, Sterrer W, Sørensen MV, Giribet G. 2019. Revisiting metazoan phylogeny with genomic sampling of all phyla. *Proc Biol Sci* **286**:20190831. doi:10.1098/rspb.2019.0831
- Li J, Wang J, Zhang P, Wang R, Mei Y, Sun Z, Fei L, Jiang M, Ma L, E W, Chen H, Wang X, Fu Y, Wu H, Liu D, Wang X, Li J, Guo Q, Liao Y, Yu C, Jia D, Wu J, He S, Liu H, Ma J, Lei K, Chen J, Han X, Guo G. 2022. Deep learning of cross-species single-cell landscapes identifies conserved regulatory programs underlying cell types. *Nat Genet* **54**:1711–1720. doi:10.1038/s41588-022-01197-7
- Marlétaz F, Peijnenburg KTCA, Goto T, Satoh N, Rokhsar DS. 2019. A New Spiralian Phylogeny Places the Enigmatic Arrow Worms among Gnathiferans. *Curr Biol* **29**:312–318.e3. doi:10.1016/j.cub.2018.11.042
- Martín-Zamora FM, Liang Y, Guynes K, Carrillo-Baltodano AM, Davies BE, Donnellan RD, Tan Y, Moggioli G, Seudre O, Tran M, Mortimer K, Luscombe NM, Hejnal A, Marlétaz F, Martín-Durán JM. 2023. Annelid functional genomics reveal the origins of bilaterian life cycles. *Nature* **615**:105–110. doi:10.1038/s41586-022-05636-7
- Matus, Halanych KM, Martindale MQ. 2007. The Hox gene complement of a pelagic chaetognath, *Flaccisagitta enflata*. *Integr Comp Biol* **47**:854.
- Nowell RW, Rodriguez F, Hecox-Lea BJ, Mark Welch DB, Arkhipova IR, Barraclough TG, Wilson CG. 2024. Bdelloid rotifers deploy horizontally acquired biosynthetic genes against a fungal pathogen. *Nat Commun* **15**:5787. doi:10.1038/s41467-024-49919-1
- Papillon D, Perez Y, Fasano L, Le Parco Y, Caubit X. 2003. Hox gene survey in the chaetognath *Spadella cephaloptera*: evolutionary implications. *Dev Genes Evol* **213**:142–148. doi:10.1007/s00427-003-0306-z
- Piovani L, Leite DJ, Yañez Guerra LA, Simpson F, Musser JM, Salvador-Martínez I, Marlétaz F, Jékely G, Telford MJ. 2023. Single-cell atlases of two lophotrochozoan larvae highlight their complex evolutionary histories. *Sci Adv* **9**:eadg6034. doi:10.1126/sciadv.adg6034

- Rao SSP, Huntley MH, Durand NC, Stamenova EK, Bochkov ID, Robinson JT, Sanborn AL, Machol I, Omer AD, Lander ES, Aiden EL. 2014. A 3D map of the human genome at kilobase resolution reveals principles of chromatin looping. *Cell* **159**:1665–1680. doi:10.1016/j.cell.2014.11.021
- Rowley MJ, Poulet A, Nichols MH, Bixler BJ, Sanborn AL, Brouhard EA, Hermetz K, Linsenbaum H, Csankovszki G, Lieberman Aiden E, Corces VG. 2020. Analysis of Hi-C data using SIP effectively identifies loops in organisms from *C. elegans* to mammals. *Genome Res* **30**:447–458. doi:10.1101/gr.257832.119
- Sebé-Pedrós A, Chomsky E, Pang K, Lara-Astiaso D, Gaiti F, Mukamel Z, Amit I, Hejnol A, Degnan BM, Tanay A. 2018. Early metazoan cell type diversity and the evolution of multicellular gene regulation. *Nat Ecol Evol* **2**:1176–1188. doi:10.1038/s41559-018-0575-6
- Takada N, Goto T, Satoh N. 2002. Expression pattern of the Brachyury gene in the arrow worm *Paraspadella gotoi* (Chaetognatha). *Genesis* **32**:240–245.
- Wilson CG, Pieszko T, Nowell RW, Barraclough TG. 2024. Recombination in bdelloid rotifer genomes: asexuality, transfer and stress. *Trends Genet* **40**:422–436. doi:10.1016/j.tig.2024.02.001
- Yasuda E, Goto T, Makabe KW, Satoh N. 1997. Expression of actin genes in the arrow worm *Paraspadella gotoi* (Chaetognatha). *Zoolog Sci* **14**:953–960. doi:10.2108/zsj.14.953

Referees' comments:

We would like to thank again the referees for their invaluable comments and support which helped to improve our manuscript throughout the reviewer process.

Referee #1 (Remarks to the Author):

The authors have comprehensively addressed all my concerns and suggestions. This is an excellent study and I fully endorse its publication in Nature.

We thank the reviewer for their support.

Referee #2 (Remarks to the Author):

The authors have thoroughly addressed our previous comments in this revision. We commend their success in resolving the mystery of their unique body plan through an integrative genomic and multi-omic approach. We have no further major concerns, but would like to offer a few minor corrections and suggestions to improve clarity and presentation.

We thank the reviewer for their support.

Line 47 – lophotrochozoans -> lophotrochozoan

Line 54 – bodyplans -> bodyplans

Line 86 – it says “rotifers and to a lesser extent chaetognaths” but the numbers show that it should be to a greater extent in chaetognaths.

Line 142 – domain -> domains

Line 222 – the abbreviation is no longer in the figure

Line 304 – there is at least one further case of Hox gene duplication: the zen/shx gene of lepidopterans, which can have up to 165 copies: See Mulhair et al. 2023

(<https://genome.cshlp.org/content/33/1/32.full>) and Mulhair et al. 2024

(<https://www.sciencedirect.com/science/article/pii/S1084952122003573>).

We made the corresponding adjustments. We agree that the case of lepidopteran Hox expansion is quite amazing, but we had to limit the reference cited for concision, we now refrain from making any direct comparison with Hox numbers in other animals.

Referee #3 (Remarks to the Author):

I enjoyed reading the revised manuscript by Piovani and collaborators, and I'd like to thank the authors for addressing my concerns.

About annotating lost genes in chaetognaths, I'd usually prefer to use a closer relative to minimise the impact of the divergence time. The amphioxus-chordates-deuterostomes nodes may have their own gene losses obfuscating the annotation. In the immortal words of Philip H. Pope (1921), "it's a long way from amphioxus" (apologies, this is my best comeback to the Haldane quote). That

said, I understand that the “closest” genomes available are very messy, and cephalochordates are a reasonable compromise.

I believe the manuscript has significantly improved and I support its publication.

We thank the reviewer for their support.

Regarding amphioxus, we actually agree that relying on a comparison with amphioxus to annotate Hox genes was too limited, and that's why we decided to use a broader gene-family-based approach.

Referee #4 (Remarks to the Author):

I would like to congratulate Piovani et al. for having done an amazingly thorough and comprehensive revision as well as a well-argued and detailed response to reviewer letter. I believe their impressive and highly interesting work shines even brighter now :-). Main figures have been very nicely revised and now reads clearly. Additional analyses (and additional suppl. figures), additional methodological explanations, and text rephrasing further supported their findings and refined their conclusions. The few raised concerns rebutted, are well argued for in their response letter. I have no further comments, and can only highly recommend publication of this very interesting and intriguingly integrative study.

We thank the reviewer for their support.

A few minor typos caught:

line 47: lophotrochozoans to lophotrochozoan

line 54: bodyplans to body plans

line 87 and 88: '1456 and 2165, respectively' '116 and 172' - switch order of figures to reflect previous line giving order rotifer, chaetognath, respectively

We made the corresponding adjustments.